# Analysis of Sensing Spectral for Signal Recovery Under a Generalized Linear Model

**Junjie Ma**
Academy of Mathematics and Systems Science
Chinese Academy of Sciences
majunjie@lsec.cc.ac.cn

**Ji Xu**
Department of Computer Science
Columbia University
jixu@cs.columbia.edu

**Arian Maleki**
Department of Statistics
Columbia University
arian@stat.columbia.edu

## Abstract

We consider a nonlinear inverse problem $\boldsymbol{y} = f(\boldsymbol{Ax})$, where observations $\boldsymbol{y} \in \mathbb{R}^m$ are the componentwise nonlinear transformation of $\boldsymbol{Ax} \in \mathbb{R}^m$, $\boldsymbol{x} \in \mathbb{R}^n$ is the signal of interest and $\boldsymbol{A}$ is a known linear mapping. By properly specifying the nonlinear processing function, this model can be particularized to many signal processing problems, including compressed sensing and phase retrieval.

Our main goal in this paper is to understand the impact of sensing matrices, or more specifically the spectrum of sensing matrices, on the difficulty of recovering $\boldsymbol{x}$ from $\boldsymbol{y}$. Towards this goal, we study the performance of one of the most successful recovery methods, i.e. the expectation propagation algorithm (EP). We define a notion for the spikiness of the spectrum of $\boldsymbol{A}$ and show the importance of this measure in the performance of the EP. Whether the spikiness of the spectrum can hurt or help the recovery performance of EP depends on $f$. We define certain quantities based on the function $f$ that enables us to describe the impact of the spikiness of the spectrum on EP recovery. Based on our framework, we are able to show that for instance, in phase-retrieval problems, matrices with spikier spectrums are better for EP, while in 1-bit compressed sensing problems, less spiky (flatter) spectrums offer better recoveries. Our results unify and substantially generalize the existing results that compare sub-Gaussian and orthogonal matrices, and provide a platform toward designing optimal sensing systems.

## 1 Introduction

### 1.1 Problem statement and contributions

Consider the problem of estimating a signal $\boldsymbol{x} \in \mathbb{R}^n$ from the nonlinear measurements:

$$\boldsymbol{y} = f(\boldsymbol{Ax}), \tag{1}$$

where $\boldsymbol{A} \in \mathbb{R}^{m \times n}$ is a measurement (or sensing) matrix and $f : \mathbb{R} \mapsto \mathcal{Y}$ is a function accounting for possible nonlinear effect of the measuring process. Here, the function $f(\cdot)$ is applied to $\boldsymbol{Ax}$ in a component-wise manner. The above model arises in many applications of signal processing [13, 10, 41], communications [56, 9, 25], and machine learning [48, 40]. For instance, the phase retrieval problem, which is a special case of (1) with $f(z) = |z|$, has received significant interest in recent years [13, 12, 15, 54, 60, 24, 26, 4, 46, 32, 5]. In this paper, we assume that the signal is generic and prior information such as sparsity is not explored.

35th Conference on Neural Information Processing Systems (NeurIPS 2021).

The main goal of our work is to understand the impact of the sensing matrix, or more specifically the spectrum of the sensing matrix, on the difficulty of recovering the signal $x$ from its measurements $y$. In many applications, one has certain level of freedom in designing the sensing matrix (e.g., transmitter design in communications or the masks used in phase retrieval application) and hence understanding the impact of the sensing matrix on the recovery algorithms is the first step toward the optimal design of such systems. Rather than studying the information theoretic limits, where the computational complexity of the recovery algorithm is ignored, we would like to study the impact of the spectrum of the sensing matrix on efficient algorithms that are used in applications. For this reason, we consider one of the most successful recovery algorithms that has received substantial attention in the last few years, i.e. expectation propagation (EP) [36, 38] (referred to as GLM-EP in this paper[1]), and study the impact of the spectrum of the sensing matrix on the performance of this algorithm. The EP algorithm studied here is an instance of the algorithm introduced in [22, 28] and is closely related to the orthogonal AMP (OAMP) [30] and vector AMP (VAMP) [42] algorithms (in that all these algorithms use divergence-free denoising functions [30]).

Similar to the approximate message passing (AMP) algorithm [17], GLM-EP has two distinguishing features: (i) Its asymptotic performance could be characterized exactly by a simple dynamical system (with very few states) called the state evolution (SE). (ii) It is conjectured that AMP or GLM-EP achieve the optimal performance among polynomial time algorithms [2, 14]. Based on the SE framework, we investigate the impact of the spectrum of the sensing matrix $A$ on the performance of GLM-EP. It turns out that the "spikiness" (or conversely "flatness") of the spectrum of the sensing matrix spectrum has a major impact on the performance of GLM-EP. To formalize this statement, we first define a measure of "spikiness" of the spectrum based on Lorenz partial order [1]. We show that whether the spikiness of the spectrum benefits or hurts GLM-EP depends on the choice of the nonlinear mapping $f$ (as well as the sampling ratio). For instance, spikier spectrums help the performance of phase retrieval problem (where $f(x) = |x|$) but hurt the performance of 1-bit compressed sensing (where $f(x) = \text{sign}(x)$). We will characterize the classes of functions on which spikiness hurts or helps GLM-EP based on the monotonicity of a function (which is related to the scalar minimum mean square error) that will be defined in this paper. As a byproduct of our studies, we will also show that when the spectrum is spiky enough, the number of measurements required by GLM-EP to achieve perfect recovery approaches the information theoretical lower bound.

## 1.2   Related Work.

Message passing algorithms [17, 7, 8, 41, 6, 45, 49, 11, 20, 21, 36, 38, 31, 30, 22, 47, 42, 51, 28, 52, 23, 50] have been used extensively for solving the estimation problems similar to the one we have in (1). As a result of such studies, it is known that partial orthogonal matrix is better than iid Gaussian matrix for noisy compressed sensing [31], and the spectral methods for phase retrieval perform better with iid Gaussian sensing matrices than coded diffraction pattern matrices [29, 37, 33, 19]. However, studying the impact of spectrum of the sensing matrix in the generality of our paper has not been done to the best of our knowledge. Recently, [35] considered the phase retrieval problem and a sensing matrix which can be written as the product of Gaussian and another matrix. They reached the conclusion that the weak recovery threshold with this type of matrices can be made arbitrarily close to zero. As a special case of our results, we will also show that if we make the spectrum of the sensing matrix spiky GLM-EP can reach the information theoretic lower bounds in the phase retrieval problem. Aubin et al. [3] considered the phase retrieval problem with generative priors in the form of deep neural networks with random weight matrices, and showed that it yields smaller statistical-to-algorithmic gap than sparse priors.

Another venue of research that is also related to our work is the derivation of the information theoretic limits for analog compression schemes. Analog compression framework was first introduced in Wu and Verdú [58, 59] for compressed sensing. It was shown in Wu and Verdú [58, 59] that the minimum number of measurements required for successful signal reconstruction in an information theoretic framework is related to the Rényi information dimension of the signal distribution. Riegler and Tauböck [44] studied the phase retrieval problem using the analog compression framework and proved that (real-valued) phase retrieval has the same fundamental limit as that of compressed sensing. In order to compare the performance of GLM-EP on matrices with different spectrums we generalize the work of [58, 59] and [44] and obtain information theoretic limit for our sensing model. Note that

---

[1]The name GLM-EP is chosen because the model (1) is an instance of generalized linear models (GLM).

while we are using such information theoretic tools, the problem we are studying in this paper is fundamentally different from the one studied in [58, 59, 44]. Here we are interested in the impact of the spectrum of the sensing matrix on the performance of GLM-EP, and information theoretic limits are mainly derived for comparison purposes (and evaluating the optimality of GLM-EP).

## 1.3 Definitions

In this section, we mention some definitions that will be frequently used throughout this paper. We first start with the Rényi information dimension of a random variable.

**Definition 1** (Information dimension [43, 58]). *Let $X$ be a real-valued random variable, and $\langle X \rangle_M = \lfloor MX \rfloor / M$ be a quantization operator.[2] Suppose the following limit exists*

$$d(X) = \lim_{M \to \infty} \frac{H(\langle X \rangle_M)}{\log M},$$

*where $H(\cdot)$ is the entropy of a discrete random variable. The limit $d(X)$ is called the information dimension of $X$. Further, if $H(\lfloor X \rfloor) < \infty$, then $0 \le d(X) \le 1$.*

As will be discussed later, $d(X)$ plays a critical role in the information theoretic lower bounds we derive for the recovery algorithms. The next lemma shows how $d(X)$ can be calculated for the simple distributions we observe in our applications.

**Lemma 1** (Information dimension of mixed distribution [43, 58]). *Let $X$ be a random variable such that $H(\lfloor X \rfloor)$ is finite. Suppose the distribution of $X$ can be represented as*

$$P_X = (1 - \rho)P_d + \rho P_c,$$

*where $P_d$ is a discrete measure and $P_c$ is an absolutely continuous measure with respect to Lebesgue, and $0 \le \rho \le 1$. Then,*

$$d(X) = \rho.$$

The minimum mean squared error (MMSE) defined below is an important notion in our analysis of GLM-EP.

**Definition 2** (MMSE for AWGN channel [27]). *Let $(Z, U)$ be a pair of random variables. The MMSE $\mathsf{mmse}(Z, \mathsf{snr})$ and the conditional MMSE $\mathsf{mmse}(Z, \mathsf{snr}|U)$ given $U$ are defined as*

$$\mathsf{mmse}(Z, \mathsf{snr}) = \mathbb{E}\left[\left(Z - \mathbb{E}[Z|\sqrt{\mathsf{snr}}Z + N]\right)^2\right],$$
$$\mathsf{mmse}(Z, \mathsf{snr}|U) = \mathbb{E}\left[\left(Z - \mathbb{E}[Z|\sqrt{\mathsf{snr}}Z + N, U]\right)^2\right],$$

(2)

*where $N \sim \mathcal{N}(0, 1)$ is independent of $(Z, U)$, and the outer expectation is taken over all random variables involved.*

## 2 Information-theoretic limit for signal recovery

Our main objective is to evaluate the impact of the spectrum of the sensing matrices on the performance of GLM-EP. Before that, it is useful to compare what GLM-EP achieves with the information theoretic lower bounds, which we derive in this section.

## 2.1 Assumptions

Before we proceed to the technical part of the paper, let us review and discuss some of the assumptions we will be making throughout the paper.

(A.1) The elements of $\boldsymbol{x}$ are independently drawn from $P_X$, which is an absolutely continuous distribution with respect to the Lebesgue measure. We assume $\mathbb{E}[X^2] = 1$.

---

[2]The notation $\lfloor z \rfloor$ denotes the largest integer that is smaller than $z$.

(A.2) Let the SVD of $\boldsymbol{A}$ be $\boldsymbol{A} = \boldsymbol{U}\boldsymbol{\Sigma}\boldsymbol{V}^\top$. $\boldsymbol{U} \in \mathbb{R}^{m \times m}$ and $\boldsymbol{V} \in \mathbb{R}^{n \times n}$ are independent Haar matrices, which are further independent of $\boldsymbol{\Sigma}$. Let $\{\sigma_i\}_{i=1}^n$ be the diagonal entries of $\boldsymbol{\Sigma}$ and define $\lambda_i \triangleq \sigma_i^2$. We assume that the empirical distribution of $\{\lambda_i\}_{i=1}^n$ converges almost surely to a deterministic limit $P_\lambda$ with a compact support $[a, b]$ where $a > 0$. Further, we assume that $n^{-1} \sum_{i=1}^n \lambda_i \overset{a.s.}{\to} \mathbb{E}[\lambda] = m/n > 1$, where the expectation is with respect to $P_\lambda$.

(A.3) $f : \mathbb{R} \mapsto \mathcal{Y}$ is a piecewise smooth function. Specifically, the domain $\mathbb{R}$ can be decomposed into $K \in \mathbb{N}_+$ non-overlapping intervals, and on each sub-interval, $f$ is continuously differentiable and either strictly monotonic or constant. Furthermore, we assume $H(\lfloor f(Z) \rfloor) < \infty$ where $Z \sim \mathcal{N}(0, 1)$.

Note that Assumption (A.2) is a standard assumption in the theoretical analysis of GLM-EP [42, 51, 20]. Furthermore, all the nonlinearities that we observe in applications satisfy Assumption (A.3). We consider generic signal and do not impose any structural assumption. Finally, the independence assumption we have made in the prior of $\boldsymbol{x}$ is again standard in the literature of approximate message passing and expectation propagation [41, 22, 7, 57, 55]. One can weaken this assumption at the expense of making assumptions about the recovery algorithm.

## 2.2 Perfect reconstruction in a noiseless setting

In this section, we derive the information theoretic lower bound on the number of measurements required by Lipschitz recovery scheme to achieve vanishing error probability. Note that the computational complexity of the recovery algorithm is *not* of any concern in these lower bounds. We will later compare our results for GLM-EP with these information theoretic lower bounds.

**Theorem 1** (Perfect reconstruction under Lipschitz decoding). *Suppose Assumptions (A.1)-(A.3) hold. Suppose that there exists an $\boldsymbol{A} \in \mathbb{R}^{m \times n}$ and a Lipschitz continuous decoder $g : \mathcal{Y}^m \mapsto \mathbb{R}^n$ such that $\mathbb{P}\{\boldsymbol{x} \neq g(f(\boldsymbol{A}\boldsymbol{x}))\} \to 0$ as $m, n \to \infty$ and $m/n \to \delta \in (1, \infty)$, then necessarily we have*

$$\delta > \delta_{\mathrm{opt}}^p \triangleq \frac{1}{d(f(Z))}, \tag{3}$$

*where $d(f(Z))$ is the information dimension of $f(Z)$, where $Z \sim \mathcal{N}(0, 1)$. Here, the error probability is taken with respect to both $\boldsymbol{x}$ and $\boldsymbol{A}$.*

The proof of this result can be found in the Supplementary Material. Intuitively speaking, $d(f(Z)) \in [0, 1]$ may be interpreted as a measurement discount factor and the total number of effective measurements is $m \cdot d(f(Z))$.

**Remark 1** (1-bit CS). *For the 1-bit compressed sensing (CS) problem, we have $f(z) = \mathrm{sign}(z)$ and $d(f(Z)) = 0$. In this case, the condition $\delta > \delta_{\mathrm{opt}}^p = +\infty$ implies that perfect recovery is impossible in the regime $m, n \to \infty$ and $m/n \to \delta \in (1, \infty)$. Notice that our result does not contradict with existing 1-bit CS results [16, 40]. For instance, Dirksen et al. [16] analyzes the number of random measurements required by a convex minimization algorithm to achieve a non-zero target distortion $\rho$, and the bound blows up to infinity as $\rho \to 0$.*

## 2.3 Stable reconstruction in the noisy setting

Theorem 1 focuses on signal reconstruction for model (1) without any noise. For practical considerations, it is desirable to make sure that a small amount of measurement noise does not cause major performance degradation. In this paper, we consider the following noisy model[3]

$$\boldsymbol{y} = f(\boldsymbol{A}\boldsymbol{x} + \boldsymbol{w}), \tag{4}$$

where $\boldsymbol{w} \sim \mathcal{N}(\boldsymbol{0}, \sigma_w^2 \boldsymbol{I})$ is independent of $\boldsymbol{A}$ and $\boldsymbol{x}$. Define the *noise sensitivity* [18, 59] of the minimum mean square error (MMSE) estimator by

$$M^*(X, f, \delta) \triangleq \sup_{\sigma_w} \limsup_{n \to \infty} \frac{\frac{1}{n}\mathsf{mmse}(\boldsymbol{x}|\boldsymbol{y}, \boldsymbol{A})}{\sigma_w^2}, \tag{5}$$

---

[3]Other types of noisy models are possible, e.g., $\boldsymbol{y} = f(\boldsymbol{A}\boldsymbol{x}) + \boldsymbol{w}$. Extending our results to these models is beyond the scope of the current paper.

where $\mathrm{mmse}(\boldsymbol{x}|\boldsymbol{y}, \boldsymbol{A}) \triangleq \mathbb{E}\big[(\boldsymbol{x} - \mathbb{E}[\boldsymbol{x}|\boldsymbol{y}, \boldsymbol{A}])^2\big]$ is the MMSE of estimating $\boldsymbol{x}$ from $\boldsymbol{y}$, and in the above limit $n \to \infty$ is understood as $n \to \infty$ and $m/n \to \delta$. Theorem 2 below shows that to achieve bounded noise sensitivity, one needs $\delta > 1/d(f(Z))$, the same necessary condition for achieving vanishing error probability in the noiseless setting. Its proof can be found in the Supplementary Material.

**Theorem 2** (Noise sensitivity)**.** *Suppose Assumptions (A.1)-(A.3) hold. Additionally, assume* $\boldsymbol{x} \sim \mathcal{N}(\boldsymbol{0}, \boldsymbol{I})$*. For any* $P_\lambda$*, a necessary condition for achieving bounded noise sensitivity, namely* $M^*(X, f, \delta) < \infty$*, is* $\delta > \delta_{\mathrm{opt}}^p$*, where* $\delta_{\mathrm{opt}}^p$ *is defined in* (3)*.*

Note that Theorem 2 is different from Theorem 1 in that it holds for any eigenvalue value distribution $P_\lambda$ satisfying Assumption (A.2). Further, we may calculate the MMSE (and hence the noise sensitivity) using the replica method [34]. However, since the correctness of the replica predictions has not been proved for the current setting, we do not pursue it in this paper and leave it as possible future work.

## 2.4 Discussion of Theorems 1 and 2

Theorems 1 and 2 show that that the quantity $d(f(Z))$ determines the fundamental limit for signal recovery from the nonlinear model (1). Notice that $f(Z)$ is a mixed discrete-continuous distribution (by Assumption (A.3)), where the discrete component in $f(Z)$ corresponds to "flat" sections of $f$; see Figure 1 for illustration. According to Lemma 1, $d(f(Z))$ is simply the weight in the continuous component of the distribution of $f(Z)$, which is the probability of $Z \sim \mathcal{N}(0, 1)$ falling into the non-flat sections of $f$. For illustration, Figure 1 shows three representative examples of $f$.

**Type I:** $f$ is a piece-wise monotone function without flat sections; see the left panel of Figure 1 for illustration. This type of functions includes the absolute value function $f = |z|$, which appears in phase retrieval problems. For such functions, $f(Z)$ has an absolutely continuous distribution when $Z \sim \mathcal{N}(0, 1)$, and hence $d(f(Z)) = 1$ according to Lemma 1.

**Type II:** $f$ consists of purely flat sections. A special case is the quantization function. Clearly, $f(Z)$ has a discrete distribution and $d(f(Z)) = 0$.

**Type III:** $f$ consists of both flat and non-flat sections, e.g., the function shown on the right panel of Figure 1. Note that such scenarios happen when sensors saturate for instance in the phase retrieval application. In this case, $f(Z)$ has a mixed discrete-continuous distribution and $0 < d(f(Z)) < 1$.

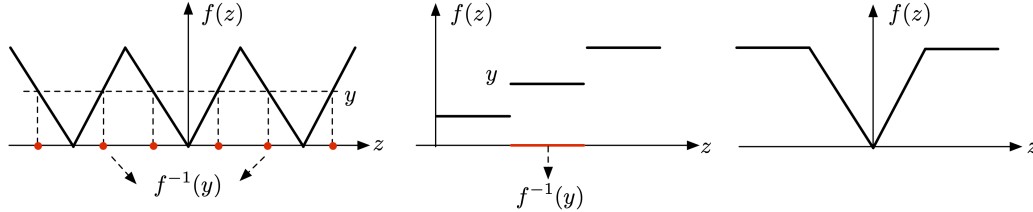

Figure 1: Three types of $f$. Left: $d(f(Z)) = 1$. Center: $d(f(Z)) = 0$. Right: $0 < d(f(Z)) < 1$.

## 3 GLM-EP algorithm and performance analysis

In this section, we introduce an expectation propagation (EP) [36, 38] type algorithm, referred to as GLM-EP, for solving our nonlinear inverse problem and derive its state evolution (SE). We then study the impact of the spectrum of the sensing matrix on the performance of this algorithm.

## 3.1 Summary of GLM-EP

The GLM-EP algorithm is summarized below. We use superscripts to represent iteration indices, and subscripts '$l$' and '$r$' to distinguish different variables.

**Initialization:** $\boldsymbol{z}_r^{-1} = \boldsymbol{0}$, $v_r^{-1} = 1$. For $t = 0, \ldots$, excute the following steps iteratively:

$$z_l^t = \frac{1}{1 - \langle \eta_z'(z_r^{t-1}, y, v_r^{t-1}) \rangle} \cdot \left( \eta_z(z_r^{t-1}, y, v_r^{t-1}) - \langle \eta_z'(z_r^{t-1}, y, v_r^{t-1}) \rangle \cdot z_r^{t-1} \right), \tag{6a}$$

$$v_l^t = v_r^{t-1} \cdot \frac{\langle \eta_z'(z_r^{t-1}, y, v_r^{t-1}) \rangle}{1 - \langle \eta_z'(z_r^{t-1}, y, v_r^{t-1}) \rangle}, \tag{6b}$$

$$\boldsymbol{R}^t \triangleq \boldsymbol{A} \left( v_l^t \boldsymbol{I} + \boldsymbol{A}^\mathsf{T} \boldsymbol{A} \right)^{-1} \boldsymbol{A}^\mathsf{T}, \tag{6c}$$

$$z_r^t = \frac{1}{1 - \frac{1}{m}\mathsf{Tr}(\boldsymbol{R}^t)} \cdot \left( \boldsymbol{R}^t - \frac{1}{m}\mathsf{Tr}(\boldsymbol{R}^t) \cdot \boldsymbol{I} \right) \cdot z_l^t, \tag{6d}$$

$$v_r^t = v_l^t \cdot \frac{\frac{1}{m}\mathsf{Tr}(\boldsymbol{R}^t)}{1 - \frac{1}{m}\mathsf{Tr}(\boldsymbol{R}^t)}, \tag{6e}$$

where $\eta_z$ is defined by

$$\eta_z(z_r, y, v) \triangleq \frac{\int_{f^{-1}(y)} u \cdot \mathcal{N}(u; z_r, v) du}{\int_{f^{-1}(y)} \mathcal{N}(u; z_r, v) du}, \tag{6f}$$

and $\eta_z'$ denotes the derivative of $\eta_z$ with respect to the first argument. When $f^{-1}(y)$ is a discrete set, the integration in the above formula is simply replaced by a summation.

**Output:** $\hat{x}_{\text{out}}^t = v_l^t(\boldsymbol{I} + v_l^t \boldsymbol{A}^\mathsf{T} \boldsymbol{A})^{-1} \boldsymbol{A}^\mathsf{T} z_l^t$.

In the above descriptions of the algorithm, we adopted the convention commonly used in the AMP literature: $\eta_z(z_r, y, v)$ denotes a vector with elements obtained by applying the scalar function $\eta_z$ to the corresponding elements of $z_r$ and $y$, and $\langle \cdot \rangle$ denotes the empirical mean of a vector.

### 3.2 Asymptotic analysis

The asymptotic performance of GLM-EP could be described by two scalar sequences $\{V_l^t, V_r^t\}_{t \geq 0}$, defined recursively by

$$V_l^t = \left( \frac{1}{\mathsf{mmse}_z\left(V_r^{t-1}\right)} - \frac{1}{V_r^{t-1}} \right)^{-1} \triangleq \phi(V_r^{t-1}), \tag{7a}$$

$$V_r^t = \left( \frac{1}{\frac{1}{\delta} \cdot \mathbb{E}\left[ \frac{V_l^t \lambda}{V_l^t + \lambda} \right]} - \frac{1}{V_l^t} \right)^{-1} \triangleq \Phi(V_l^t), \tag{7b}$$

where $V_r^{t-1}\big|_{t=0} = 1$, $\mathsf{mmse}_z(V_r) \triangleq \mathsf{mmse}\left(Z, V_r^{-1} - 1 | f(Z)\right)$, and the expectation in (7b) is w.r.t. the limiting eigenvalue distribution of $\boldsymbol{A}^\mathsf{T} \boldsymbol{A}$. (Recall that $\mathsf{mmse}(Z, \mathsf{snr}|U)$ denotes a conditional MMSE; see (2)). Equations (7a) and (7b) are known as the state evolution (SE) for the GLM-EP.

Roughly speaking, the deterministic sequences $\{V_l^t, V_r^t\}_{t \geq 0}$ are expected to be accurate predictions of $\{v_l^t, v_r^t\}_{t \geq 0}$ (which are generated by GLM-EP) asymptotically. We will formalize this claim below. Further, we will show that the per coordinate MSE of $\boldsymbol{x}_{\text{out}}^t$ (see Lemma 2 below) is characterized by

$$\mathsf{MSE}_\lambda(V_l^t) \triangleq \mathbb{E}\left[ \frac{V_l^t}{V_l^t + \lambda} \right]. \tag{7c}$$

The subscript emphasizes the fact that the MSE depends on the limiting eigenvalue distribution $P_\lambda$.

Lemma 2 below gives a formal statement of the accuracy of SE, and its proof is mainly based on that of [23, Theorem 1]. Note that [23] requires the continuity of $f$ and $\eta_z$. While this assumption holds for some problems, e.g. phase retrieval problem, it is violated for some other problems, e.g. one-bit quantization. Hence, we construct a new algorithm, called GLM-EP-app hereafter, which satisfies the requirements of [23]. This allows us to use SE for predicting the performance of this algorithm. Please note that if $f$ and $\eta_z$ are Lipschitz continuous, then we can use GLM-EP as well. GLM-EP-app uses the following iterations:

$$z_l^t = C_t \cdot \left( \tilde{\eta}_z(z_r^{t-1}, y, V_r^{t-1}) - \mathbb{E}\left[ \tilde{\eta}_z'(Z^{t-1}, Y, V_r^{t-1}) \right] \cdot z_r^{t-1} \right), \tag{8a}$$

$$z_r^t = \frac{1}{1 - \frac{1}{m}\mathsf{Tr}(\boldsymbol{R}^t)} \cdot \left( \boldsymbol{R}^t - \frac{1}{m}\mathsf{Tr}(\boldsymbol{R}^t) \cdot \boldsymbol{I} \right) \cdot z_l^t, \tag{8b}$$

where $\tilde{\eta}$ is a function for which $\mathbb{E}\left[\tilde{\eta}'_z(Z_r^{t-1}, Y, V_r^{t-1})\right]$ exists, $\boldsymbol{R}^t \triangleq \boldsymbol{A}\left(V_l^t\boldsymbol{I} + \boldsymbol{A}^\mathsf{T}\boldsymbol{A}\right)^{-1}\boldsymbol{A}^\mathsf{T}$, and $C_t$ is a sequence of fixed numbers. The choices we choose for $\tilde{\eta}$ and $C_t$ (to make them close enough to GLM-EP) is discussed in the proof of Lemma 2. Finally, similar to GLM-EP the output of GLM-EP-app is given by

$$\hat{\boldsymbol{x}}_{\text{out}}^t = V_l^t(\boldsymbol{I} + V_l^t\boldsymbol{A}^\mathsf{T}\boldsymbol{A})^{-1}\boldsymbol{A}^\mathsf{T}\boldsymbol{z}_l^t.$$

Lemma 2 shows that the performance of GLM-EP-app could be arbitrarily close to the SE prediction. The details of the proof can be found in Supplementary Material.

**Lemma 2.** *Suppose Assumptions (A.1)-(A.3) hold. Additionally, assume $f : \mathbb{R} \mapsto \mathcal{Y}$ to be Lipschitz continuous. Let $\{V_l^t, V_r^t\}_{t \geq 0}$ be generated according to (7). There exists $\tilde{\eta}_z$ and $\{C_t\}_{t \geq 0}$ such that $\hat{\boldsymbol{x}}_{\text{out}}^t$ of* GLM-EP-app *satisfies*

$$\mathsf{MSE}_\lambda(V_l^t) - \epsilon \leq \frac{1}{m}\left\|\hat{\boldsymbol{x}}_{\text{out}}^t - \boldsymbol{x}\right\|^2 < \mathsf{MSE}_\lambda(V_l^t) + \epsilon, \quad \forall \epsilon > 0, \tag{9}$$

*almost surely as $m, n \to \infty$ with $m/n \to \delta \in (1, \infty)$, where $\mathsf{MSE}_\lambda$ is defined in (7c).*

According to Lemma 2, the asymptotic MSE of GLM-EP-app in the large system limit as $t \to \infty$ can be obtained from the limiting value of $V_r^t$ (or $V_l^t$). Because this quantity is of particular importance to us we will characterize it in the following lemma.

**Lemma 3** (MSE performance). *Suppose $\delta > 1$. Define $V_r^\star$ by*

$$V_r^\star \triangleq \inf\left\{v \in [0, 1] \; : \; P(v_r) > 0, \forall v_r \in [v, 1]\right\}. \tag{10}$$

*where*

$$P(v_r) \triangleq \mathbb{E}\left[\frac{\phi(v_r)}{\phi(v_r) + \lambda}\right] - \underbrace{\left[1 - \delta\left(1 - \frac{\mathsf{mmse}_z(v_r)}{v_r}\right)\right]}_{g(v_r)}. \tag{11}$$

*Let $\{V_l^t, V_r^t\}_{t \geq 0}$ be sequences generated according to (7) with $V_r^{-1} = 1$. We have*

$$\lim_{t \to \infty} V_r^t = V_r^\star.$$

*Further, the final MSE is given by $\mathsf{MSE}_\lambda^\star \triangleq \mathsf{MSE}_\lambda(\phi(V_r^\star))$, where $\phi$ is defined in (7a).*

The proof of this lemma can be found in Supplementary Material. A direct consequence of Lemma 3 is the perfect reconstruction condition stated in Lemma 4 below.

**Lemma 4** (Perfect reconstruction condition). *Let $\{V_l^t, V_r^t\}_{t \geq 0}$ be a sequence generated through (7) with $V_r^{t-1}\big|_{t=0} = 1$. Then, the predicted final MSE $\mathsf{MSE}_\lambda^\star$ is zero if and only if*

$$P(v_r) > 0, \quad \forall v_r \in (0, 1], \tag{12}$$

*where $P(v_r)$ is defined in (11). Furthermore, a necessary condition for (12) is $\delta > \delta_{\text{opt}}^p$ where $\delta_{\text{opt}}^p$ is defined in (3).*

The proofs can be found in the Supplementary Material.

## 4 Impact of sensing matrix spectrum

In this section, we use Lemmas 3 and 4 to study the impact of the sensing matrix on the MSE performance of GLM-EP-app. Before presenting our detailed analysis, we first discuss the so-called Lorenz order that compares the "spikiness" of different distributions.

### 4.1 A measure of spikiness of distributions

A natural tool to compare the spikiness of the distributions of two non-negative random variables is Lorenz partial order [1]. (Since it is a partial order, there exist distributions that are incomparable in the Lorenz sense.) Lorenz order is well-known in economics to characterize the wealth inequality of different populations. Lorenz order is closely related to majorization, a tool that has been extensively studied for transceiver design in communication systems [39].

**Definition 3** (Lorenz partial order [1]). *Consider a nonnegative random variable with cumulative density function $F(x)$. Let $F^{-1}(y)$ be the quantile function defined by*

$$F^{-1}(y) = \sup\{x : F(x) \le y\}, \quad 0 < y < 1. \tag{13}$$

*The Lorenz curve corresponding to $F(x)$ is defined by*

$$L(u) = \frac{\int_0^u F^{-1}(y)dy}{\int_0^1 F^{-1}(y)dy}, 0 \le u \le 1.$$

*Let $X$ and $Y$ be two nonnegative random variables, and $L_X(u)$ and $L_Y(u)$ be the corresponding Lorenz curves. We say that $X$ is less spiky than $Y$ in the Lorenz sense, denoted as $X \preceq_L Y$, if $L_X(u) \ge L_Y(u)$ for every $u \in [0,1]$. Conversely, $X \succeq_L Y$ if $L_X(u) \le L_Y(u)$ for every $u \in [0,1]$.*

An important property of Lorenz partial ordering is the following.

**Lemma 5** ([1]). *Suppose $X \ge 0$, $Y \ge 0$ and $\mathbb{E}[X] = \mathbb{E}[Y]$. We have $X \preceq_L Y$ if and only if $\mathbb{E}[h(X)] \le \mathbb{E}[h(Y)]$ for every continuous convex function $h : \mathbb{R}_+ \to \mathbb{R}$.*

### 4.2 Impact on MSE

Let $\lambda_1 \sim P_{\lambda_1}$ and $\lambda_2 \sim P_{\lambda_2}$ be two limiting eigenvalue distributions of $\boldsymbol{A}^\mathsf{T}\boldsymbol{A}$. Let $V_{\lambda_1}^\star$ and $V_{\lambda_2}^\star$ denote the corresponding limiting values of $V_r^t$ (as $t \to \infty$) in (7) (proving that the iterations (7) converge to a fixed point is straightforward). The associated MSEs, denoted as $\mathsf{MSE}_{\lambda_1}^\star$ and $\mathsf{MSE}_{\lambda_2}^\star$, can be compared according to the following lemma.

**Lemma 6.** *Let $\delta > 1$. Suppose $P_{\lambda_1}$ is more spiky than $P_{\lambda_2}$ in the Lorenz sense, i.e., $\lambda_1 \succeq_L \lambda_2$. Define*

$$G(v_r) \triangleq \max\big(g(v_r), 0\big), \quad \forall v_r \in [0,1], \tag{14}$$

*where $g(\cdot)$ is defined in (11). We have*

- *If $G(v_r)$ is increasing on $v_r \in [0,1]$, then $\mathsf{MSE}_{\lambda_1}^\star \le \mathsf{MSE}_{\lambda_2}^\star$;*

- *If $G(v_r)$ is decreasing on $v_r \in [0,1]$, then $\mathsf{MSE}_{\lambda_1}^\star \ge \mathsf{MSE}_{\lambda_2}^\star$;*

- *If $G(v_r)$ is not monotonic, then the comparison of $\mathsf{MSE}_{\lambda_1}^\star$ and $\mathsf{MSE}_{\lambda_2}^\star$ is not definite.*

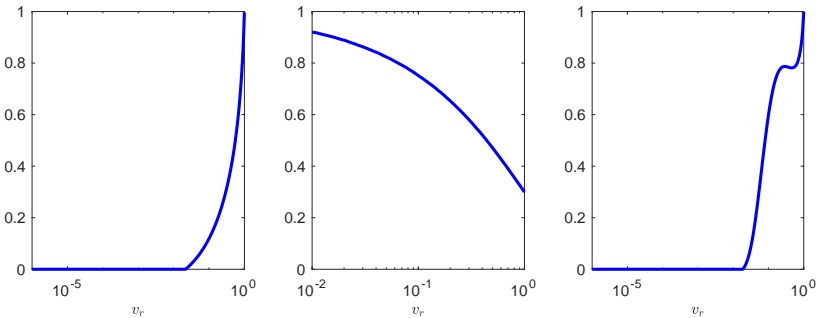

Figure 2: Illustration of $G(v_r)$ for three choices of $f$. **Left:** $f(z) = |z|$. **Middle:** $f(z) = \text{sign}(z)$. **Right:** $f(z) = |z|\mathbf{1}_{|z|<1} + (|z|-1)\mathbf{1}_{|z|\ge 1}$. $\delta = 1.1$.

**Remark 2.** *Notice that the function $G(v_r)$ also depends on the sampling ratio $\delta$, as can be seen from the definitions in (14) and (11). (To keep notation light, we do not make such dependency explicit.) Hence, for a given $f$, the monotonicity of $G(v_r)$ could change as $\delta$ varies.*

Lemma 6 shows that the impact of the spectrum on the final MSE performance of GLM-EP-app depends on the monotonicity of the function $G(v_r)$ (which further depends on $f$) and the sampling ratio $\delta$. For a given $f$ and $\delta$, the function $G(v_r)$ can be numerically computed and its monotonicity

can be easily checked. Below are three examples of $f$, corresponding to each of the cases discussed in Lemma 6.

**Example 1:** It can be shown that that $G(v_r)$ of the following $f$ is monotonically increasing for all $\delta > 1$:
$$f(z) = |z|.$$
See left panel of Figure 2. Hence, spiky spectrums are beneficial for MSE performance.

**Example 2:** The $G(v_r)$ of the following function is decreasing for all $\delta > 1$:
$$f(z) = \text{sign}(z).$$
See the middle panel of Figure 2. In this case, flatter spectrums are better.

**Example 3:** Consider the following function
$$f(z) = \begin{cases} |z|, & \text{if } |z| < 1 \\ |z| - 1, & \text{if } |z| \geq 1. \end{cases} \tag{15}$$
In this case, $G(v_r)$ is not monotonic (See right panel of Figure 2), and other features of the spectrum affect the performance of GLM-EP-app.

### 4.3 Impact of spectrum on perfect recovery threshold

We have shown that the impact of the spikiness of the spectrum on the MSE performance is related to the monotonicity of the function $G(v_r)$ which depends on the nonlinear function $f$ and the sampling ratio $\delta$. In this section, we will show that if our goal is to *minimize the number of measurements required for perfect reconstruction*, then more spiky spectrum benefit GLM-EP-app *for all $f$*. Furthermore, the information theoretic lower bound $\delta_{\text{opt}}^{\text{P}}$ can be reached (as close as we wish) if the spectrum of $A$ is spiky enough. Theorem 3, whose proof can be found in the Supplementary Material, summarizes the above discussions.

**Theorem 3.** *For a given nonlinearity $f$ and eigenvalue distribution $P_\lambda$, let $\delta_\lambda^{\text{P}}$ be the minimum $\delta$ required for perfectly recovering the signal, i.e.,*
$$\delta_\lambda^{\text{P}} \triangleq \inf\left\{\delta \; : \; \text{MSE}_\lambda^\star = 0\right\}, \tag{16}$$
*where $\text{MSE}_\lambda^\star$ is defined in Lemma 3. Let $\lambda_1$ and $\lambda_2$ represent two limiting eigenvalue distributions and $\delta_{\lambda_1}^{\text{P}}$ and $\delta_{\lambda_2}^{\text{P}}$ the corresponding thresholds for perfect reconstruction. We have*

*(i) If $\lambda_1 \succeq_L \lambda_2$, then $\delta_{\lambda_1}^{\text{P}} \leq \delta_{\lambda_2}^{\text{P}}$;*

*(ii) For any $f$ satisfying $\text{mmse}_z(1) < 1$ and $\delta_{\text{opt}}^{\text{P}} < \infty$, there exists a distribution of $\lambda$ for which $\delta_\lambda^{\text{P}}$ is arbitrarily close to $\delta_{\text{opt}}^{\text{P}} \triangleq 1/d(f(Z))$, where $Z \sim \mathcal{N}(0, 1)$.*

### 4.4 Noise Sensitivity Analysis

Up to now, we only studied the performance of GLM-EP-app in the noiseless setting. In practice, it is also important to guarantee that the reconstruction performance does not significantly worsen due to the presence of a small amount of measurement noise. We consider the noisy model in (4). GLM-EP-app remains unchanged except that $\eta_z$ is replaced by a posterior mean estimator that takes the noise effect into consideration.

The following lemma analyzes the MSE performance of GLM-EP-app in the high SNR regime, and shows that its reconstruction is stable. The proof of Lemma 7 and other details about GLM-EP-app in the noisy setting are provided in the Supplementary Material.

**Lemma 7.** *Assume $d(f(Z)) \neq 0$. Let $\delta > \delta_\lambda^{\text{P}}$, where $\delta_\lambda^{\text{P}}$ is defined in Theorem 3. Let $\text{MSE}_\lambda^\star(\sigma_w^2) \triangleq \lim_{t\to\infty} \text{MSE}_\lambda(V_l^t)$ be the MSE in the noisy setting. As $\sigma_w^2 \to 0$, we have*
$$\text{MSE}_\lambda^\star(\sigma_w^2) = C(\delta, f)\mathbb{E}\left[\lambda^{-1}\right]\sigma_w^2 \cdot (1 + o(1)),$$
*where $0 < C(\delta, f) < \infty$ is a constant depending only on $\delta$ and $f$.*

This lemma confirms that as long as $\delta > \delta_\lambda^{\text{P}}$, GLM-EP-app can offer stable recovery. However, the minimum mean square error in this case depends on another feature of the spectrum, namely $\mathbb{E}\left[\lambda^{-1}\right]$. The optimal sensing mechanism should be designed by considering both features based on the expected noise level in the system.

# 5 Numerical examples

Let $A = U\Sigma V^\top$. In our experiments, we approximate the random orthogonal matrix $U$ in the following way: $U = P_1 U_{\mathrm{d}} P_2 U_{\mathrm{d}}^\top P_3$ where $P_1, P_2, P_3$ are three diagonal matrices with entries independently chosen from $\pm 1$ with equal probability, and $U_{\mathrm{d}}$ is a discrete cosine transform (DCT) matrix. Note that all matrices are square. The hope is that by injecting enough randomness in these matrices, we can make them look like Haar orthogonal matrices for GLM-EP. In addition, such constructions allow fast implementation of GLM-EP using the DCT. Following [53], we consider a geometric distribution for the limiting empirical distribution of $\mathrm{diag}(\Sigma^\top \Sigma)$:

$$P_\lambda(\alpha, \beta) = \begin{cases} \frac{1}{\beta\lambda}, & \text{if } \lambda \in \left(\alpha A(\beta) e^{-\beta}, \alpha A(\beta)\right], \\ 0, & \text{otherwise,} \end{cases} \tag{17}$$

where $\alpha > 0$ is the mean, $\beta \geq 0$ controls the spikeness of the distribution (with $\beta = 0$ corresponding to a flat spectrum), and $A(\beta) = \frac{\beta}{1-e^{-\beta}}$. In all of our numerical experiments, the eigenvalues of $A^\top A$ are independently drawn from $P_\lambda(\alpha, \beta)$.

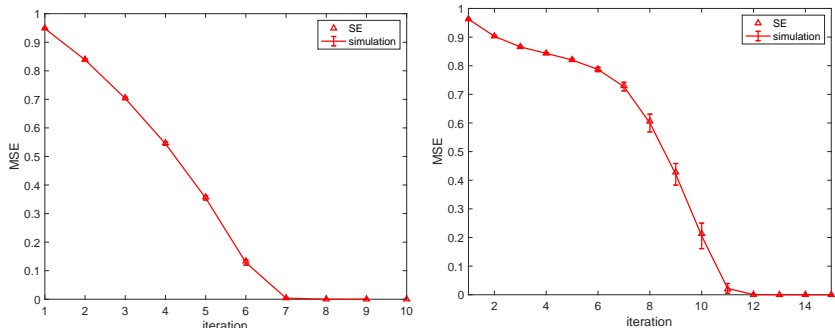

Figure 3: Noiseless reconstruction performance of GLM-EP. **Left:** $f(z) = |z|$. **Right:** $f(z)$ defined in (15). $n = 2 \times 10^5$. $m = \lceil 1.01 \cdot n \rceil$. $\beta = 20$. 1000 independent runs. The markers labeled 'SE' are predictions obtained from state evolution.

Figure 3 demonstrates the performances of GLM-EP for $f(z) = |z|$ and the function defined in (15). Clearly, $d(f(Z)) = 1$ for both functions. As Theorem 3, shows, the GLM-EP algorithm could achieve perfect reconstruction as soon as $\delta > 1$ with a very spiky sensing matrix. Here, we considered the geometric eigenvalue setup with $\beta = 20$. From Fig. 3, we see that GLM-EP recovers the signal accurately when $\delta$ is only slightly larger than the lower bound ($\delta = 1.01$). In our experiments, to get rid of the uninformative fixed point we set $z_r^{-1} = (1 + V)^{-1}(z + \sqrt{V}n)$ where $n$ is standard Gaussian and $V$ is a large constant (here we set $V = 20$).

More experimental results can be found in the Supplementary Material.

# 6 Future work

The results in this paper can serve as the first step toward the optimal design of sensing matrices for the nonlinear model $y = f(Ax)$. However, there are several directions that require further investigation before one can apply our results to real-world applications: (i) Because the structure of the signal is often used in recovery algorithms, the role of the structure should be studied more carefully when we deal with spiky sensing matrices. (ii) While we discussed the high-signal-to-noise ratio regime in the paper, some applications have low signal-to-noise ratios. The impact of the spectrum of the sensing matrix in such cases requires more careful considerations.

## Acknowledgments and Disclosure of Funding

J. Ma was partially supported by National Natural Science Foundation of China (Grant NO. 12101592) and the Key Research Program of the Chinese Academy of Sciences (Grant NO. XDPB15). A. Maleki was partially supported by a Google Faculty Research Award.

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
