## 5.2 Performance for medium-sized systems

Fig. 4 shows the performance of GLM-EP for very medium-sized sensing matrices ($n = 2000$). Other settings are the same as Fig. 3. In this case, we can observe a mismatch between the performance of GLM-EP and its theoretical predictions. Nevertheless, GLM-EP still achieve very good reconstruction result considering the fact that $\delta \approx 1.01$ is very close to the information theoretical lower bound.

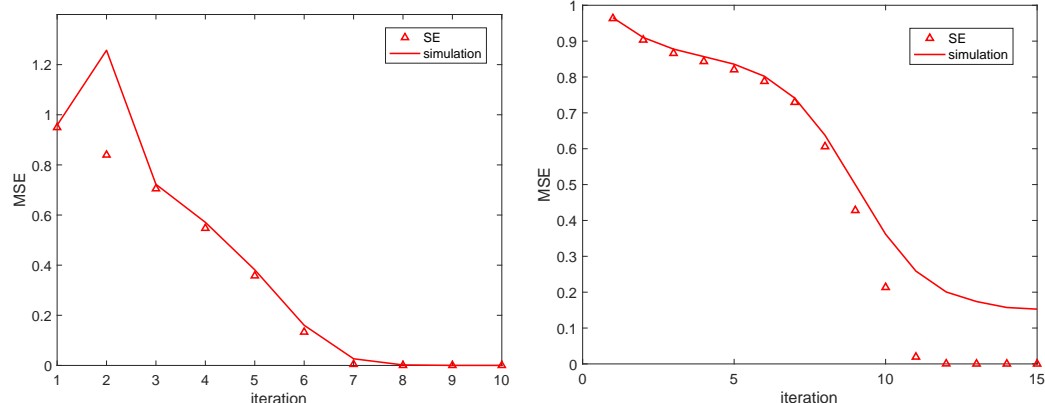

Figure 4: MSE performance of GLM-EP for medium-size systems. **Left:** $f(z) = |z|$. **Right:** $f(z)$ defined in (15). $n = 2000$. MSE are averaged over 1000 independent runs. Other settings are the same as those of Fig. 3.

### 5.3  1-bit CS performance

For the 1-bit compressed sensing (CS) problem, it is impossible to recover the signal accurately (namely, achieve zero MSE) at finite $\delta$. Tab. 1 lists the MSE of GLM-EP for 1-bit CS under different values of $\delta$. As expected, its performance improves as $\delta$ increases.

| $\delta$ | 1.5 | 2 | 2.5 | 3 | 3.5 | 4 | 4.5 | 5 |
|---|---|---|---|---|---|---|---|---|
| MSE | 0.2771 | 0.2091 | 0.1622 | 0.1286 | 0.1042 | 0.0846 | 0.0714 | 0.0607 |

Table 1: MSE of GLM-EP for the 1-bit CS problem. $n = 10^5$. $\beta = 0$ (i.e., $\boldsymbol{A}$ is column-orthogonal). The MSE is averaged over 100 independent runs. The number of iterations is 20.

### 5.4  Noisy measurements

Lemma 7 analyzes the stability of the GLM-EP reconstruction for the noisy model $\boldsymbol{y} = f(\boldsymbol{Ax} + \boldsymbol{w})$. Tab. 2 shows that the performance of GLM-EP for noisy phase retrieval. Here, the signal-to-noise ratio (SNR) is defined by

$$\mathrm{SNR} \triangleq \frac{\mathbb{E}[\|\boldsymbol{Ax}\|^2]}{\mathbb{E}[\|\boldsymbol{w}\|^2]}.$$

Results in Tab. 2 suggests that the performance of GLM-EP degrades gracefully as the noise variance increases.

| SNR | 30dB | 35dB | 40dB | 45dB | 50dB |
|---|---|---|---|---|---|
| MSE | 1.28e-01 | 5.92e-02 | 2.18e-02 | 6.94e-03 | 2.14e-03 |

Table 2: MSE of GLM-EP for noisy phase retrieval. $\delta = 1.1$. $n = 10^5$. $\beta = 10$. The MSE is averaged over 100 independent runs. The number of iterations is 10.

### 5.5  Phase transition

To test the impact of the sensing spectral on the performance of GLM-EP, we carry out phase transition study of GLM-EP in Fig. 5 under various values of $\beta$. We consider two instances of $f$, the absolute value function and that defined in (15). We see that for both functions, the empirical perfect recovery threshold of $\delta$ improves as $\beta$ increases (corresponding to spikier spectrum), which is consistent with the claim of Theorem 3.

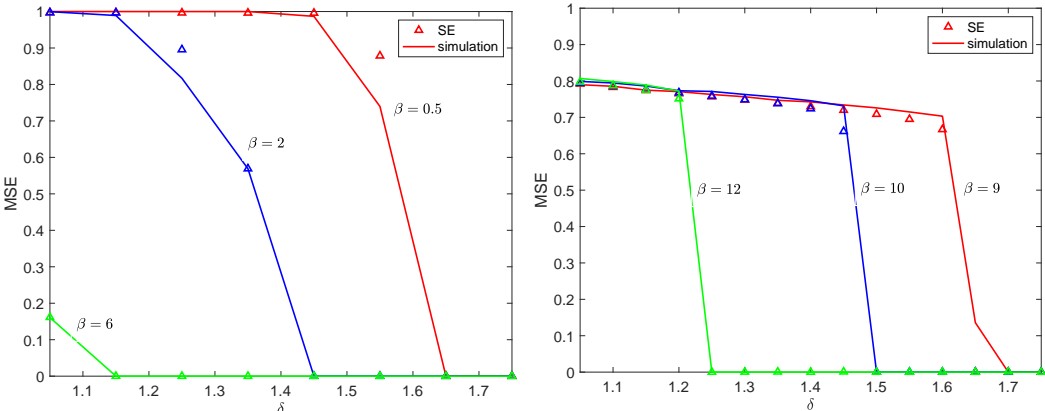

Figure 5: Phase transition of GLM-EP under various sensing matrix spectral. **Left:** $f(z) = |z|$. **Right:** $f(z)$ defined in (15). $n = 2 \times 10^5$. MSE are averaged over 100 independent runs.

## 6  Conclusion and future work

We studied the impact of the spectrum of the sensing matrix on the performance of the expectation propagation (EP) algorithm in recovering signals from the nonlinear model $y = f(Ax)$. We defined a notion of spikiness of the distributions and showed that depending on $f(\cdot)$, the spikiness of the distribution can help or hurt the performance of EP. We also showed that spiky sensing matrices can always reduce the number of observations required for the exact recovery of $x$ from $y$.

The results in this paper can serve as the first step toward the optimal design of sensing matrices. However, there are several directions that require further investigation before one can apply our results to real-world applications: (i) Because the structure of the signal is often used in recovery algorithms, the role of the structure should be studied more carefully when we deal with spiky sensing matrices. (ii) While we discussed the high-signal-to-noise ratio regime in the paper, some applications have low signal-to-noise ratios. The impact of the spectrum of the sensing matrix in such cases requires more careful considerations.

## Acknowledgments and Disclosure of Funding

J. Ma was partially supported by National Natural Science Foundation of China (Grant NO. 12101592) and the Key Research Program of the Chinese Academy of Sciences (Grant NO. XDPB15). A. Maleki was partially supported by a Google Faculty Research Award.

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

## A  Proof of Theorem 1

We first recall a few definitions and useful lemmas from [63, 65] in Section A.1. Then, we introduce our main technical lemma in Section A.2, and finally prove Theorem 1 in Section A.3.

### A.1 Minkowski dimension

In the almost lossless analog signal compression framework developed in [63, 65], the description complexity of bounded sets is gauged via their Minkowski dimension. Minkowski dimension is also called box-counting dimension [43] (hence the subscript $B$ in the notation $\overline{\dim}_B$).

**Definition 4** (Minkowski Dimension). *Let $\mathcal{S}$ be a nonempty bounded subset of a metric space. The upper Minkowski dimension of $\mathcal{S}$ is defined as*

$$\overline{\dim}_B(\mathcal{S}) = \limsup_{\epsilon \to 0} \frac{\log N_S(\epsilon)}{\log \frac{1}{\epsilon}}, \tag{18}$$

*where $N_S(\epsilon)$ is the $\epsilon$-covering number of $\mathcal{S}$, that is*

$$N_S(\epsilon) \triangleq \min \left\{ k : \mathcal{S} \subset \bigcup_{i=1}^{k} B(x_i, \epsilon), X_i \in \mathcal{S} \right\},$$

*where $B(x_i, \epsilon)$ denotes a ball centered at $x_i$ with radius $\epsilon$.*

For a probability measure, we define its $\epsilon$-Minkowski dimension [65] as the smallest Minkowski dimension among all sets with measure at least $1 - \epsilon$.

**Definition 5** ($\epsilon$-Minkowski Dimension). *Let $\mu$ be a probability measure on $\mathbb{R}^n$. Define the $\epsilon$-Minkowski dimension of $\mu$ as*

$$\overline{\dim}_B^{\epsilon}(\mu) = \inf\{\overline{\dim}_B(\mathcal{S}) : \mu(\mathcal{S}) \geq 1 - \epsilon\}. \tag{19}$$

An asymptotic version of the $\epsilon$-Minkowski dimension (called the Minkowski dimension compression rate) was introduced in [63]. Wu and Verdú [63] proved that the probability measure of an i.i.d. source concentrates on sets with Minkowski dimension approximately equal to the Rényi information dimension of the measure.

We will use the following lemma from [63] in the proof of the auxiliary lemma in Section A.2.

**Lemma 8** (Minkowski dimension in Euclidean spaces). *Let $\mathcal{S}$ be a bounded subset in $(\mathbb{R}^n, \|\cdot\|_2)$. The Minkowski dimension satisfies*

$$\overline{\dim}_B(\mathcal{S}) = \limsup_{q \to \infty} \frac{\log \left| \langle \mathcal{S} \rangle_{2^q} \right|}{q} \tag{20}$$

*where $\langle x \rangle_p \triangleq \lfloor px \rfloor / p$, and $\langle \mathcal{S} \rangle_p \triangleq \{\langle x \rangle_p : x \in \mathcal{S}\}$, and the logarithm uses base 2.*

Lemma 8 shows that in Euclidean spaces, we could replace $\epsilon$-balls by mesh cubes in defining covering number for Minkowski dimension. The similar forms of (20) and Definition 1 suggest the close relationship between Minkowski dimension and information dimension. Roughly speaking, Minkowski dimension counts the number of small pieces needed to cover the set while the information dimension also takes into account the probability of each piece and replaces the $\log N_S(\epsilon)$ term in (18) by an entropy term.

### A.2 An Auxiliary Lemma

We introduce a few definitions. First, recall the definition

$$\mathcal{Q}_f \triangleq \{y : f^{-1}(y) \text{ contains an interval}\},$$

where $f^{-1}(y) \triangleq \{z : f(z) = y\}$. We assumed $\mathcal{Q}_f$ to be a finite set. For $\boldsymbol{y} \in \mathbb{R}^m$, let

$$\text{Spt}(\boldsymbol{y}) \triangleq \{i = 1, \ldots, m : y_i \in \mathbb{R} \backslash \mathcal{Q}_f\} \tag{21}$$

be a kind of generalized support of $\boldsymbol{y}$ [63] (i.e., locations of the components of $\boldsymbol{y}$ that do not fall into the "flat" sections of $f$).

For convenience, we introduce the following definitions:

$$\mathcal{A}_\alpha \triangleq \left\{ s \in \mathbb{R}^n : y = f(A(s)), \frac{|\text{Spt}(y)|}{m} \leq \alpha \right\} \quad \text{and} \quad \mathcal{B}_\alpha \triangleq \left\{ y \in \mathbb{R}^m : \frac{|\text{Spt}(y)|}{m} \leq \alpha \right\},$$

$$\mathcal{A}_r \triangleq \left\{ s \in \mathbb{R}^n : y = f(A(s)), \|y\| \leq r \right\} \quad \text{and} \quad \mathcal{B}_r \triangleq \left\{ y \in \mathbb{R}^m : \|y\| \leq r \right\}. \tag{22}$$

Further, let $\mathcal{A}$ and $\mathcal{B}$ be the set of signals and measurements that can be perfectly reconstructed under decoder $g$. More specifically,

$$\mathcal{A} \triangleq \{ s \in \mathbb{R}^n : g(f(As)) = s \} \quad \text{and} \quad \mathcal{B} \triangleq \{ f(As) : s \in \mathcal{A} \}.$$

Clearly, the composite function $f \circ A$ is invertible (with $g$ being the inverse function) if we restrict its domain and co-domain to $\mathcal{A}$ and $\mathcal{B}$ respectively. With the above definitions, we have

$$\mathcal{B} \cap \mathcal{B}_\alpha \cap \mathcal{B}_r = \left\{ y : y \in \mathcal{B}, \frac{|\text{Spt}(y)|}{m} \leq \alpha, \|y\| \leq r \right\},$$

and

$$
\begin{aligned}
g\left( \mathcal{B} \cap \mathcal{B}_\alpha \cap \mathcal{B}_r \right) &= \left\{ g(y) : y \in \mathcal{B}, \frac{|\text{Spt}(y)|}{m} \leq \alpha, \|y\| \leq r \right\} \\
&= \left\{ s : s \in \mathcal{A}, \frac{|\text{Spt}(g^{-1}(s))|}{m} \leq \alpha, \|g^{-1}(s)\| \leq r \right\} \\
&= \left\{ s : s \in \mathcal{A}, \frac{|\text{Spt}(f(As))|}{m} \leq \alpha, \|f(As)\| \leq r \right\} \\
&= \mathcal{A} \cap \mathcal{A}_\alpha \cap \mathcal{A}_r.
\end{aligned}
$$

Lemma 9, which is a variation of [65, Theorem 5], is key to our proof of Theorem 1. Notice that Lemma 9 is a non-asymptotic result. Also, the radius $r$ of the boundedness constraint does not appear in (24).

**Lemma 9.** *Let $P_X$ be an arbitrary absolutely continuous distribution with respect to the Lebesgue measure and $x \sim \prod_{i=1}^n P_X(x_i)$ a random vector. Suppose that for some $\alpha \in (0, 1]$, $r > 0$ and $\epsilon \in (0, 1)$, there exists a matrix $A \in \mathbb{R}^{m \times n}$ and a Lipschitz continuous decoder $g : \mathcal{Y}^m \mapsto \mathbb{R}^n$ such that*

$$\mathbb{P}\{x \in \mathcal{A} \cap \mathcal{A}_\alpha \cap \mathcal{A}_r\} \geq 1 - \epsilon. \tag{23}$$

*Then, necessarily we have*

$$\frac{m}{n} \geq \frac{1 - \epsilon}{\alpha}. \tag{24}$$

*Proof.* Our proof follows from the following chain of inequalities:

$$
\begin{aligned}
\alpha \cdot m &\overset{(a)}{\geq} \overline{\dim}_B(\mathcal{B}_\alpha \cap \mathcal{B}_r) \\
&\geq \overline{\dim}_B(\mathcal{B}_\alpha \cap \mathcal{B}_r \cap \mathcal{B}) \\
&\overset{(b)}{\geq} \overline{\dim}_B(g(\mathcal{B}_\alpha \cap \mathcal{B}_r \cap \mathcal{B})) \\
&= \overline{\dim}_B(\mathcal{A}_\alpha \cap \mathcal{A}_r \cap \mathcal{A}) \\
&\overset{(c)}{\geq} \overline{\dim}_B^\epsilon(P_x) \\
&\overset{(d)}{\geq} \bar{d}(x) - \epsilon n \\
&\overset{(e)}{=} (1 - \epsilon)n
\end{aligned}
$$

where step (b) is from the fact that Minkowski dimension does not increase under Lipschitz mapping [20, Proposition 2.5], and step (c) is from the definition of $\epsilon$-Minkowski dimension (see Definition 5) and $\mathbb{P}\{x \in g(\mathcal{B} \cap \mathcal{B}_{\alpha,r})\} \geq 1 - \epsilon$, step (d) is proved in [65, Theorem 5], and step (e) is from the fact that $\bar{d}(x) = n \cdot d(X) = n$ when $x \sim \prod_i^n P_X(x_i)$ and $P_X$ is absolutely continuous.

It remains to prove step (a). Now, we use Lemma 8:

$$\overline{\dim}_B(\mathcal{B}_\alpha \cap \mathcal{B}_r) = \limsup_{M\to\infty} \frac{\log\left|\langle\mathcal{B}_\alpha \cap \mathcal{B}_r\rangle_M\right|}{\log M},$$

where $\langle\mathcal{B}_\alpha\cap\mathcal{B}_r\rangle_M$ is a set obtained by applying the discretization operator $\langle y\rangle_M = \lfloor My\rfloor/M$ (which has $2rM$ quantization levels in $y \in [-r, r]$) component-wisely to all of the elements in $\mathcal{B}_\alpha \cap \mathcal{B}_r$. From our definition of $\mathcal{B}_\alpha \cap \mathcal{B}_r$, we have

$$\begin{aligned}
\left|\langle\mathcal{B}_\alpha \cap \mathcal{B}_r\rangle_M\right| &= \left|\left\{\langle\boldsymbol{y}\rangle_M : \boldsymbol{y} \in \mathbb{R}^m,\ \frac{\mathrm{Spt}(\boldsymbol{y})}{m} \le \alpha,\ \|\boldsymbol{y}\| \le r\right\}\right|\\
&\le \left|\left\{\langle\boldsymbol{y}\rangle_M : \frac{\mathrm{Spt}(\boldsymbol{y})}{m} \le \alpha,\ \boldsymbol{y} \in [-r, r]^m\right\}\right|\\
&\le \sum_{i=0}^{\alpha m} C_m^i (2rM)^i |\mathcal{Q}_f|^{m-i}\\
&\le \sum_{i=0}^{\alpha m} C_m^i (2rM)^{\alpha m} |\mathcal{Q}_f|^{(1-\alpha)m} \quad \text{for } M > |\mathcal{Q}_f|/(2r)\\
&\le 2^m (2rM)^{\alpha m} |\mathcal{Q}_f|^{(1-\alpha)m},
\end{aligned}$$

where $\mathcal{Q}_f \triangleq \{y : f^{-1}(y) \text{ contains an interval}\}$, and we assumed $|\mathcal{Q}_f| < \infty$. Hence,

$$\limsup_{M\to\infty} \frac{\log\left|\langle\mathcal{B}_\alpha \cap \mathcal{B}_r\rangle_M\right|}{\log M} \le \alpha \cdot m.$$

Combining all the above arguments yields

$$\alpha \cdot m \ge (1 - \epsilon)n,$$

and hence the claimed lower bound on $m/n$. $\qquad\square$

In view of Lemma 9, we can now prove the converse result in Theorem 1.

### A.3   Proof of Theorem 1

*Proof.* From Lemma 10 below, as $m, n \to \infty$ with $m, n \to \delta \in (1, \infty)$, the empirical distribution of $\boldsymbol{z} = \boldsymbol{Ax}$ converges to standard Gaussian in probability

$$\frac{1}{m}\sum_{i=1}^m \mathbb{I}(z_i \le t) \xrightarrow{P} \Phi(t), \quad \forall t \in \mathbb{R}.$$

Consequently,

$$\frac{|\{i = 1, \ldots, m : z_i \in \mathbb{R}\backslash f^{-1}(\mathcal{Q}_f)|}{m} \xrightarrow{P} 1 - \mathbb{P}\{Z \in f^{-1}(\mathcal{Q}_f)\} \triangleq d(Y),$$

where $Z \sim \mathcal{N}(0, 1)$ and $\mathcal{Q}_f \triangleq \{y : f^{-1}(y) \text{ contains an interval}\}$. This is equivalent to (see 21)

$$\frac{|\mathrm{Spt}(f(\boldsymbol{Ax}))|}{m} = \frac{|\mathrm{Spt}(f(\boldsymbol{z}))|}{m} = \frac{|\{i = 1, \ldots, m : f(z_i) \in \mathbb{R}\backslash\mathcal{Q}_f|}{m} \xrightarrow{P} d(Y). \tag{25}$$

Hence, for any $\kappa > 0$,

$$\lim_{n\to\infty} \mathbb{P}\left\{\left|\frac{|\mathrm{Spt}(f(\boldsymbol{Ax}))|}{m} - d(Y)\right| < \kappa\right\} = 1.$$

It is understood that in the above limit $m$ and $n$ tend to infinity with $m/n \to \delta$. In view of the definition of $\mathcal{A}_\alpha$ in (22), we have

$$\lim_{n\to\infty} \mathbb{P}\left\{\boldsymbol{x} \in \mathcal{A}_\alpha\right\} = 1, \quad \text{for } \alpha = d(Y) + \kappa.$$

Hence, for any $\epsilon > 0$ and $\alpha = d(Y) + \kappa$, there exists sufficiently large $n, m$ such that

$$\mathbb{P}\left\{\boldsymbol{x} \in \mathcal{A}_\alpha\right\} \geq 1 - \frac{\epsilon}{3}.$$

Further, since $\lim_{r \to \infty} \mathbb{P}\left\{\boldsymbol{x} \in \mathcal{A}_\alpha \cap \mathcal{A}_r\right\} = \mathbb{P}\left\{\boldsymbol{x} \in \mathcal{A}_\alpha\right\}$, there exists sufficiently large $r$ such that

$$\mathbb{P}\left\{\boldsymbol{x} \in \mathcal{A}_\alpha \cap \mathcal{A}_r\right\} \geq \mathbb{P}\left\{\boldsymbol{x} \in \mathcal{A}_\alpha\right\} - \frac{\epsilon}{3} \geq 1 - \frac{2\epsilon}{3}. \tag{26}$$

Suppose that the decoding error probability does not exceed $\epsilon/3$, namely,

$$\mathbb{P}\{\boldsymbol{x} \in \mathcal{A}\} \geq 1 - \frac{\epsilon}{3}. \tag{27}$$

For $\alpha = d(Y) + \kappa$, and sufficiently large $r$ and $m, n$, we have

$$\mathbb{P}\left\{\boldsymbol{x} \in \mathcal{A} \cap \mathcal{A}_\alpha \cap \mathcal{A}_r\right\} \geq \mathbb{P}\left\{\boldsymbol{x} \in \mathcal{A}\right\} + \mathbb{P}\left\{\boldsymbol{x} \in \mathcal{A}_\alpha \cap \mathcal{A}_r\right\} - 1$$
$$\geq 1 - \epsilon,$$

where the second step is form (26). Now, using Lemma 9, we must have

$$\frac{m}{n} \geq \frac{1 - \epsilon/3}{\alpha} = \frac{1 - \epsilon/3}{d(Y) + \kappa}.$$

Since $\kappa > 0$ is arbitrary, $m/n > \frac{1-\epsilon/3}{d(Y)}$. Hence, a necessary condition for achieving vanishing decoding error as $m, n \to \infty$ with $m/n \to \delta$ is

$$\delta \geq \frac{1}{d(Y)}.$$

$\square$

**Lemma 10.** *Let $\boldsymbol{z} \triangleq \boldsymbol{A}\boldsymbol{x}$. Under Assumptions (A.1)-(A.3), the following holds almost surely as $m, n \to \infty$ and $m/n \to \delta \in (1, \infty)$:*

$$\boldsymbol{z} \overset{W_2}{\to} Z \sim \mathcal{N}(0, 1),$$

*where $\overset{W_2}{\to}$ denotes convergence in Wasserstein space of order two. In particular, this implies*

$$\frac{1}{m} \sum_{i=1}^{m} \mathbb{I}\left(z_i \leq t\right) \overset{a.s.}{\to} \Phi(t), \quad \forall t \in \mathbb{R}. \tag{28}$$

*Proof.* Let $\boldsymbol{A} = \boldsymbol{U}\boldsymbol{\Sigma}\boldsymbol{V}^\mathsf{T}$. We first prove the following convergence result

$$\lim_{n \to \infty} \frac{1}{n} \|\boldsymbol{\Sigma}\boldsymbol{V}^\mathsf{T}\boldsymbol{x}\| \overset{a.s.}{=} 1.$$

Since $\boldsymbol{V}$ is Haar distributed, we have $\boldsymbol{V}^\mathsf{T}\boldsymbol{x} \overset{d}{=} \|\boldsymbol{x}\|/\|\boldsymbol{g}\| \cdot \boldsymbol{g}$ where $\boldsymbol{g} \sim \mathcal{N}(\boldsymbol{0}, \boldsymbol{I})$, and $\overset{d}{=}$ means that the random vectors on the left and right hand sides have the same distribution [46]. Hence,

$$\boldsymbol{\Sigma}\boldsymbol{V}^\mathsf{T}\boldsymbol{x} \overset{d}{=} \frac{\|\boldsymbol{x}\|}{\|\boldsymbol{g}\|} \cdot \boldsymbol{\Sigma}\boldsymbol{g}, \quad \boldsymbol{g} \sim \mathcal{N}(\boldsymbol{0}, \boldsymbol{I}).$$

As $\boldsymbol{x}$ has i.i.d. entries with unit variance, by the law of large numbers, we have $\lim_{n \to \infty} \|\boldsymbol{x}\|/\|\boldsymbol{g}\| \overset{a.s.}{=} 1$. By Assumption (A.2), the empirical distribution of $\{\sigma_i\}_{i=1}^{n}$ converges to $P_\lambda$ almost surely, and further $n^{-1} \sum_{i=1}^{n} \sigma_i^2 \overset{a.s.}{\to} 1$. In other words, the empirical distribution of $\{\sigma_i\}_{i=1}^{n}$ converges, almost surely, in Wasserstein distance of order two; see [58, Definition 6.7]. Further, $\boldsymbol{g} \overset{W_2}{\to} Z \sim \mathcal{N}(0, 1)$. Then, by Fan [21, Proposition B.4], $\boldsymbol{\Sigma}\boldsymbol{g} \overset{W_2}{\to} \Lambda Z$ almost surely, where $\Lambda \sim P_\lambda$ and $Z \sim \mathcal{N}(0, 1)$. Let $F(v) = v^2$. Convergence in $W_2$ implies [58, Definition 6.7]

$$\frac{1}{n} \sum_{i=1}^{n} F((\boldsymbol{\Sigma}\boldsymbol{g})_i) \overset{a.s.}{\to} \mathbb{E}[F(\Lambda Z)] = 1,$$

where the equality follows the normalization $\mathbb{E}[\Lambda^2] = 1$. We then have $\|\boldsymbol{\Sigma}\boldsymbol{g}\|/\sqrt{n} \overset{a.s.}{\to} 1$. Combining these arguments yields the following result

$$\frac{1}{\sqrt{n}}\|\boldsymbol{\Sigma}\boldsymbol{V}^\mathsf{T}\boldsymbol{x}\| \overset{a.s.}{\to} 1.$$

Then, by [21, Proposition C.1], almost surely we have

$$\boldsymbol{z} \triangleq \boldsymbol{U}\boldsymbol{\Sigma}\boldsymbol{V}^T\boldsymbol{x} \overset{W_2}{\to} Z, \quad Z \sim \mathcal{N}(0,1),$$

which further implies the weak convergence of (28); see [58, Definition 6.7] for details about convergence of probability measure in Wasserstein distance. $\qquad\square$

# B  Proof of Theorem 2

The proof is analogous to [65, Theorem 9]. Notice that we assumed $\boldsymbol{x} \sim \mathcal{N}(\boldsymbol{0}, \boldsymbol{I})$ in Theorem 2.

Let $R_X(D)$ be the rate distortion functions of $P_X$ with mean square error distortion:

$$R_X(D) = \inf_{\mathbb{E}[d(X,\hat{S})] \le D, X \sim P_X} I(X; \hat{S}),$$

where $d(X, \hat{S}) := (X - \hat{S})^2$, $I(X; \hat{S})$ denotes the mutual information between $X$ and $\hat{S}$, and the infimum in the above definition is over the transition probability $P_{\hat{S}|X}$ subject to average distortion constraint. Notice that $R_X(D)$ can be equivalently defined as [25, Theorem 9.6.1]

$$R_X(D) = \inf_{\mathbb{E}[d_n(\boldsymbol{x},\hat{\boldsymbol{s}})] \le D, \{x_i\} \overset{i.i.d.}{\sim} P_X} \frac{1}{n} I(\boldsymbol{x}; \hat{\boldsymbol{s}}),$$

where $d_n(\boldsymbol{x}, \hat{\boldsymbol{s}}) := \frac{1}{n}\sum_{i=1}^n (x_i - \hat{s}_i)^2$.

Consider the MMSE estimator $\hat{\boldsymbol{x}} = \mathbb{E}[\boldsymbol{x}|\boldsymbol{y}, \boldsymbol{A}]$ with mean square distortion

$$D_n(\sigma_w) \triangleq \frac{1}{n}\mathsf{mmse}(\boldsymbol{x}|\boldsymbol{y}, \boldsymbol{A}) = \frac{1}{n}\sum_{i=1}^n \mathbb{E}(x_i - \hat{x}_i)^2,$$

where the expectation is over the joint distribution of $\boldsymbol{x}$ and $\hat{\boldsymbol{x}}$. In what follows, we will sometimes write $D_n(\sigma_w)$ as $D_n$ for notational convenience. By the definition of rate distortion functions,

$$n \cdot R_X(D_n) \le I(\boldsymbol{x}, \hat{\boldsymbol{x}}). \tag{29}$$

Denote by $I(\hat{\boldsymbol{x}}; \boldsymbol{A}, \boldsymbol{x})$ the mutual information between $\hat{\boldsymbol{x}}$ and $(\boldsymbol{A}, \boldsymbol{x})$. We have

$$\begin{aligned} I(\boldsymbol{x}; \hat{\boldsymbol{x}}) &\le I(\boldsymbol{x}; \boldsymbol{A}, \hat{\boldsymbol{x}}) \\ &= \underbrace{I(\boldsymbol{x}; \boldsymbol{A})}_{0} + I(\boldsymbol{x}; \hat{\boldsymbol{x}}|\boldsymbol{A}). \end{aligned}$$

Hence,

$$n \cdot R_X(D_n) \le I(\boldsymbol{x}; \hat{\boldsymbol{x}}|\boldsymbol{A}). \tag{30}$$

Denote $\boldsymbol{z} \triangleq \boldsymbol{A}\boldsymbol{x}$ and $\boldsymbol{y} = f(\boldsymbol{z} + \boldsymbol{w})$. For every realization of $\boldsymbol{A}$, we have the following Markov chain:

$$\boldsymbol{x} \to \boldsymbol{z}_A \to \boldsymbol{y}_A \to \hat{\boldsymbol{x}}_A,$$

where the subscript "$_A$" is added to emphasize the fact that $\boldsymbol{A}$ is fixed. From data processing inequality [25, Theorem 4.3.3], we have $I(\boldsymbol{x}; \hat{\boldsymbol{x}}_A) \le I(\boldsymbol{z}_A; \boldsymbol{y}_A)$. Further averaging over $\boldsymbol{A}$ yields

$$\begin{aligned} I(\boldsymbol{x}; \hat{\boldsymbol{x}}|\boldsymbol{A}) &\le I(\boldsymbol{z}; \boldsymbol{y}|\boldsymbol{A}) \\ &\le \sum_{i=1}^m I(z_i; y_i|\boldsymbol{A}) \\ &= \sum_{i=1}^m I(z_i; y_i|\boldsymbol{a}_i) \\ &= m \cdot I(z; y|\boldsymbol{a}), \end{aligned} \tag{31}$$

where $\boldsymbol{a}_i$ denotes the $i$-th row of $\boldsymbol{A}$, the second inequality is follows from [25, Eq. (7.2.19)] (note that $\{y_i\}$ are conditionally independent given $\boldsymbol{z}$), and in the last inequality we dropped the subscripts as the joint distributions of $\{(\boldsymbol{z}_i, \boldsymbol{y}_i, \boldsymbol{a}_i)\}$ are identical due to the rotationally-invariance of $\boldsymbol{A}$. Combining (30) and (31) gives us the following lower bound on $m/n$:

$$\frac{m}{n} \geq \frac{R_X(D_n)}{I(z; y|\boldsymbol{a})}. \tag{32}$$

Now, suppose that

$$M^*(X, f, \delta) = \sup_{\sigma_w} \limsup_{n \to \infty} \frac{\frac{1}{n}\mathsf{mmse}(\boldsymbol{x}|\boldsymbol{y}, \boldsymbol{A})}{\sigma_w^2} < \infty.$$

Then, there exits $C > 0$ such that the following holds for sufficiently large $n$

$$D_n = \frac{1}{n}\mathsf{mmse}(\boldsymbol{x}|\boldsymbol{y}, \boldsymbol{A}) \leq C \cdot \sigma_w^2, \quad \forall \sigma_w > 0.$$

The following arguments are similar to the proof of [65, Theorem 9]. Let $R_X^{-1}$ be the inverse function of $R_X$. (Since $R_X$ is a monotonically decreasing function, its inverse exists.) We have

$$R_X(D_n) \geq R_X(C \cdot \sigma_w^2), \quad \forall \sigma_w > 0.$$

Hence, the following holds for any $\sigma_w > 0$,

$$\begin{aligned}
\frac{n}{m} &\leq \frac{I(z; y|\boldsymbol{a})}{R_X(C \cdot \sigma_w^2)} \\
&= \frac{I(z; y|\boldsymbol{a})}{\frac{1}{2}\log\frac{1}{C \cdot \sigma_w^2}} \cdot \frac{\frac{1}{2}\log\frac{1}{\sigma_w^2}}{R_X(C \cdot \sigma_w^2)}.
\end{aligned} \tag{33}$$

When $X \sim \mathcal{N}(0, 1)$, we have [63]

$$\lim_{\sigma_w \to 0} \frac{R_X(C \cdot \sigma_w^2)}{\frac{1}{2}\log\frac{1}{C \cdot \sigma_w^2}} = 1. \tag{34}$$

Further, from Lemma 11, we have

$$\limsup_{\sigma_w \to 0} \frac{I(z; y|\boldsymbol{a})}{\frac{1}{2}\log\frac{1}{\sigma_w^2}} \leq 1 - \mathbb{E}_{\boldsymbol{a}}[\mathbb{P}\{z_a \in f^{-1}(\mathcal{Q}_f)\}], \tag{35}$$

where $z_a \sim \mathcal{N}(0, \|\boldsymbol{a}\|^2)$, and $\boldsymbol{a}$ has the same distribution as the first row of $\boldsymbol{A}$. Note that the proof of Lemma 10 shows that $\|\boldsymbol{A}\boldsymbol{s}\| \overset{a.s.}{\to} 1$ as $m, n \to \infty$ with $m/n \to \delta$, whenever $\|\boldsymbol{s}\| \to 1$. Hence, $\|\boldsymbol{a}\| \overset{a.s.}{\to} 1$ where $\boldsymbol{a}$ is an arbitrary row of $\boldsymbol{A}$. Recall that $z \sim \mathcal{N}(0, \|\boldsymbol{a}\|^2)$. It is easy to show that $\mathbb{P}\{z_a \in f^{-1}(\mathcal{Q}_f)\}$ is a continuous function of $\|\boldsymbol{a}\|^2$, and by continuous mapping theorem we have $\mathbb{P}\{z_a \in f^{-1}(\mathcal{Q}_f)\} \overset{a.s.}{\to} \mathbb{P}\{Z \in \mathcal{Q}_f\}$ where $Z \sim \mathcal{N}(0, 1)$. Then by dominated convergence theorem, the following holds as $m, n \to \infty$ with $m/n \to \delta$,

$$\mathbb{E}_{\boldsymbol{a}}[\mathbb{P}\{z_a \in f^{-1}(\mathcal{Q}_f)\}] \to \mathbb{P}\{Z \in \mathcal{Q}_f\}. \tag{36}$$

Combining (33)-(36) and passing to the limit yields our desired result.

**Lemma 11.** *Let $\boldsymbol{a}$ be the first row of $\boldsymbol{A}$, and $z = (\boldsymbol{A}\boldsymbol{x})_1$, $y = f(z + w)$, where $w \sim \mathcal{N}(0, \sigma_w^2)$ and $w \perp\!\!\!\perp (\boldsymbol{A}, \boldsymbol{x})$. We have*

$$\limsup_{\sigma_w \to 0} \frac{I(z; y|\boldsymbol{a})}{\log(\sigma_w^{-2})} \leq 1 - \mathbb{E}\left[\mathbb{P}\{z_a \in f^{-1}(Q_f)\}\right],$$

*where $z_a \sim \mathcal{N}(0, \|\boldsymbol{a}\|^2)$.*

*Proof.* We use $z_a$ to denote a random variable with distribution $P_{z|\boldsymbol{a}}$, $y_a = f(z_a + w)$ and $z_a \perp\!\!\!\perp w$. Note that $z = \boldsymbol{a}^\mathsf{T}\boldsymbol{x}$, and so $z_a \sim \mathcal{N}(\boldsymbol{0}, \|\boldsymbol{a}\|^2)$. Then, by the reverse Fatou lemma, we have

$$\limsup_{\sigma_w \to 0} \frac{I(z; y|\boldsymbol{a})}{\frac{1}{2}\log\sigma_w^{-2}} = \limsup_{\sigma_w \to 0} \mathbb{E}\left[\frac{I(z_a; y_a)}{\frac{1}{2}\log\sigma_w^{-2}}\right] \leq \mathbb{E}\left[\limsup_{\sigma_w \to 0} \frac{I(z_a; y_a)}{\frac{1}{2}\log\sigma_w^{-2}}\right].$$

In what follows, we prove

$$\limsup_{\sigma_w \to 0} \frac{I(z_a; y_a)}{\frac{1}{2}\log \sigma_w^{-2}} = \mathbb{P}\{z_a \in \mathbb{R}\backslash \mathcal{Q}_f\}, \tag{37}$$

where $z_a \in \mathcal{N}(0, \|\boldsymbol{a}\|^2)$. For convenience, define

$$p_a \triangleq z_a + w. \tag{38}$$

Using this notation, $y_a = f(p_a)$. We introduce an auxiliary random variable

$$Q \triangleq \mathbb{I}(p_a \in f^{-1}(\mathcal{Q}_f)).$$

Then,

$$
\begin{aligned}
I(z_a; f(z_a + w)) &\leq I(z_a; f(z_a + w), Q) \\
&= I(z_a; Q) + I(z_a; f(z_a + w)|Q) \\
&= I(z_a; Q) + \mathbb{P}\{Q = 1\} \cdot I(z_a; f(z_a + w)|Q = 1) + \mathbb{P}\{Q = 0\} \cdot I(z_a; f(z_a + w)|Q = 0) \\
&\leq 1 + \log(|\mathcal{Q}_f|) + \mathbb{P}\{Q = 0\} \cdot I(z_a; f(z_a + w)|Q = 0)
\end{aligned}
\tag{39}
$$

where the last inequality follows from the fact that $I(z_a; Q) \leq 1$ for the binary random variable $Q$, and $I(z_a; f(z_a + w)|Q = 1) \leq \log(|\mathcal{Q}_f|)$ since $f(z_a + w)$ takes at most $|\mathcal{Q}_f|$ different values conditional on $Q = 1$.

Since $z_a \in \mathcal{N}(0, \|\boldsymbol{a}\|^2)$ and $w \sim \mathcal{N}(0, \sigma_w^2)$, we can represent $p_a = z_a + w$ as

$$z_a \overset{d}{=} \underbrace{\frac{\|\boldsymbol{a}\|^2}{\|\boldsymbol{a}\|^2 + \sigma_w^2}}_{\alpha} \cdot p_a + \underbrace{\sqrt{\frac{\|\boldsymbol{a}\|^2 \sigma_w^2}{\|\boldsymbol{a}\|^2 + \sigma_w^2}}}_{\beta} \cdot N$$

where $N$ is standard Gaussian and independent of $p_a$. Hence, $N$ is still independent of $p_a$ conditioned on $Q = \mathbb{I}(p_a \in f^{-1}(\mathcal{Q}_f)) = 0$. Hence, the conditional distribution of $(z_a, p_a)$ is characterized by

$$\tilde{z}_a = \alpha \cdot \tilde{p}_a + \beta N,$$

where $\tilde{p}_a \sim P_{p_a|Q=0} = P_{p_a|p_a \in \mathbb{R}\backslash f^{-1}(\mathcal{Q}_f)}$. Since $f(\tilde{p}_a) \to \tilde{p}_a \to z_a$ forms a Markov chain, by data processing inequality, we have

$$
\begin{aligned}
I(z_a; f(z_a + w)|Q = 0) = I(\tilde{z}_a; f(\tilde{p}_a)) &\leq I(\tilde{z}_a; \tilde{p}_a) \\
&= \frac{1}{2}\log\left(1 + \frac{\alpha^2}{\beta^2}\mathbb{E}[\tilde{p}_a^2]\right) \\
&= \frac{1}{2}\log\left(1 + \frac{\|\boldsymbol{a}\|^2}{\|\boldsymbol{a}\|^2 + \sigma_w^2}\frac{\mathbb{E}[\tilde{p}_a^2]}{\sigma_w^2}\right).
\end{aligned}
$$

It is easy to show that $\mathbb{E}[\tilde{p}_a^2]$ converges to a positive constant as $\sigma_w \to 0$, and

$$\limsup_{\sigma_w \to 0} \frac{I(z_a; f(z_a + w)|Q = 0)}{\frac{1}{2}\log(\sigma_w^{-2})} \leq 1, \tag{40a}$$

and

$$\lim_{\sigma_w \to 0} \mathbb{P}\{Q = 0\} = \mathbb{P}\{z_a \in \mathbb{R}\backslash \mathcal{Q}_f\}. \tag{40b}$$

Combining (39) and (40) proves Lemma 11. $\qquad\square$

# C  Proof of Lemma 2

We use the smoothing argument of Zheng et al. [67, Theorem 1]. Roughly speaking, we construct a sequence of smoothed functions $\tilde{\eta}_z$ (indexed by $\xi, M, \sigma$; see (43)), and show that the performance of the corresponding GLM-EP-app algorithm tends to the predicted performance of GLM-EP as $\xi, M, \sigma$ approaches a certain limit. This implies the performance of GLM-EP-app could be made arbitrarily close to the predicted one with proper choice of $\xi, M, \sigma$.

**Remark 3.** *We emphasize that the GLM-EP-app algorithm is introduced mainly for performance analysis purposes. In practice, GLM-EP is preferable. Our simulations suggest that the asymptotic prediction is accurate even for the original GLM-EP under wide choices of $f$ (including the quantization function).*

As many steps of the proof are the same as [67, Theorem 1], we only sketch the main idea here.

## C.1 Constructions of $\tilde{\eta}_z$ and $C_t$

Let $\mathcal{Q}_f \triangleq \{y_q : 1 \le q \le Q\}$ (where $Q < \infty$) be the set for which $f^{-1}$ contains an interval. Let $\xi < \frac{1}{2} \min\{y_p - y_q, p \ne q\}$ and define

$$\eta_z^\xi(z_r, y) \triangleq \begin{cases} \eta_q(z_r) & \text{for } z_r \in \mathbb{R}, y \in (y_q - \xi, y_q + \xi), \forall q = 1, \dots, Q \\ \eta_z(z_r, y) & \text{for } z_r \in \mathbb{R}, y \in \mathcal{Y} \backslash \bigcup_{q=1}^Q (y_q - \xi, y_q + \xi) \\ 0 & \text{for } z_r \in \mathbb{R}, y \in \mathbb{R} \backslash \mathcal{Y} \end{cases} \tag{41a}$$

where we denoted

$$\eta_q(z_r) \triangleq \eta_z(z_r, y_q). \tag{41b}$$

Here, we extended the definition of $\eta_z$ at the isolated points $\{y_1, \dots, y_Q\}$ to their neighborhoods. This treatment ensures $\eta_z^\xi(z_r, y)$ to be continuous at $(z_r, y_q)$, which is a useful property for our analysis. We apply an additional truncation to $\eta_z^\xi(z_r, y)$ (where $M > \max\{|y_1|, \dots, |y_Q|\}$):

$$\eta_z^{\xi,M}(z_r, y) \triangleq \eta_z^\xi(z_r, y) \cdot \mathbb{I}_{[-M,M]^2}(z_r, y), \tag{42}$$

where $\mathbb{I}_{[-M,M]^2}(z_r, y)$ is an indicator function that equals one when $(z_r, y) \in [-M, M]^2$ and zero elsewhere. Finally, we smooth $\eta_z^{\xi,M}(z_r, y)$ by convolving it with a Gaussian kernel[4]:

$$\begin{aligned} \eta_z^{\sigma,\xi,M}(z_r, y) &\triangleq \eta_z^{\xi,M}(z_r, y) \star \phi_\sigma(z_r, y) \\ &= \iint_{\mathbb{R}^2} \eta^{\xi,M}(s, t) \cdot \frac{1}{2\pi\sigma^2} \exp\left(-\frac{(s - z_r)^2 + (t - y)^2}{2\sigma^2}\right) ds\, dt \end{aligned} \tag{43}$$

Some useful properties of $\eta_z^{\sigma,\xi,M}(z_r, y)$ and $\eta_z$ are given in Section C.3 (see Lemma 12 and Lemma 13).

In the GLM-EP-app algorithm (see (8)), the function $\tilde{\eta}_z$ and the constant $C_t$ are given by

$$\tilde{\eta}_z(z_r, y, v_r) = \eta_z^{\sigma,\xi,M}(z_r, y, v_r), \tag{44a}$$

and

$$C_t = \frac{1}{1 - \mathbb{E}\left[\eta_z'(Z_r^{t-1}), Y, V_r^{-1}\right]} \tag{44b}$$

where the expectation in (44b) are taken with respect to $Z_r^{t-1} \sim \mathcal{N}(0, 1 - V_r^{t-1})$, $Z = Z_r + \mathcal{N}(0, V_r^{t-1})$ and $Y = f(Z)$. Notice that $C_t$ depends on the the original function $\eta_z$, not the smoothed and truncated function $\eta_z^{\sigma,\xi,M}$. This choice is for the purpose for simplifying our analysis.

## C.2 Proof sketch for Lemma 2

Our proof for Lemma 2 follows the approach proposed in Zheng et al. [67, Theorem 1]. *As many steps are similar to Lemma 2, we will not provide the full details of the proof, and only sketch the main idea.* The proof has two main steps:

(Step 1) As the smoothed function $\eta_z^{\sigma,\xi,M}$ is Lipschitz continuous, the asymptotic MSE of GLM-EP-app could be characterized by a state evolution (SE) recursion;

(Step 2) Using the SE platform, we show that the asymptotic MSE of GLM-EP-app converges to the (expected) MSE of GLM-EP, as $\sigma \to 0$, and $\xi \to 0$, $M \to \infty$ sequentially. This implies that, with proper choice of $\sigma, \xi, M$, the asymptotic performance of GLM-EP-app is arbitrarily close to that of GLM-EP.

Step 1 is a consequence of [24, Theorem 1]. Note that the model considered in this paper is a special case of that adopted in [24, Theorem 1]. Also, here we assumed $f$ to be Lipschitz continuous, as required by [24, Theorem 1]. The crucial assumption of [24, Theorem 1] is the Lipschitz continuity of $\eta_z^{\sigma,\xi,M}$, which we prove in Lemma 12 (see Section C.3).

---

[4]The smoothing parameter $\sigma$ should not be confused with $\sigma_w$, which denotes the noise variance in Section 2.3.

A caveat is that [24, Theorem 1] assumes $\eta_z^{\sigma,\xi,M}(z_r, y, v_r)$ to be uniform Lipschitz (see definition in [24]) w.r.t. to $(z_r, y)$ and $v_r$. However, since GLM-EP-app uses the deterministic sequences $\{V_r^t, V_l^t\}_{t\geq 0}$ instead of their empirical counterparts $\{v_r^t, v_l^t\}$, this additional uniform continuity assumption is not required here.

Step 2 follows the same argument as in [67, Theorem 1]. First, the state evolution of GLM-EP-app is slightly more complicated than that of GLM-EP, and involve four sequences $\{\alpha_l^t, \tau_l^t, \alpha_r^t, \tau_r^t\}_{t\geq 0}$. (The SE of GLM-EP can be viewed as a special case of this more general SE.) *Note that these sequences all depend on the parameters $\sigma, \xi, M$, but to keep notation light we do not make such dependency explicit.* Intuitively speaking, $(\alpha_l^t, \tau_l^t)$ describes the correlation matrix of the components of $(z, z_l^t)$ (where $z \triangleq Ax$):

$$\mathrm{Cov}(Z, Z_l^t) \triangleq \begin{bmatrix} \mathbb{E}[Z^2] & \mathbb{E}[ZZ_l^t] \\ \mathbb{E}[ZZ_l^t] & \mathbb{E}[(Z_l^t)^2] \end{bmatrix} = \begin{bmatrix} 1 & \alpha_l^t \\ \alpha_l^t & \tau_l^t \end{bmatrix}.$$

Similarly, $(\alpha_r^t, \tau_r^t)$ describes the correlation of the components of $(z, z_r^t)$ The SE describing the recursive relationship of $\{\alpha_l^t, \tau_l^t, \alpha_r^t, \tau_r^t\}_{t\geq 0}$ is given by

$$\alpha_l^t = \phi_1^{\sigma,\xi,M}(\alpha_r^t, \sigma_r^t), \quad \text{and} \quad \tau_l^t = \phi_2^{\sigma,\xi,M}(\alpha_r^t, \sigma_r^t),$$
$$\alpha_r^t = \Phi_1(\alpha_r^t, \sigma_r^t), \quad \text{and} \quad \tau_r^t = \Phi_2(\alpha_r^t, \sigma_r^t),$$

where GLM-EP and GLM-EP-app start from the same initializations, i.e., $\alpha_r^{-1} = \tau_r^{-1} = V_r^{-1}$. A formal definition of these functions may be found in, e.g., [24].

Our goal is to show that the limit the covariance $\mathrm{Cov}(Z, Z_l^t)$ for GLM-EP and GLM-EP-app for all $t \geq 0$. Note that if $\mathrm{Cov}(Z, Z_l^t)$ for GLM-EP and GLM-EP-app are the same, then $\mathrm{Cov}(Z, Z_r^t)$ would also be the same, as the second steps of the two algorithms are identical (cf. (6) and (8b)).

As in [67, Theorem 1], the argument proceeds inductively on $t$. Because the steps are straightforward, we do not provide the full details and only consider the first iteration. Basically, we need to prove the following:

$$\lim_{\xi\to 0, M\to\infty} \lim_{\sigma\to 0} \mathbb{E}[Z \cdot \eta_z^{\sigma,\xi,M}(Z_r, Y)] = \mathbb{E}[Z \cdot \eta_z(Z_r, Y)] \tag{45a}$$

$$\lim_{\xi\to 0, M\to\infty} \lim_{\sigma\to 0} \mathbb{E}[Z_r \cdot \eta_z^{\sigma,\xi,M}(Z_r, Y)] = \mathbb{E}[Z_r \cdot \eta_z(Z_r, Y)] \tag{45b}$$

$$\lim_{\xi\to 0, M\to\infty} \lim_{\sigma\to 0} \mathbb{E}[(\eta_z^{\sigma,\xi,M}(Z_r, Y))^2] = \mathbb{E}[(\eta_z(Z_r, Y))^2] \tag{45c}$$

where $Z \sim \mathcal{N}(0, 1), Y = f(Z)$. Note that these results hold for any $Z_r$ as long as it is joint Gaussian with $Z$ (non-degenerate).

We next prove (45a). Other results can be proved in the same way. We first calculate its limit of $\mathbb{E}[Z \cdot \eta_z^{\sigma,\xi,M}(Z_r, Y)]$ as $\sigma \to 0$. From Lemma 12, the function $\eta_z^{\sigma,\xi,M}$ is bounded, and using DCT we get

$$\lim_{\sigma\to 0} \mathbb{E}[Z \cdot \eta_z^{\sigma,\xi,M}(Z_r, Y)] = \mathbb{E}\left[Z \cdot \lim_{\sigma\to 0} \eta_z^{\sigma,\xi,M}(Z_r, Y)\right]$$
$$= \mathbb{E}\left[Z \cdot \eta_z^{\xi,M}(Z_r, Y)\right],$$

where the last step follows from

$$\lim_{\sigma\to 0} \eta_z^{\sigma,\xi,M}(Z_r, Y) = \eta_z^{\xi,M}(Z_r, Y) \quad \text{a.s.} \tag{46}$$

To see (46), note that Lemma 12 shows $\lim_{\sigma\to 0} \eta_z^{\sigma,\xi,M}(z_r, y) = \eta_z^{\xi,M}(z_r, y)$ whenever $\eta_z^{\xi,M}$ is continuous at $(z_r, y)$. In particular, our construction of $\eta_z^{\xi,M}$ (see (41a)) guarantees that $\eta_z^{\xi,M}$ at $(z_r, y) \in \mathbb{R} \times \{y_1, \ldots, y_Q\}$. Similar to the proof of Lemma 12-(P.1), it can be shown that the set of points at which $\eta_z^{\xi,M}$ is discontinuous has zero probability (with respect to the distribution of $(Z_r, Y)$).

It remains to prove

$$\lim_{\xi\to 0, M\to\infty} \mathbb{E}\left[Z \cdot (\eta_z^{\xi,M}(Z_r, Y) - \eta_z(Z_r, Y))\right] = 0.$$

Similar to 12-(P.1), it can be shown that $\eta_z^{\xi,M}(Z_r, Y)$ is almost surely bounded w.r.t, the distribution of $Z_r, Y$. Also, by Lemma 13, $\eta_z(Z_r, Y) = \eta_z(Z_r, f(Z)) \leq C \cdot (1 + \|(Z_r, Z)\|)$. Hence, we could apply DCT to show

$$\lim_{\xi \to 0, M \to \infty} \mathbb{E}\left[ Z \cdot (\eta_z^{\xi,M}(Z_r, Y) - \eta_z(Z_r, Y)) \right]$$

$$= \mathbb{E}\left[ \lim_{\xi \to 0, M \to \infty} Z \cdot (\eta_z^{\xi,M}(Z_r, Y) - \eta_z(Z_r, Y)) \right]$$

$$= 0.$$

### C.3 Auxiliary results

**Lemma 12.** *Let $v_r, \xi, M, \sigma > 0$. The following hold*

(P.1) $\eta_z^\xi(z_r, y)$ *is continuous a.e. with respect to the Lebesgue measure. Further, $\eta_z^{\xi,M}(z_r, y)$ is a.e. bounded;*

(P.2) $\eta_z^{\sigma,\xi,M}(z_r, y)$ *is Lipschitz continuous and bounded on $\mathbb{R}^2$;*

(P.3) $\lim_{\sigma \to 0} \eta_z^{\sigma,\xi,M}(z_r, y) = \eta_z^{\xi,M}(z_r, y)$ *whenever $\eta_z^{\xi,M}$ is continuous at $(z_r, y)$.*

*Proof. Proof of (P.1):* We note that $\eta_q(z_r)$ $(q = 1, \ldots, Q)$ is a continuous function of $z_r \in \mathbb{R}$:

$$\eta_q(z_r) \triangleq \eta_z(z_r, y_q) = \frac{\int_{f^{-1}(y_q)} u \cdot \mathcal{N}(u; z_r, v_r) du}{\int_{f^{-1}(y_q)} \mathcal{N}(u; z_r, v_r) du}.$$

By definition of $y_q$, $f^{-1}(y_q)$ contains an interval (could be union of intervals), and it is straightforward to show that $\eta_q(z_r)$ is continuous on $\mathbb{R}$.

When $y \in \mathcal{Y} \backslash \mathcal{Q}_f$, $f^{-1}(y)$ is a finite set and we have (see (6f))

$$\eta_z(z_r, y) = \frac{\sum_{u_i \in f^{-1}(y)} u_i \cdot \mathcal{N}(u_i; z_r, v)}{\sum_{u_i \in f^{-1}(y)} \mathcal{N}(u_i; z_r, v)}.$$

By the piecewise monotone (and continuous) assumption of $f$, it can be shown that $\mathcal{Y} \backslash \mathcal{Q}_f$ can be further decomposed into several non-overlapping intervals, denoted as $\mathcal{Y} \backslash \mathcal{Q}_f = \bigcup_{j=1}^J \mathcal{Y}_j$ (where $J < \infty$), such that $\eta_z(z_r, y)$ is continuous on $\mathbb{R} \times \mathcal{Y}_j, \forall j$. This is due to the fact that each point of $f^{-1}(y)$ is a continuous function of $y$ for $y \in \mathcal{Y}_j$. Specifically, it is possible to write $f^{-1}(y)$ as

$$f^{-1}(y) = \{F_j^1(y), \ldots, F_j^{K_j}(y)\}, \quad \forall y \in \mathcal{Y}_j,$$

where $K_j < \infty$, and each $F_j^k(y)$ is a continuous function of $y$ (by piecewise continuity of $f$). Hence,

$$\eta_z(z_r, y) = \frac{\sum_{k=1}^{K_j} F_j^k(y) \cdot \mathcal{N}(F_j^k(y); z_r, v)}{\sum_{k=1}^{K_j} \mathcal{N}(F_j^k(y); z_r, v)}, \quad \forall (z_r, y) \in \mathbb{R} \times \mathcal{Y}_j,$$

and it is continuous on *the interior* of $\mathbb{R} \times \mathcal{Y}_j$. As an example, consider $f$ given in (15) (see illustration on the left panel of Figure 1). In this case,

$$f^{-1}(y) = \begin{cases} \{-y-1, y+1\}, & \text{for } y > 1 \\ \{-y-1, y+1, -y, y\}, & \text{for } 0 \leq y \leq 1 \end{cases}$$

It can be shown that $\eta_z(z_r, y)$ is continuous on $\mathbb{R} \times (1, \infty)$ and $\mathbb{R} \times (0, 1)$.

The claimed a.e. continuity of $\eta_z^\xi$ (see definition in (41a)) follows from the above properties of $\eta_z(z_r, y)$.

Since $\eta_z^\xi(z_r, y)$ is continuous almost everywhere (with respect to the Lebesgue measure), $\eta_z^{\xi,M}(z_r, y) = \eta^\xi(z_r, y) \cdot \mathbb{I}_{[-M,M]^2}(z_r, y)$ is bounded almost everywhere. Let $M' < \infty$ denote

this a.e. bound of $|\eta_{\dot{z}}^{\xi,M}|$. Then, the smoothed function $|\eta_z^{\sigma,\xi,M}(z_r, y)|$ is upper bounded by $M'$ on $\mathbb{R}^2$.

*Proof of (P.2):* With slight abuse of notations, let $\phi_\sigma(s, t)$ denote the bivariate and univariate Gaussian pdf functions respectively. (Namely, $\phi_\sigma(s, t) = \phi_\sigma(s)\phi_h(t)$). To prove Lipschitz continuity, note that

$$|\eta_z^{\sigma,\xi,M}(z_1, y_1) - \eta_z^{\sigma,\xi,M}(z_2, y_2)|$$
$$\leq \iint \eta_{\dot{z}}^{\xi,M}(s, t) \, |\phi_\sigma(z_1 - s)\phi_\sigma(y_1 - t) - \phi_\sigma(z_2 - s)\phi_\sigma(y_2 - t)| \, dsdt$$
$$\leq 8M'M^2 \|\phi_\sigma\|_\infty \|\phi_\sigma'\|_\infty \cdot \|(y_1, z_1) - (y_2, z_2)\|,$$

where we used

$$|\phi_\sigma(z_1 - s)\phi_\sigma(y_1 - t) - \phi_\sigma(z_2 - s)\phi_\sigma(y_2 - t)|$$
$$\leq \|\phi_\sigma\|_\infty \cdot (|\phi_\sigma(y_1 - t) - \phi_\sigma(y_2 - t)| + |\phi_\sigma(z_1 - t) - \phi_\sigma(z_2 - t)|)$$
$$\leq \|\phi_\sigma\|_\infty \|\phi_\sigma'\|_\infty \cdot (|y_1 - y_2| + |z_1 - z_2|) \quad \text{(mean value theorem)}$$
$$\leq 2\|\phi_\sigma\|_\infty \|\phi_\sigma'\|_\infty \cdot \|(y_1, z_1) - (y_2, z_2)\|.$$

Hence, the function $\eta^{\sigma,\xi,M}$ is Lipschitz continuous.

*Proof of (P.3):* Let $(z_0, y_0)$ be a point at which $\eta^{\xi,M}$ is continuous. Since $\eta^{\xi,M}$ is bounded almost everywhere, we could apply DCT to get

$$\lim_{\sigma \to 0} \eta^{\sigma,\xi,M}(z_0, y_0) = \lim_{\sigma \to 0} \iint_{\mathbb{R}^2} \eta^{\xi,M}(z_0 + s, y_0 + t) \cdot \frac{1}{2\pi h^2} \exp\left(-\frac{s^2 + t^2}{2\sigma^2}\right) dsdt$$
$$= \lim_{\sigma \to 0} \iint_{\mathbb{R}^2} \eta^{\xi,M}(z_0 + \sigma s, y_0 + \sigma t) \cdot \frac{1}{2\pi} \exp\left(-\frac{s^2 + t^2}{2}\right) dsdt$$
$$= \iint_{\mathbb{R}^2} \lim_{\sigma \to 0} \eta^{\xi,M}(z_0 + \sigma s, y_0 + \sigma t) \cdot \frac{1}{2\pi} \exp\left(-\frac{s^2 + t^2}{2}\right) dsdt$$

Since $\eta^{\xi,M}$ is continuous at $(z_0, y_0)$, we have

$$\lim_{\sigma \to 0} \eta^{\xi,M}(z_0 + \sigma s, y_0 + \sigma t) = \eta^{\xi,M}(z_0, y_0), \quad \forall (s, t) \in \mathbb{R}^2.$$

Combining the two steps completes the proof. $\qquad\square$

**Lemma 13.** *Let $v_r > 0$. There exists a constant $C > 0$ such that*

$$\eta_z(z_r, f(z), v_r) < C \cdot (1 + \|(z_r, z)\|), \quad \forall (z_r, z) \in \mathbb{R}^2. \tag{47}$$

*Proof.* We differentiate between two cases: $f(z) \in \mathcal{Q}_f$ and $f(z) \in \mathcal{Y} \backslash \mathcal{Q}_f$, where $\mathcal{Q}_f = \{y_1, \ldots, y_Q\}$ correspond to the flat sections of $f$.

*Case 1: $f(z) \in \mathcal{Q}_f$.* Assume $f(z) = y_i$ and denote $\mathcal{I}_i \triangleq f^{-1}(y_i)$. In what follows, we will prove

$$\eta_z(z_r, y_i, v_r) = \frac{\int_{\mathcal{I}_i} u \cdot \mathcal{N}(u; z_r, v_r) du}{\int_{\mathcal{I}_i} \mathcal{N}(u; z_r, v_r) du} < C_i \cdot (1 + |z_r|), \quad \forall z_r \in \mathbb{R}.$$

Suppose $\mathcal{I}_i$ can be written as $\mathcal{I}_i = \bigcup_{k=1}^K (a_k, b_k)$, where $a_k$ could be $-\infty$ and $b_k$ could be $\infty$ (we do not index $a_k, b_k$ by $i$ to simplify notation.) Then,

$$\eta_z(z_r, y_i, v_r) = \frac{\int_{\mathcal{I}_i} u \cdot \mathcal{N}(u; z_r, v_r) du}{\int_{\mathcal{I}_i} \mathcal{N}(u; z_r, v_r) du}$$
$$= \sum_{k=1}^K \left( \frac{\int_{(a_k, b_k)} u \cdot \mathcal{N}(u; z_r, v_r) du}{\sum_{j=1}^K \int_{(a_j, b_j)} \mathcal{N}(u; z_r, v_r) du} \right)$$

We have

$$
\begin{aligned}
|\eta_z(z_r, y_i, v_r)| &\leq \sum_{k=1}^{K} \left( \frac{\left| \int_{(a_k, b_k)} u \cdot \mathcal{N}(u; z_r, v_r) du \right|}{\sum_{j=1}^{K} \int_{(a_j, b_j)} \mathcal{N}(u; z_r, v_r) du} \right) \\
&\leq \sum_{k=1}^{K} \left( \frac{\left| \int_{(a_k, b_k)} u \cdot \mathcal{N}(u; z_r, v_r) du \right|}{\int_{(a_k, b_k)} \mathcal{N}(u; z_r, v_r) du} \right)
\end{aligned}
\tag{48}
$$

We bound the terms inside the summation seperately. First, assume both $a_k$ and $b_k$ are finite. Then,

$$
\begin{aligned}
\frac{\left| \int_{(a_k, b_k)} u \cdot \mathcal{N}(u; z_r, v_r) du \right|}{\int_{(a_k, b_k)} \mathcal{N}(u; z_r, v_r) du} &= \frac{\left| \int_{(a_k, b_k)} u \cdot \mathcal{N}(u; z_r, v_r) du \right|}{\int_{(a_k, b_k)} \mathcal{N}(u; z_r, v_r) du} \\
&= \left| z_r + \frac{\int_{(a_k, b_k) - z_r} t \cdot \mathcal{N}(t; 0, v_r) dt}{\int_{(a_k, b_k) - z_r} \mathcal{N}(t; 0, v_r) dt} \right| \\
&\leq |z_r| + \max\{|a_k - z_r|, |b_k - z_r|\} \\
&\leq C'(1 + |z_r|)
\end{aligned}
$$

Now, suppose $a_k = -\infty$. (The argument is similar for the case $b_k = \infty$.) We have

$$
\left| \frac{\int_{(-\infty, b_k - z_r)} t \cdot \mathcal{N}(t; 0, v_r) dt}{\int_{(-\infty, b_k - z_r)} \mathcal{N}(t; 0, v_r) dt} \right| = \left| \sqrt{v_r} \frac{\phi_1 \left( \frac{b_k - z_r}{\sqrt{v_r}} \right)}{\Phi_1 \left( \frac{b_k - z_r}{\sqrt{v_r}} \right)} \right| \leq C''(1 + |z_r|)
$$

where $\phi_1$ and $\Phi_1$ denote the pdf and cdf functions of standard Gaussian distribution, respectively, and the last step is from mean value theorem together with the following elementary result

$$
\left| \left( \frac{\phi_1(x)}{\Phi_1(x)} \right)' \right| \leq 1, \quad \forall x \in \mathbb{R}.
$$

*Case 2:* $f(z) \in \mathcal{Y} \backslash \mathcal{Q}_f$. In this case, $f^{-1}(f(z))$ is a finite set, and

$$
\eta_z(z_r, f(z), v_r) = \frac{\sum_{u_i \in f^{-1}(f(z))} u_i \cdot \exp(-E_i)}{\sum_{u_i \in f^{-1}(f(z))} \exp(-E_i)}, \quad E_i \triangleq \frac{(u_i - v_r)^2}{2v_r}
\tag{49}
$$

Hence,

$$
\begin{aligned}
|\eta_z(z_r, f(z), v_r)| &\leq \frac{\sum_{u_i \in f^{-1}(f(z))} |u_i| \cdot \exp(-E_i)}{\sum_{u_i \in f^{-1}(f(z))} \exp(-E_i)} \\
&\leq \sum_{u_i \in f^{-1}(f(z))} |u_i| \cdot \exp(-E_i + E_{\min}),
\end{aligned}
\tag{50}
$$

where

$$
E_{\min} = \min \{E_j\}
\tag{51}
$$

From the piecewise assumption of $f$, we have that $|f^{-1}(f(z))| < K$ for all $f(z) \in \mathcal{Y} \backslash \mathcal{Q}_f$. It suffices to prove the following for $1 \leq i \leq |f^{-1}(f(z))|$:

$$
|u_i| \cdot \exp(-E_i + E_{\min}) < C(1 + \|(z, z_r)\|) \quad \forall (z, z_r) \in \mathbb{R}^2.
$$

Denote

$$
t_i \triangleq \exp(-E_i + E_{\min}) = \exp\left( -\frac{(u_i - v_r)^2}{2v_r} + E_{\min} \right).
$$

(As $E_i \geq E_{\min}$, we have $0 < t_i \leq 1$.) From this definition,

$$
|u_i - z_r| = \sqrt{2v_r \cdot \left( E_{\min} + \log \frac{1}{t_i} \right)}.
$$

Hence,

$$|u_i| \leq |z_r| + \sqrt{2v_r \cdot \left(E_{\min} + \log \frac{1}{t_i}\right)}.$$

Then,

$$|u_i| \cdot \exp\left(-E_i + E_{\min}\right) = |u_i| \cdot t_i$$

$$\leq |z_r| \cdot t_i + \sqrt{2v_r \cdot \left(t_i^2 \cdot E_{\min} + t_i^2 \cdot \log \frac{1}{t_i}\right)}$$

$$\overset{(a)}{\leq} |z_r| \cdot t_i + \sqrt{2v_r \cdot \left(t_i^2 \cdot \frac{(z - z_r)^2}{2v_r} + t_i^2 \cdot \log \frac{1}{t_i}\right)}$$

$$\overset{(b)}{\leq} |z_r| + \sqrt{(z - z_r)^2 + 0.4v_r}$$

$$< C \cdot (1 + \|(z, z_r)\|),$$

where step (a) is from the definition of $E_{\min}$ and the fact that $z \in f^{-1}(f(z)))$, and step (b) is due to $0 < t_i \leq 1$ and $t_i^2 \log(1/t_i) < 0.2$. $\qquad\square$

## D  Proofs of Lemma 3 and Lemma 4

### D.1  Proof of Lemma 3

From (7), the state evolution recursion for $V_r$ reads

$$V_r^{t+1} = \Phi\left(\phi(V_r^t)\right),$$

where $V_r^0 = 1$. Lemma 19 in Appendix H implies that the composite function $\Phi(\phi(V_r)))$ is continuously increasing in $[0, 1]$. Further, $\Phi(\phi(V_r)) \geq 0$ for any $V_r \in [0, 1]$. An induction argument shows that $\{V_r^t\}$ monotonically converges if and only if

$$\Phi(\phi(V_r^1)) \leq V_r^1, \tag{52}$$

which holds since $V_r^1 = 1$, $\phi(1) \geq 0$ (see Lemma 19) and $\Phi(v) \leq 1$ for all $v \geq 0$. Further, if $\phi(1) \neq \infty$, then the sequence $\{V_r^t\}$ converges to $V_r^\star$, where $V_r^\star$ is the smallest $v$ so that the following holds for all $V_r \in [v, 1]$, i.e.,

$$V_r^\star = \inf\left\{v \in [0, 1] \,:\, \Phi\left(\phi(v_r)\right) < v_r, \forall v_r \in [v, 1]\right\}.$$

Substituting in the definitions of $\phi$ and $\Phi$ in (7), it is straightforward to show that the above definition of $V_r^\star$ is equivalent to that in (10).

For the degenerate case where $\phi(1) = \infty$ (which corresponds to $\mathsf{mmse}_z(1) = 1$ and happens when $f$ is an even function), $P(1) = 0$ and so $V_r^\star$ in (10)) is not defined. Lemma 3 holds by defining $V_r^\star = 1$ for this degenerate case.

### D.2  Proof of Lemma 4

Clearly, $\mathsf{MSE}_\lambda^\star \triangleq \mathsf{MSE}_\lambda(\phi(V_r^\star)) = 0$ (see definition in Lemma 3) if and only if $\phi(V_r^\star) = 0$. When $f$ is not an invertible function, $\phi$ is continuous and strictly increasing (see Lemma 19), and so $\phi(V_r^\star) = 0$ if and only if $\phi(0) \triangleq \lim_{v_r \to 0} \phi(v_r) = 0$ and $V_r^\star = 0$. Therefore, to prove the lemma, it remains to prove $\phi(0) = 0$ and $V_r^\star = 0$ hold if and only if (12) is satisfied.

*Necessity:* It is straightforward to see that $V_r^\star = 0$ is equivalent to (12). Hence, (12) is a necessary condition.

*Sufficiency:* (12) already implies $V_r^\star = 0$. To show the sufficiency of (12), we only need to prove that (12) implies $\phi(0) = 0$.

Due to the continuity of $\phi$, if (12) holds, then we must have

$$P(0) \triangleq \lim_{v_r \to 0} P(v_r) \geq 0, \tag{53}$$

otherwise there will exist a neighbor of $v_r = 0$ where (12) is violated. From (11), we have

$$
\begin{aligned}
P(0) &= \mathbb{E}\left[\frac{\phi(0)}{\phi(0) + \lambda}\right] - 1 + \delta \cdot \left[1 - \lim_{v_r \to 0} \frac{\mathsf{mmse}_z(v_r)}{v_r}\right] \\
&\overset{(a)}{=} \mathbb{E}\left[\frac{\phi(0)}{\phi(0) + \lambda}\right] - 1 + \delta \cdot \left[1 - \lim_{v_r \to 0} \frac{\mathsf{mmse}(Z, v_r^{-1} - 1 | f(Z))}{v_r}\right] \\
&= \mathbb{E}\left[\frac{\phi(0)}{\phi(0) + \lambda}\right] - 1 + \delta \cdot \left[1 - \lim_{\mathsf{snr}_{\mathrm{eff}} \to \infty} (\mathsf{snr}_{\mathrm{eff}} + 1) \cdot \mathsf{mmse}(Z, \mathsf{snr}_{\mathrm{eff}} | f(Z))\right] \quad (\mathsf{snr}_{\mathrm{eff}} := v_r^{-1} - 1) \\
&\overset{(b)}{=} \mathbb{E}\left[\frac{\phi(0)}{\phi(0) + \lambda}\right] - 1 + \delta \cdot \left[1 - \mathscr{D}(Z | f(Z))\right],
\end{aligned}
$$

where step (a) is from the definition of $\mathsf{mmse}_z$ below (7), and step (b) is from definition of $\mathscr{D}(Z | f(Z))$ exists and the fact that $\mathsf{mmse}_z(0) = 0$. Since $0 \leq \mathscr{D}(Z | f(Z)) \leq 1$, we consider the cases $\mathscr{D}(Z | f(Z)) = 1$ and $\mathscr{D}(Z | f(Z)) < 1$ separately. When $\mathscr{D}(Z | f(Z)) = 1$, we have

$$
P(0) = \mathbb{E}\left[\frac{\phi(0)}{\phi(0) + \lambda}\right] - 1 < 0,
$$

violating (53), which is a necessary condition for (12). Hence, if (12) holds, then we must have $\mathscr{D}(Z | f(Z)) < 1$. Further, when $\mathscr{D}(Z | f(Z)) < 1$, Lemma 19 shows that $\phi(0) = 0$. Hence, (12) implies $\phi(0) = 0$, and so perfect reconstruction. This completes the proof of the sufficiency of (12).

We have shown that $P(0) \geq 0$ is a necessary condition for perfect reconstruction. It is straightforward to see that $P(0) \geq 0$ is equivalent to

$$
\delta \geq \frac{1}{1 - \mathscr{D}(Z | f(Z))} = \frac{1}{d(Y)} \tag{54}
$$

where the second equality is from Lemma 16.

## E  Proof of Lemma 6

Lemma 3 shows that the MSE of the GLM-EP algorithm is given by

$$
\mathsf{MSE}_\lambda^\star \triangleq \mathbb{E}\left[\frac{\phi(v_r^\star)}{\phi(v_r^\star) + \lambda}\right], \tag{55}
$$

where

$$
v_\lambda^\star = \inf\left\{v \in [0, 1] : \mathbb{E}\left[\frac{\phi(v_r)}{\phi(v_r) + \lambda}\right] > g(v_r), \forall v_r \in [v, 1]\right\}. \tag{56}
$$

Since $\mathbb{E}\left[\frac{\phi(v_r)}{\phi(v_r) + \lambda}\right] \geq 0$, $v_\lambda^\star$ can be equivalently defined as

$$
v_\lambda^\star = \inf\left\{v \in [0, 1] : \mathbb{E}\left[\frac{\phi(v_r)}{\phi(v_r) + \lambda}\right] > G(v_r), \forall v_r \in [v, 1]\right\}. \tag{57}
$$

where $G(v_r)$ is defined in (14). We next prove that $v_r^\star$ must satisfy

$$
\mathsf{MSE}_\lambda^\star \triangleq \mathbb{E}\left[\frac{\phi(v_r^\star)}{\phi(v_r^\star) + \lambda}\right] = G(v_r^\star). \tag{58}
$$

Eq. (52) implies $\mathbb{E}\left[\frac{\phi(1)}{\phi(1) + \lambda}\right] \geq g(1)$. Further, $\phi(1) \geq 0$, and thus $\mathbb{E}\left[\frac{\phi(1)}{\phi(1) + \lambda}\right] \geq 0$. Together, we have $\mathbb{E}\left[\frac{\phi(1)}{\phi(1) + \lambda}\right] \geq G(1)$. The only possibility (58) does not hold is when

$$
\mathbb{E}\left[\frac{\phi(v_r)}{\phi(v_r) + \lambda}\right] > G(v_r) \quad \forall v_r \in [0, 1]. \tag{59}
$$

We next show that (59) cannot hold. We only need to prove (59) cannot hold for $v_r = 0$. We consider two case $\mathscr{D}(Z | f(Z)) < 1$ and $\mathscr{D}(Z | f(Z)) = 1$ separately. When $\mathscr{D}(Z | f(Z)) < 1$, Lemma 19

guarantees $\phi(0) = 0$, and thus $\mathbb{E}\left[\frac{\phi(0)}{\phi(0)+\lambda}\right] = 0 \le G(0)$, where the inequality is from the definition of $G(\cdot)$. When $\mathscr{D}(Z|f(Z)) = 1$, we have

$$\lim_{v_r \to 0} 1 - \delta \cdot \left[1 - \frac{\mathsf{mmse}_z(v_r)}{v_r}\right] = 1 - \delta\left[1 - \mathscr{D}(Z|f(Z))\right] = 1.$$

Hence, from (14), we have $G(0) = 1$. On the other hand, $\mathbb{E}\left[\frac{\phi(0)}{\phi(0)+\lambda}\right] \le 1$ since $\phi(0) \ge 0$. Hence, (59) cannot hold at $v_r = 0$. Combining the previous arguments proves (58).

At this point, we can compare $\mathsf{MSE}^\star_{\lambda_1}$ and $\mathsf{MSE}^\star_{\lambda_2}$. Note that $C/(C + \lambda)^{-1}$ is a convex function of $\lambda$ for every $C > 0$. Hence, Lemma 5 implies that the following holds for all $\gamma_l > 0$:

$$\lambda_1 \succeq_L \lambda_2 \implies \mathbb{E}\left[\frac{\phi(v_r)}{\phi(v_r)+\lambda_1}\right] \ge \mathbb{E}\left[\frac{\phi(v_r)}{\phi(v_r)+\lambda_2}\right], \quad \forall \phi(v_r) \ge 0,$$

where $\succeq_L$ means spikier in the Lorenz sense (see Definition 3). From the definition of $v^\star_\lambda$, we have

$$\lambda_1 \succeq_L \lambda_2 \implies v^\star_{\lambda_1} \le v^\star_{\lambda_2}. \tag{60}$$

To compare $\mathsf{MSE}^\star_{\lambda_1}$ and $\mathsf{MSE}^\star_{\lambda_2}$, it is not very convenient to directly use (55) since the expectation in (55) itself depends on the distribution of $\lambda$. Instead, due to (58), we only need to compare $G(v^\star_{\lambda_1})$ and $G(v^\star_{\lambda_2})$. Since $v^\star_{\lambda_1} \le v^\star_{\lambda_2}$, the claims in the lemma follow directly.

# F   Proof of Theorem 3

From Lemma 3, the GLM-EP algorithm cannot achieve perfect recovery if $\mathscr{D}(Z|f(Z)) = 1$. Therefore, we will only consider the case $\mathscr{D}(Z|f(Z)) < 1$. In this case, we have $\phi(0) = 0$ (see Lemma 19).

*Part (i):* When $\phi(0) = 0$, we have $\mathsf{MSE}^\star_\lambda = 0$ if and only if $V^\star_\lambda = 0$, where $V^\star_\lambda$ is defined in (57). Therefore, we can equivalently define $\delta^\mathsf{p}_\lambda$ as

$$\delta^\mathsf{p}_\lambda = \inf\left\{\delta \,:\, v^\star_\lambda = 0\right\}. \tag{61}$$

We have proved in (60) that if $\lambda_1 \succeq_L \lambda_2$, then $V^\star_{\lambda_1} \le v^\star_{\lambda_2}$ and hence $\delta^\mathsf{p}_{\lambda_1} \le \delta^\mathsf{p}_{\lambda_2}$ (from (61)).

*Part (ii):* We assume $f$ is not an invertible function, otherwise perfect recovery is trivial. Suppose $\delta > \delta^\mathsf{p}_\star = 1/(1 - \mathscr{D}(Z|f(Z)))$. Our aim is to show that there exists a eigenvalue distribution under which GLM-EP can achieve perfect recovery. This boils down to proving (see Lemma 4):

$$\mathbb{E}\left[\frac{\phi(v_r)}{\phi(v_r)+\lambda}\right] > g(v_r), \quad \forall v_r \in (0, 1]. \tag{62}$$

We first prove

$$\sup_{v_r \in (0,1]} g(v_r) < 1. \tag{63}$$

where $g(v_r)$ is defined by (see (11))

$$g(v_r) \triangleq 1 - \delta\left(1 - \frac{\mathsf{mmse}_z(v_r)}{v_r}\right).$$

As shown in (106) (Appendix H), $\mathsf{mmse}_z(v_r) \le v_r$ for all $v_r \in (0, 1)$. Further, the inequality is strict when $f(Z)$ and $Z$ are not independent. This holds when $f$ is not a constant function, which is guaranteed by the assumption $d(f(Z)) \ne 0$. Therefore,

$$g(v_r) < 1, \quad \forall v_r \in (0, 1).$$

Further, when $v_r = 1$, $\mathsf{mmse}_z(1) < 1$ (by assumption) and so $g(1) < 1$. When $v_r \to 0$,

$$g(0) \triangleq \lim_{v_r \to 0_+} g(v_r) = \lim_{v_r \to 0} 1 - \delta\left(1 - \frac{\mathsf{mmse}_z(v_r)}{v_r}\right) = 1 - \delta(1 - \mathscr{D}(Z|f(Z))) < 0, \tag{64}$$

where $\delta > \delta_\star^{\text{p}} = 1/(1 - \mathscr{D}(Z|f(Z)))$. Combining the above facts (together with the continuity of $g$) proves (63).

To prove (62), consider the following two-point distribution parameterized by $P \in (0, 1)$ and $a \in (0, \delta)$:

$$P_\lambda = \begin{cases} a & \text{with prob. } P \\ \frac{\delta - aP}{1-P} & \text{with prob. } 1 - P. \end{cases} \tag{65}$$

This distribution satisfies the normalization assumption $\mathbb{E}[\lambda] = \delta$. Under this distribution, the left-hand side of (62) becomes

$$\mathbb{E}\left[\frac{\phi(v_r)}{\phi(v_r) + \lambda}\right] = P \cdot \frac{\phi(v_r)}{\phi(v_r) + a} + (1 - P) \cdot \frac{\phi(v_r)}{\phi(v_r) + b} \quad \left(b \triangleq \frac{\delta - aP}{1-P}\right)$$
$$> P \cdot \frac{\phi(v_r)}{\phi(v_r) + a} \quad (b > 0). \tag{66}$$

We next show that there exists $a \in (0, \delta)$ and $P \in (0, 1)$ for which the following holds,

$$P \cdot \frac{\phi(v_r)}{\phi(v_r) + a} > g(v_r). \quad \forall v_r \in (0, 1].$$

Since $\phi(v_r)$ is non-negative (see Lemma 19), It suffices to prove

$$a < \phi(v_r) \cdot \left(\frac{P}{g(v_r)} - 1\right), \quad \forall v_r \in \mathbb{D} \triangleq \{v_r \in (0, 1] : g(v_r) \geq 0\}. \tag{67}$$

Consider an arbitrary $P \in (\sup_{v \in \mathbb{D}} g(v), 1)$. Due to (63), this choice of $P$ is valid.

Let

$$a_{\min}(P) \triangleq \inf_{v_r \in \mathbb{D}} \phi(v_r) \cdot \left(\frac{P}{g(v_r)} - 1\right).$$

We conclude our proof by showing $a_{\min}(P) > 0$ for $P \in (\sup_{v \in \mathbb{D}} g(v), 1)$, and setting $a \in (0, \min(a_{\min}(P), \delta))$. To this end, we note

$$\inf_{v_r \in \mathbb{D}} \phi(v_r) > 0, \tag{68a}$$

and

$$\inf_{v_r \in \mathbb{D}} \left(\frac{P}{g(v_r)} - 1\right) > 0. \tag{68b}$$

Eq. (68a) is due to the following facts: (i) $\phi(v_r) > 0$ for all $v_r \neq 0$ when $f$ is not invertible (see Lemma 19); and (ii) $\mathbb{D} \triangleq \{v_r \in (0, 1] : g(v_r) \geq 0\} \subset (\hat{v}, 1]$ for some $\hat{v} > 0$. (Since $g(0) < 0$ and $g$ is continuous.) Eq. (68b) is due to the definition $P \in (\sup_{v \in \mathbb{D}} g(v), 1)$.

# G   Proof of Lemma 7

We first present the major part of the proof. Some auxiliary results are postponed to Section G.2. Throughout this appendix, we assume $\delta > \delta_\lambda^{\text{p}} \geq \delta_{\text{opt}}^{\text{p}} = 1/d(Y)$, where the second inequality is a consequence of the necessary condition for perfect reconstruction given in Lemma 4.

## G.1   Main Proof

Let $g(v_r, \sigma_w^2)$ and $P(v_r, \sigma_w^2)$ be the noisy counterparts of $g(v_r)$ and $P(v_r)$, respectively:

$$g(v_r, \sigma_w^2) \triangleq 1 - \delta\left(1 - \frac{\text{mmse}_z(v_r, \sigma_w^2)}{v_r}\right), \tag{69a}$$

$$P(v_r, \sigma_w^2) \triangleq \mathbb{E}\left[\frac{\phi(v_r, \sigma_w^2)}{\phi(v_r, \sigma_w^2) + \lambda}\right] - g(v_r, \sigma_w^2), \tag{69b}$$

where $\text{mmse}_z(v_r, \sigma_w^2)$ and $\phi(v_r, \sigma_w^2)$ are defined in (109) and (110), respectively.

The behaviors of $g$ and $P$ around $v_r = 0$ are different under the noiseless and noisy settings. Specifically,

$$\lim_{v_r \to 0} g(v_r, \sigma_w^2) = \begin{cases} 1 - \delta\left(1 - \mathscr{D}(Z|f(Z))\right) < 0 & \text{if } \sigma_w^2 = 0, \\ 1 & \text{if } \sigma_w^2 \neq 0, \end{cases}$$

which is from the definition of conditional MMSE dimension and the fact that the distribution $P_{Z|Y_\sigma}$ (where $Y_\sigma \triangleq f(Z + \sigma_w W)$) is absolutely continuous when $\sigma_w^2 \neq 0$. Further, from Lemma 21,

$$0 < \lim_{v_r \to 0} \phi(v_r, \sigma_w^2) < \infty.$$

Hence,

$$\lim_{v_r \to 0} P(v_r, \sigma_w^2) = \begin{cases} \delta\left(1 - \mathscr{D}(Z|f(Z))\right) - 1 > 0 & \text{if } \sigma_w^2 = 0, \\ \mathbb{E}\left[\frac{\sigma_w^2}{\sigma_w^2 + \lambda}\right] - 1 < 0 & \text{if } \sigma_w^2 \neq 0, \end{cases}$$

Since $g(0) < 0$ (where $g(v_r)$ is a shorthand for $g(v_r, 0)$), there exists a neighbor of $v_r = 0$ for which $g(v_r) < 0$. Define

$$v^\diamond \triangleq \inf\{v \in [0, 1] : g(v_r) = 0\}. \tag{70}$$

If $g(v_r) > 0$ for all $v_r \in [0, 1]$, we set $v^\diamond = 1$. Note that $P$ and $g$ are continuous functions of $\sigma_w^2 \geq 0$ whenever $v_r \neq 0$. Let $\epsilon \in (0, v^\diamond)$ be an arbitrary constant. By continuity, for sufficiently small $\sigma_w^2$, we have

$$P(v_r, \sigma_w^2) > 0, \quad \forall v_r \in (\epsilon, 1), \tag{71}$$

and

$$g(\epsilon, \sigma_w^2) < 0. \tag{72}$$

Since $g(0, \sigma_w^2) = 1$, $g(v_r, \sigma_w^2) = 0$ has at least one solution in $v_r \in (0, \epsilon)$. Let $v_\epsilon^\diamond(\sigma_w^2)$ be the largest one, i.e.,

$$v_\epsilon^\diamond(\sigma_w^2) \triangleq \sup\left\{v \in (0, \epsilon) : g(v_r, \sigma_w^2) = 0\right\}. \tag{73}$$

This definition ensures $g(v_r, \sigma_w^2) < 0, \forall v_r \in (v_\epsilon^\diamond(\sigma_w^2), \epsilon)$ (see (72)). This further ensures $P(v_r, \sigma_w^2) > 0$ for $v_r \in (v_\epsilon(\sigma_w^2), \epsilon)$, since the first term in (69b) is positive. Together with (71), we have

$$P(v_r, \sigma_w^2) > 0 \quad \forall v_r \in (v_\epsilon^\diamond(\sigma_w^2), 1). \tag{74}$$

Now, let us define

$$v_r^\star(\sigma_w^2) = \sup\left\{v \in [0, 1] : P(v_r, \sigma_w^2) = 0\right\}, \tag{75}$$

which is the fixed point reached by the state evolution. As a consequence of (74) and (75), we have (for small enough $\sigma_w^2$)

$$v_r^\star(\sigma_w^2) \leq v_\epsilon^\diamond(\sigma_w^2).$$

By the monotonicity of $\phi(v_r, \sigma_w^2)$ with respect to $v_r$ (see Lemma 21), we have the following for small $\sigma_w^2$

$$\phi(v_r^\star(\sigma_w^2), \sigma_w^2) \leq \phi(v_\epsilon^\diamond(\sigma_w^2), \sigma_w^2)$$
$$\overset{(a)}{=} (\delta - 1) \cdot v_\epsilon^\diamond(\sigma_w^2) \tag{76}$$
$$\overset{(b)}{\leq} (\delta - 1) \cdot C(\delta, f) \cdot \sigma_w^2,$$

where step (a) follows from (69a) and the fact that $v_\epsilon^\diamond(\sigma_w^2)$ is a solution to $g(v_r, \sigma_w^2) = 0$, and step (b) is due to Lemma 14. Together with Lemma 21, we finally have

$$\sigma_w^2 \leq \phi(v^\star(\sigma_w^2), \sigma_w^2) \leq (\delta - 1) \cdot C(\delta, f) \cdot \sigma_w^2. \tag{77}$$

In Section G.2, we will prove that (see Lemma 14) the following holds for small $\sigma_w^2$:

$$v_\epsilon^\diamond(\sigma_w^2) \leq C(\delta, f) \cdot \sigma_w^2, \tag{78}$$

where $C(\delta, f)$ is some constant depending on $\delta$ and the nonlinear measure function $f$.

Finally, for small $\sigma_w^2$, the MSE is given by

$$\mathsf{MSE}_\lambda^\star(\sigma_w^2, \lambda) = \mathbb{E}\left[\frac{\phi(v_r^\star(\sigma_w^2), \sigma_w^2)}{\phi(v_r^\star(\sigma_w^2), \sigma_w^2) + \lambda}\right]$$
$$= \phi\left(v^\star(\sigma_w^2), \sigma_w^2\right) \cdot \left(\mathbb{E}[\lambda^{-1}] + o(1)\right).$$

From (77), we have

$$\sigma_w^2 \cdot \left(\mathbb{E}[\lambda^{-1}] + o(1)\right) \leq \mathsf{MSE}_\lambda^\star(\sigma_w^2, \lambda) \leq (\delta - 1) \cdot C(\delta, f) \cdot \sigma_w^2 \cdot \left(\mathbb{E}[\lambda^{-1}] + o(1)\right).$$

This completes our proof.

## G.2 Auxiliary Results

**Lemma 14.** *Suppose $\sigma_w^2 \neq 0$ and $\delta > \delta_\lambda^{\mathrm{p}} \geq 1/(1 - \mathscr{D}(Z|f(Z)))$, where $Z \sim \mathcal{N}(0,1)$. Define $v^\diamond \triangleq \inf\{v \in [0,1] : g(v_r) = 0\}$. For arbitrary $\epsilon \in (0, v^\diamond)$, define*

$$v_\epsilon^\diamond(\sigma_w^2) \triangleq \sup\left\{ v \in (0, \epsilon) : \mathsf{mmse}_z(v_r, \sigma_w^2) = \left(1 - \frac{1}{\delta}\right) v_r \right\}, \tag{79}$$

*where $\mathsf{mmse}_z(v_r, \sigma_w^2)$ is defined in (109). Then, the following holds as $\sigma_w^2 \to 0$*

$$v_\epsilon^\diamond(\sigma_w^2) \leq C(\delta, f) \cdot \sigma_w^2, \tag{80}$$

*where $0 < C(\delta, f) < \infty$ is a constant depending on $\delta$ and $f$.*

*Proof.* Our proof is mainly concerned with proving the following upper bound of $\mathsf{mmse}_z(v_r, \sigma_w^2)$ as $v_r + \sigma_w^2 \to 0$:

$$\mathsf{mmse}_z(v_r, \sigma_w^2) \leq v_r \cdot \mathscr{D}(Z|f(Z)) + o(v_r) + C \cdot \sigma_w^2, \tag{81}$$

where $C$ is some constant depending on $\delta$ and $f$. Using this, we can upper bound $v_\epsilon^\diamond(\sigma_w^2)$ by the solution to the solution to the following equation:

$$v_r \cdot \mathscr{D}(Z|f(Z)) + o(v_r) + C \cdot \sigma_w^2 = \left(1 - \frac{1}{\delta}\right) v_r. \tag{82}$$

Namely,

$$v_\epsilon^\diamond(\sigma_w^2) \leq \frac{C\sigma_w^2}{1 - \frac{1}{\delta} - \mathscr{D}(Z|f(Z)) - o(1)},$$

which yields the desired result.

The rest of this section is devoted to the proof of (81). Define

$$\begin{aligned} Y_\sigma &= f(Z + \sigma_w W), \\ U_\sigma &= Z + \sigma_w W, \\ Z_r &= (1 - v_r)Z + \sqrt{v_r(1 - v_r)}N, \\ R &= \sqrt{v_r}Z - \sqrt{1 - v_r}N, \end{aligned} \tag{83}$$

where $Z, N, W, R$ are standard Gaussian RVs, and $(Z, N, W)$ are mutually independent and $R \perp\!\!\!\perp (Z_r, W)$. (Here, $A \perp\!\!\!\perp B$ denotes $A, B$ are independent RVs.) Notice that

$$Z = Z_r + \sqrt{v_r}R.$$

Lemma 15 (at the end of this section) shows that the following holds

$$\lim_{v_r + \sigma_w^2 \to 0} \frac{1}{v_r} \left( \mathsf{mmse}(v_r, \sigma_w^2) - \mathsf{mmse}_{\mathrm{app}}(v_r, \sigma_w^2) \right) = 0.$$

As a consequence,

$$\mathsf{mmse}_z(v_r, \sigma_w^2) = \mathsf{mmse}_{\mathrm{app}}(v_r, \sigma_w^2) + o(1) \cdot v_r. \tag{84}$$

In what follows, we prove that the following holds for all $v_r \in (0, 1)$

$$\mathsf{mmse}_{\mathrm{app}}(v_r, \sigma_w^2) = \mathsf{mmse}_{\mathrm{app}}(v_r, 0) + O(\sigma_w^2). \tag{85}$$

We first recall that $\mathsf{mmse}_{\mathrm{app}}$ is defined as

$$\mathsf{mmse}_{\mathrm{app}}(v_r, \sigma_w^2) \triangleq \underbrace{v_r \mathbb{E}\left( \left( \frac{\sigma_w^2 R}{v_r + \sigma_w^2} - \frac{\sqrt{v}\sigma_w W}{v_r + \sigma_w^2} \right)^2 \mathbb{I}(\mathcal{E}_1) \right)}_{\text{Part 1}} + \underbrace{v_r \mathbb{E}\left( R^2 \mathbb{I}(\mathcal{E}_1^c) \right)}_{\text{Part 2}} \tag{86}$$

Clearly, Part one is $O(\sigma_w^2)$. We next show that the difference between Part two and $\mathsf{mmse}_{\mathrm{app}}(v_r, 0)$ is $O(\sigma_w^2)$. To this end, notice that $R$ is correlated with $U_\sigma$, and it is convenient to decompose it as

$$R = \frac{\sqrt{v_r}}{1 + \sigma_w^2} U_\sigma + \sqrt{\frac{1 - v_r + \sigma_w^2}{1 + \sigma_w^2}} S,$$

where $S \sim \mathcal{N}(0,1)$ and $S \perp\!\!\!\perp U_\sigma$. Then,

$$\text{Part } 2 = v_r \, \mathbb{E}\left(R^2 \mathbb{I}(\mathcal{E}_1^c)\right)$$

$$= v_r \, \mathbb{E}\left(\left(\frac{\sqrt{v_r}}{1+\sigma_w^2}U_\sigma + \sqrt{\frac{1-v_r+\sigma_w^2}{1+\sigma_w^2}}S\right)^2 \mathbb{I}(\mathcal{E}_1^c)\right)$$

$$= \frac{v_r^2}{1+\sigma_w^2}\mathbb{E}\left(U_\sigma^2 \mathbb{I}(\mathcal{E}_1^c)\right) + \frac{v_r(1-v_r+\sigma_w^2)}{1+\sigma_w^2}\cdot \mathbb{P}(\mathcal{E}_1^c)$$

We notice the following facts: (i) $U_\sigma = Z + \sigma_w W$; (ii) $\mathcal{E}_1^c = \mathbb{I}(U_\sigma \in \{x : f^{-1}(f(x)) \text{ is an interval}\})$. It can be shown that there exists a constant $C < \infty$ such that the following hold for all $v_r \in (0,1)$

$$\mathbb{E}\left(U_\sigma^2 \mathbb{I}(\mathcal{E}_1^c)\right) \leq \mathbb{E}\left(U_\sigma^2 \mathbb{I}(\mathcal{E}_1^c)\right)\big|_{\sigma_w=0} + C \cdot \sigma_w^2,$$

$$\mathbb{P}(\mathcal{E}_1^c) \leq \mathbb{P}(\mathcal{E}_1^c)\big|_{\sigma_w=0} + C \cdot \sigma_w^2,$$

as $\sigma_w^2 \to 0$. We skip the details here. Combining the above arguments proves (85).

Finally, combining (84) and (85), we have

$$\mathsf{mmse}_z(v_r, \sigma_w^2) = \mathsf{mmse}_{\text{app}}(v_r, \sigma_w = 0) + o(1)v_r + O(\sigma_w^2),$$

as $v_r + \sigma_w^2 \to 0$. Notice that

$$\mathsf{mmse}_{\text{app}}(v_r, \sigma_w = 0) = \mathbb{E}\left(v_r Z - \sqrt{v_r(1-v_r)}N\right)^2 \mathbb{I}(\mathcal{E}_1^c)$$

where without slight abuse of notation $\mathcal{E}_1^c = \mathbb{I}(Z \in \{x : f^{-1}(f(x)) \text{ is an interval}\})$ (namely, it is the previous defined $\mathcal{E}_1^c$ at $\sigma_w = 0$). This term has the same behavior as $\mathsf{mmse}_z(v_r)$ for small $v_r$. Here, the $O(v_r)$ term is

$$v_r(1-v_r)\cdot \mathbb{E}[N^2 \mathbb{I}(\mathcal{E}_1^c)] = v_r(1-v_r)\cdot \mathscr{D}(Z|f(Z)).$$

Hence, overall we have

$$\mathsf{mmse}_z(v_r, \sigma_w^2) \leq v_r \cdot \mathscr{D}(Z|f(Z)) + o(v_r) + C \cdot \sigma_w^2,$$

as $v_r + \sigma_w^2 \to 0$. $\qquad\square$

**Lemma 15.** *Let* $\mathsf{mmse}_z(v_r, \sigma_w^2)$ *be the noisy MMSE defined in* (109)*). Define*

$$\mathsf{mmse}_{\text{app}}(v_r, \sigma_w^2) \triangleq v_r \mathbb{E}\left(\left(\frac{\sigma_w^2 R}{v_r + \sigma_w^2} - \frac{\sqrt{v}\sigma_w W}{v_r + \sigma_w^2}\right)^2 \mathbb{I}(\mathcal{E}_1)\right) + v_r \mathbb{E}\left(R^2 \mathbb{I}(\mathcal{E}_1^c)\right). \qquad (87)$$

*where*

$$\mathcal{E}_1 \triangleq \{U_\sigma \in \mathbb{R}\backslash \mathcal{Q}_f\}, \qquad (88)$$

$U_\sigma = Z + \sigma_w N$ *and* $\mathcal{E}_1^c$ *is the complement of* $\mathcal{E}_1$*. Then, the following holds*

$$\lim_{v_r + \sigma_w^2 \to 0} \frac{1}{v_r}\cdot \left(\mathsf{mmse}_z(v_r, \sigma_w^2) - \mathsf{mmse}_{\text{app}}(v_r, \sigma_w^2)\right) = 0. \qquad (89)$$

*Proof.* By the definitions of $\mathsf{mmse}_z$ and $\mathsf{mmse}_{\text{app}}$, we have

$$\frac{1}{v_r}\left(\mathsf{mmse}_z(v_r, \sigma_w^2) - \mathsf{mmse}_{\text{app}}(v_r, \sigma_w^2)\right)$$

$$= \mathbb{E}\left[\frac{1}{v_r}\left(Z - \mathbb{E}[Z|Y_\sigma, Z_r]\right)^2 - \left(\frac{\sigma_w^2 R}{v_r + \sigma_w^2} - \frac{\sqrt{v_r}\sigma_w W}{v_r + \sigma_w^2}\right)^2 \mathbb{I}(\mathcal{E}_1) - R^2 \mathbb{I}(\mathcal{E}_1^c)\right].$$

We bound the term inside the expectation by

$$\left| \frac{1}{v_r}\left(Z - \mathbb{E}[Z|Y_\sigma, Z_r]\right)^2 - \left(\frac{\sigma_w^2 R}{v_r + \sigma_w^2} - \frac{\sqrt{v_r}\sigma_w W}{v_r + \sigma_w^2}\right)^2 \mathbb{I}(\mathcal{E}_1) - R^2\mathbb{I}(\mathcal{E}_1^c)\right|$$

$$\stackrel{(a)}{=} \left|\left(R - \mathbb{E}[R|Y_\sigma, Z_r]\right)^2 - \left(\frac{\sigma_w^2 R}{v_r + \sigma_w^2} - \frac{\sqrt{v_r}\sigma_w W}{v_r + \sigma_w^2}\right)^2 \mathbb{I}(\mathcal{E}_1) - R^2\mathbb{I}(\mathcal{E}_1^c)\right| \tag{90}$$

$$\leq 2(R^2 + \mathbb{E}^2[R|Y_\sigma, Z_r]) + \left(\frac{\sigma_w^2 R}{v_r + \sigma_w^2} - \frac{\sqrt{v_r}\sigma_w W}{v_r + \sigma_w^2}\right)^2 \mathbb{I}(\mathcal{E}_1) + R^2\mathbb{I}(\mathcal{E}_1^c)$$

$$\leq 2(R^2 + \mathbb{E}^2[R|Y_\sigma, Z_r]) + 2\left(R^2 + \frac{1}{4}W^2\right) + R^2$$

where step (a) follows from the definition $Z = Z_r + \sqrt{v_r}R$. Since

$$\mathbb{E}\left[2(R^2 + \mathbb{E}^2[R|Y_\sigma, Z_r]) + \left(R^2 + \frac{1}{4}W^2\right) + R^2\right] < \infty,$$

by dominated convergence theorem we have

$$\lim_{v_r + \sigma_w^2 \to 0} \frac{1}{v_r}\mathsf{mmse}_z(v_r, \sigma_w^2) - \mathsf{mmse}_{\mathrm{app}}(v_r, \sigma_w^2)$$

$$= \lim_{v_r + \sigma_w^2 \to 0} \frac{1}{v_r}\mathbb{E}\left(Z - \mathbb{E}[Z|Y_\sigma, Z_r]\right)^2 - \mathsf{mmse}_{\mathrm{app}}(v_r, \sigma_w^2)$$

$$= \mathbb{E}\left[\lim_{v_r + \sigma_w^2 \to 0} \frac{1}{v_r}\left(Z - \mathbb{E}[Z|Y_\sigma, Z_r]\right)^2 - \left(\frac{\sigma_w^2 R}{v_r + \sigma_w^2} - \frac{\sqrt{v}\sigma_w W}{v_r + \sigma_w^2}\right)^2 \mathbb{I}(\mathcal{E}_1) - R^2\mathbb{I}(\mathcal{E}_1^c)\right]$$

$$= \mathbb{E}[T_1] + \mathbb{E}[T_2],$$

where

$$T_1 \triangleq \lim_{v_r + \sigma_w^2 \to 0} \frac{1}{v_r}\left(Z - \mathbb{E}[Z|Y_\sigma, Z_r]\right)^2 \mathbb{I}(\mathcal{E}_1) - \left(\frac{\sigma_w^2 R}{v_r + \sigma_w^2} - \frac{\sqrt{v}\sigma_w W}{v_r + \sigma_w^2}\right)^2 \mathbb{I}(\mathcal{E}_1)$$

$$T_2 \triangleq \lim_{v_r + \sigma_w^2 \to 0} \frac{1}{v_r}\left(Z - \mathbb{E}[Z|Y_\sigma, Z_r]\right)^2 \mathbb{I}(\mathcal{E}_1^c) - R^2\mathbb{I}(\mathcal{E}_1^c). \tag{91}$$

We next prove $\mathbb{E}[T_1] = 0$ and $\mathbb{E}[T_2] = 0$ separately.

*Analysis of $T_1$:* Direct calculations yield

$$\mathbb{E}[Z|Y_\sigma = y, Z_r = z_r] = \frac{\int_{f^{-1}(y)} \mathcal{N}(u; z_r, v_r + \sigma_w^2)\frac{v_r u + \sigma_w^2 z_r}{v_r + \sigma_w^2}\mathrm{d}u}{\int_{f^{-1}(y)} \mathcal{N}(u; z_r, v_r + \sigma_w^2)\mathrm{d}u} \tag{92a}$$

$$= z_r + \frac{v_r}{v_r + \sigma_w^2}\frac{\int_{\mathcal{I}} u\mathcal{N}(u; 0, v_r + \sigma_w^2)\mathrm{d}u}{\int_{\mathcal{I}} \mathcal{N}(u; 0, v_r + \sigma_w^2)\mathrm{d}u}, \tag{92b}$$

where $\mathcal{N}(x; m, v) \triangleq \frac{1}{\sqrt{2\pi v}}\exp\left(-\frac{(x-m)^2}{2v}\right)$, $\mathcal{I} \triangleq f^{-1}(y) - z_r$, and the second step is due to a change of variable. We emphasize that $\mathcal{I}$ is indexed by $y$ and $z_r$, but to make notation light we did not make such dependency explicit. When $f^{-1}(y)$ is a discrete set, the integration is simply replaced by a summation.

With slight abuse of notations, let $(z, w, n, y, z_r)$ be an instance of $(Z, W, N, Y_\sigma, Z_r)$. From (83), we have $z_r = (1 - v_r)z + \sqrt{v_r(1 - v_r)}n$ and $y = f(z + \sigma_w w)$. Then,

$$
\begin{aligned}
\frac{1}{v_r}\left(z - \mathbb{E}[Z|Y_\sigma = y, Z_r = z_r]\right)^2 &= \frac{1}{v_r}\left(z - z_r - \frac{v_r}{v_r + \sigma_w^2}\frac{\int_\mathcal{I} u\mathcal{N}(u; 0, v_r + \sigma_w^2)\mathrm{d}u}{\int_\mathcal{I} \mathcal{N}(u; 0, v_r + \sigma_w^2)\mathrm{d}u}\right)^2 \\
&= \frac{1}{v_r}\left(v_r z - \sqrt{v_r(1 - v_r)}n - \frac{v_r}{v_r + \sigma_w^2}\frac{\int_\mathcal{I} u\mathcal{N}(u; 0, v_r + \sigma_w^2)\mathrm{d}u}{\int_\mathcal{I} \mathcal{N}(u; 0, v_r + \sigma_w^2)\mathrm{d}u}\right)^2 \\
&= \left(\sqrt{v_r}z - \sqrt{1 - v_r}n - \frac{\sqrt{v_r}}{v_r + \sigma_w^2}\frac{\int_\mathcal{I} u\mathcal{N}(u; 0, v_r + \sigma_w^2)\mathrm{d}u}{\int_\mathcal{I} \mathcal{N}(u; 0, v_r + \sigma_w^2)\mathrm{d}u}\right)^2 \\
&= \left(r - \frac{\sqrt{v_r}}{v_r + \sigma_w^2}\frac{\int_\mathcal{I} u\mathcal{N}(u; 0, v_r + \sigma_w^2)\mathrm{d}u}{\int_\mathcal{I} \mathcal{N}(u; 0, v_r + \sigma_w^2)\mathrm{d}u}\right)^2,
\end{aligned}
\tag{93}
$$

where the last step is due to the definition of the r.v. $R$ in (83). Recall that $\mathcal{E}_1 = \{Z + \sigma_\sigma W \in \mathbb{R} \backslash \mathcal{Q}_f\}$, where $\mathcal{Q}_f = \{z : f^{-1}(f(z))$ contains an interval$\}$. Conditioned on $\mathcal{E}_1$, $f^{-1}(y)$ is a discrete set, and so is $\mathcal{I} \triangleq f^{-1}(y) - z_r$. Hence, conditioned on $\mathcal{E}_1$, the integration in the above formula is replaced by summation over the elements in $\mathcal{I}$. Since $y = f(z + \sigma_w w)$, we have $z + \sigma_w w \in f^{-1}(y)$. Further, $z = z_r + \sqrt{v_r}r$, and thus

$$z + \sigma_w w - z_r = \sqrt{v_r}r + \sigma_w w \in f^{-1}(y) - z_r = \mathcal{I}.$$

Let $\mathcal{E}_2$ be the event that there does not exist $x \in f^{-1}(y)$ and $x \neq z + \sigma_w w$ such that $|z + \sigma_w w| = |x - z_r|$. Then, on the event $\mathcal{E}_1 \cap \mathcal{E}_2$,

$$\lim_{v_r + \sigma_w^2 \to 0} \frac{\int_\mathcal{I} u\mathcal{N}(u; 0, v_r + \sigma_w^2)\mathrm{d}u}{\int_\mathcal{I} \mathcal{N}(u; 0, v_r + \sigma_w^2)\mathrm{d}u} - (\sqrt{v_r}r + \sigma_w w) = 0.$$

This is due to the fact that $\mathcal{I}$ is a discrete set and the term with minimum exponent dominates. Hence,

$$\lim_{v_r + \sigma_w^2 \to 0} \frac{\sqrt{v_r}}{v_r + \sigma_w^2}\frac{\int_\mathcal{I} u\mathcal{N}(u; 0, v_r + \sigma_w^2)\mathrm{d}u}{\int_\mathcal{I} \mathcal{N}(u; 0, v_r + \sigma_w^2)\mathrm{d}u} - \frac{\sqrt{v_r}(\sqrt{v_r}r + \sigma_w w)}{v_r + \sigma_w^2} = 0,$$

Hence, conditioned $\mathcal{E}_1 \cap \mathcal{E}_2$, we have (see (93))

$$
\begin{aligned}
\frac{1}{v_r}\left(z - \mathbb{E}[Z|Y_\sigma = y, Z_r = z_r]\right)^2 &= \left(r - \frac{\sqrt{v_r}}{v_r + \sigma_w^2}\frac{\int_\mathcal{I} u\mathcal{N}(u; 0, v_r + \sigma_w^2)\mathrm{d}u}{\int_\mathcal{I} \mathcal{N}(u; 0, v_r + \sigma_w^2)\mathrm{d}u}\right)^2 + o(v_r + \sigma_w^2) \\
&= \left(r - \frac{\sqrt{v_r}(\sqrt{v_r}r + \sigma_w w)}{v_r + \sigma_w^2}\right)^2 + o(v_r + \sigma_w^2) \\
&= \left(\frac{\sigma_w^2 r - \sqrt{v_r}\sigma_w w}{v_r + \sigma_w^2}\right)^2 + o(v_r + \sigma_w^2)
\end{aligned}
$$

Since $\mathbb{P}(\mathcal{E}_2^c) = 0$, overall we have

$$
\begin{aligned}
\mathbb{P}(T_1 = 0) &= \mathbb{P}\left\{\lim_{v_r + \sigma_w^2 \to 0} \mathbb{I}(\mathcal{E}_1) \cdot \left[\frac{1}{v_r}(Z - \mathbb{E}[Z|Y_\sigma, Z_r])^2 - \left(\frac{\sigma_w^2 R}{v_r + \sigma_w^2} - \frac{\sqrt{v}\sigma_w W}{v_r + \sigma_w^2}\right)^2\right] = 0\right\} \\
&= 1.
\end{aligned}
$$

Hence, $\mathbb{E}[T_1] = 0$.

*Analysis of $T_2$:* Let $(z, n, w, r, y, z_r)$ be an instance of $(Z, N, W, R, Y_\sigma, Z_r)$. From (93), we have

$$
\frac{1}{v_r}\left(z - \mathbb{E}[Z|y, z_r]\right)^2 = \left(\sqrt{v_r}z - \sqrt{1 - v_r}n - \frac{\sqrt{v_r}}{v_r + \sigma_w^2}\frac{\int_\mathcal{I} u\mathcal{N}(u; 0, v_r + \sigma_w^2)\mathrm{d}u}{\int_\mathcal{I} \mathcal{N}(u; 0, v_r + \sigma_w^2)\mathrm{d}u}\right)^2 \tag{94a}
$$

$$
= \left(\sqrt{v_r}z - \sqrt{1 - v_r}n - \frac{\sqrt{v_r}}{v_r + \sigma_w^2}\frac{\int_{\hat{\mathcal{I}}} u\mathcal{N}(u; 0, 1)\mathrm{d}u}{\int_{\hat{\mathcal{I}}} \mathcal{N}(u; 0, 1)\mathrm{d}u}\right)^2 \tag{94b}
$$

where $\hat{\mathcal{I}} \triangleq \frac{\mathcal{I}}{\sqrt{v_r + \sigma_w^2}} = \frac{f^{-1}(y) - z_r}{\sqrt{v_r + \sigma_w^2}}$. Let $\mathcal{E}_3$ be the event that $z_r$ is not on the boundary of $f^{-1}(y)$. Consider the third term in (94b) under $\mathcal{E}_1^c \cap \mathcal{E}_3$. From the definition of $\mathcal{E}_1^c$, $\hat{\mathcal{I}}$ only consists of intervals. If $0$ is an interior point of $\hat{\mathcal{I}}$, we have

$$\left| \int_{\hat{\mathcal{I}}} \mathcal{N}(u; 0, 1) \mathrm{d}u - 1 \right| \leq \int_{\hat{\mathcal{I}}^c} \mathcal{N}(u; 0, 1) \mathrm{d}u = O\left( e^{-c/(v_r + \sigma_w^2))} \right).$$

where $\hat{\mathcal{I}}^c = \mathbb{R} \backslash \hat{\mathcal{I}}$ and $c > 0$ is some constant. Similarly, for the numerator,

$$\left| \int_{\hat{\mathcal{I}}} u \mathcal{N}(u; 0, 1) \mathrm{d}u \right| \leq \int_{\hat{\mathcal{I}}^c} |u| \mathcal{N}(u; 0, 1) \mathrm{d}u = O\left( e^{-c/(v_r + \sigma_w^2))} \right).$$

Hence, when $0$ is an interior point of $\hat{\mathcal{I}}$, we have

$$\lim_{v_r + \sigma_w^2 \to 0} \frac{\sqrt{v_r}}{v_r + \sigma_w^2} \frac{\int_{\hat{\mathcal{I}}} u \mathcal{N}(u; 0, 1) \mathrm{d}u}{\int_{\hat{\mathcal{I}}} \mathcal{N}(u; 0, 1) \mathrm{d}u} = 0.$$

Next, we decompose $S \triangleq (z - \mathbb{E}[Z|y, z_r])^2 / v_r$ as

$$S = S \cdot \mathbb{I}\left( 0 \in (f^{-1}(y) - z_r) \right) + S \cdot \mathbb{I}\left( 0 \notin (f^{-1}(y) - z_r) \right).$$

We note that as $v_r + \sigma_w^2 \to 0$, we have $z_r \to z$. Further, $z \in f^{-1}(y)$. Therefore,

$$\lim_{v_r + \sigma_w^2 \to 0} \mathbb{I}\left( 0 \in (f^{-1}(y) - z_r) \right) = 1.$$

We have shown in (90) that $S < \infty$. Hence,

$$\lim_{v_r + \sigma_w^2 \to 0} \frac{1}{v_r} (z - \mathbb{E}[Z|y, z_r])^2 = \lim_{v_r + \sigma_w^2 \to 0} S \cdot \mathbb{I}\left( 0 \in (f^{-1}(y) - z_r) \right) + \lim_{v_r + \sigma_w^2 \to 0} S \cdot \mathbb{I}\left( 0 \notin (f^{-1}(y) - z_r) \right)$$

$$= \lim_{v_r + \sigma_w^2 \to 0} S \cdot \mathbb{I}\left( 0 \in (f^{-1}(y) - z_r) \right)$$

$$= \left( \sqrt{v_r} z - \sqrt{1 - v_r} n \right)^2.$$

Since $\mathbb{P}(\mathcal{E}_3^c) = 0$, we have

$$\mathbb{P}(T_2 = 0) = \mathbb{P}\left( \lim_{v_r + \sigma_w^2 \to 0} \frac{1}{v_r} (z - \mathbb{E}[Z|y, z_r])^2 \, \mathbb{I}(\mathcal{E}_2) = \frac{1}{v_r} \left( v_r z - \sqrt{v_r (1 - v_r)} n \right)^2 \mathbb{I}(\mathcal{E}_2) \right)$$

$$= 1.$$

$\square$

## H  Some Auxiliary Results

In this section, after introducing the conditional MMSE dimension $\mathscr{D}(Z|Y)$, we present a few properties of $\mathsf{mmse}_z(\cdot)$ and the SE maps $\phi(\cdot)$, $\Phi(\cdot)$.

### H.1  MMSE Dimension and information dimension

The MMSE dimension $\mathscr{D}(Z)$ defined below characterizes the high SNR behavior of the MMSE $\mathsf{mmse}(Z, \mathsf{snr})$. Similarly, $\mathscr{D}(Z|U)$ characterizes the high SNR behavior of $\mathsf{mmse}(Z, \mathsf{snr}|U)$.

**Definition 6** (MMSE dimension [64]). *The following limits, if exist, is called the MMSE dimension (resp. conditional MMSE dimension):*

$$\mathscr{D}(Z) = \lim_{\mathsf{snr} \to \infty} \mathsf{snr} \cdot \mathsf{mmse}(Z, \mathsf{snr}),$$
$$\mathscr{D}(Z|U) = \lim_{\mathsf{snr} \to \infty} \mathsf{snr} \cdot \mathsf{mmse}(Z, \mathsf{snr}|U). \tag{95}$$

The following lemma establishes the connection between the conditional MMSE dimension $\mathscr{D}(Z|Y)$ and the information dimension $d(Y)$ (see Section 1.3).

**Lemma 16.** *Suppose Assumption (A.3) holds. Let $Z \sim \mathcal{N}(0,1)$ and $Y = f(Z)$. We have*

$$d(Y) = 1 - \mathscr{D}(Z|Y).$$

*Proof.* The conditional MMSE dimension can be calculated as follows:

$$
\begin{aligned}
\mathscr{D}(Z|Y) &= \lim_{\mathsf{snr}\to\infty} \mathsf{snr} \cdot \mathsf{mmse}(Z, \mathsf{snr}|Y) \\
&= \lim_{\mathsf{snr}\to\infty} \mathsf{snr} \cdot \mathbb{E}\left[ \left( Z - \mathbb{E}[Z|\sqrt{\mathsf{snr}}Z + N, Y = y] \right)^2 \right] \\
&\stackrel{a}{=} \lim_{\mathsf{snr}\to\infty} \mathsf{snr} \cdot \mathbb{E}\left[ \left( Z - \mathbb{E}[Z_y|\sqrt{\mathsf{snr}}Z_y + N] \right)^2 \right] \\
&\stackrel{\triangle}{=} \lim_{\mathsf{snr}\to\infty} \mathsf{snr} \cdot \mathbb{E}_Y\left[ \mathsf{mmse}(Z_y, \mathsf{snr}) \right]
\end{aligned}
\tag{96}
$$

where $Z_y \sim P_{Z|Y=y} = P_{Z|Z \in f^{-1}(y)}$ denotes a random variable indexed by $y$, and $N$ is independent of $Z$. Note that $\mathsf{mmse}(Z_y, \mathsf{snr}) \le \mathsf{snr}^{-1}$ [30][5] and so $\mathsf{snr} \cdot \mathsf{mmse}(Z_u, \mathsf{snr}) \le 1$. Hence, by Lebesgue's dominated convergence theorem we have

$$
\begin{aligned}
\mathscr{D}(Z|Y) &= \mathbb{E}_Y\left[ \lim_{\mathsf{snr}\to\infty} \mathsf{snr} \cdot \mathsf{mmse}(Z_y, \mathsf{snr}) \right] \\
&= \mathbb{E}_Y\left[ \mathscr{D}(Z_y) \right],
\end{aligned}
\tag{97}
$$

provided that $\lim_{\mathsf{snr}\to\infty} \mathsf{snr} \cdot \mathsf{mmse}(Z_y, \mathsf{snr})$ exists almost surely. From [64, Theorem 10 and Theorem 11],

$$
\mathscr{D}(Z_y) = \begin{cases} 0 & \text{if } P_{Z|Y=y} \text{ is discrete} \\ 1 & \text{if } P_{Z|Y=y} \text{ is absolutely continuous w.r.t. Lebesgue measure} \end{cases}
$$

This implies that

$$
\mathscr{D}(Z_y) = \begin{cases} 0 & \text{if } y \in \mathbb{R} \backslash \mathcal{Q}_f \\ 1 & \text{if } y \in \mathcal{Q}_f. \end{cases}
$$

Hence,

$$\mathscr{D}(Z|Y) = \mathbb{P}\{f(Z) \in \mathcal{Q}_f\} = 1 - d(Y).$$

$\square$

## H.2   A property of $\mathsf{mmse}_z(v_r)$

Note that $\eta_z(z_r, y, v)$ in GLM-EP (see (6f)) is an MMSE estimator:

$$\eta_z(z_r, y, v) = \mathbb{E}[Z|Y = y, Z_r = z_r] = \frac{\int_{f^{-1}(y)} u \cdot \mathcal{N}(u; z_r, v) du}{\int_{f^{-1}(y)} \mathcal{N}(u; z_r, v) du}, \tag{98}$$

where $(Z, Z_r) \sim \mathcal{N}(\mathbf{0}, \mathbf{\Sigma})$ where

$$\mathbf{\Sigma} \triangleq \begin{bmatrix} 1 & 1 - v_r \\ 1 - v_r & 1 - v_r \end{bmatrix}, \tag{99}$$

and $Y = f(Z)$. Recall that $\mathsf{mmse}_z(v_r)$ is defined as

$$\mathsf{mmse}_z(v_r) = \mathbb{E}\left( Z - \mathbb{E}[Z|Z_r, Y] \right)^2, \tag{100}$$

Lemma 17 below is a consequence of the covariance structure of $(Z, Z_r)$ defined in (99).

**Lemma 17.** *Let $\mathsf{mmse}_z(v_r)$ be the MMSE defined in (100). Let $Z \sim \mathcal{N}(0,1)$, $Y = f(Z)$ and $v_r \in (0,1]$. We have*

$$\mathsf{mmse}_z(v_r) = \mathsf{mmse}(Z, v_r^{-1} - 1|Y),$$

*where the right hand side is a conditional MMSE defined in (2).*

---

[5]This is true even when the moments of $Z_u$ do not exist. To see this, consider $\tilde{Y} = \sqrt{\mathsf{snr}}Z_u + N$ and the linear estimator $\tilde{Y}/\sqrt{\mathsf{snr}}$. The MSE of this linear estimator is $\mathsf{snr}^{-1}$ and hence $\mathsf{mmse}(Z_u, \mathsf{snr}) \le \mathsf{snr}^{-1}$.

### H.3 Properties of the SE maps

In this appendix, we discuss a few properties of the maps $\phi$ and $\Phi$ in (7):

$$\phi(v_r) = \left( \frac{1}{\mathsf{mmse}_z(v_r)} - \frac{1}{v_r} \right)^{-1}. \tag{101a}$$

$$\Phi(v_l) = \left( \frac{1}{\frac{1}{\delta} \cdot \mathbb{E}\left[ \frac{v_l \lambda}{v_l + \lambda} \right]} - \frac{1}{v_l} \right)^{-1}, \tag{101b}$$

where the expectation in $\Phi$ is over $\lambda$, which is distributed according to the asymptotic eigenvalue distribution of $\boldsymbol{A}^\mathsf{T}\boldsymbol{A}$, and

$$\mathsf{mmse}_z(v_r) \triangleq \mathsf{mmse}\left( Z, v_r^{-1} - 1 | f(Z) \right).$$

The following lemmas collect some useful properties of the MMSE [29], and the maps $\phi$, $\Phi$.

**Lemma 18** (Properties of $\mathsf{mmse}(Z, \mathsf{snr}|U)$)**.** *The following hold:*

(i) *Assume $Z \sim \mathcal{N}(0,1)$. Then, $\mathsf{mmse}(Z, \mathsf{snr}|U) \leq \frac{1}{1+\mathsf{snr}}$, $\forall \mathsf{snr} > 0$. Further, the inequality is strict if $U$ is not independent of $Z$.*

(ii) $\frac{\mathrm{d}}{\mathrm{dsnr}}\mathsf{mmse}(Z, \mathsf{snr}|U) = -\mathbb{E}\left( \mathsf{var}^2[Z|\sqrt{\mathsf{snr}}Z + N, U] \right)$, *where* $\mathsf{var}[Z|\sqrt{\mathsf{snr}}Z + N, U] \triangleq \mathbb{E}[Z^2|\sqrt{\mathsf{snr}}Z + N, U] - \mathbb{E}^2[Z|\sqrt{\mathsf{snr}}Z + N, U]$, *and $N \sim \mathcal{N}(0,1)$ is independent of $(Z, U)$.*

**Lemma 19** (Properties of $\phi$ and $\Phi$)**.** *The functions $\phi$ and $\Phi$ defined in* (101) *have the following properties:*

(i) $\phi(v_r)$ *is continuous and non-decreasing in $v_r \in (0,1)$. If $f(z)$ is not an invertible function, $\phi(v_r)$ is strictly increasing. Suppose that $f(Z)$ is not independent of $Z \sim \mathcal{N}(0,1)$. Then, $0 \leq \phi(v_r) < \infty$ and $\phi(0) = 0$ if $d(f(Z)) \neq 0$, and $\phi(1) < \infty$ if $\mathbb{E}[Z|f(Z)] \neq 0$;*

(ii) $\Phi(v_l)$ *is continuous and strictly increasing in $v_l \in (0, \infty)$. Further, $\Phi(0) = 0$ and $\Phi(\infty) = 1$.*

*Proof.* *Proof of (i):* The continuity of $\phi(v_r)$ is due to the continuity of the function $\mathsf{mmse}_z(v_r) = \mathsf{mmse}(Z, \mathsf{snr}|Y)$, where $\mathsf{snr} = v_r^{-1} - 1$ [29].

We next prove that $\phi$ is strictly increasing. Differentiation yields (see (101))

$$\phi'(v_r) = \frac{v_r^2 \cdot \mathsf{mmse}_z'(v_r) - \mathsf{mmse}_z^2(v_r)}{\left( v_r - \mathsf{mmse}_z(v_r) \right)^2}. \tag{102a}$$

Hence, we only need to prove

$$\mathsf{mmse}_z'(v_r) > \frac{1}{v_r^2} \cdot \mathsf{mmse}_z^2(v_r), \quad \forall v_r \in (0,1]. \tag{103a}$$

Recall the definition

$$\mathsf{mmse}_z(v_r) = \mathsf{mmse}(Z, \mathsf{snr}|Y), \quad \mathsf{snr} \triangleq v_r^{-1} - 1,$$

and the derivative formula of the conditional MMSE in Lemma 18, we have

$$\mathsf{mmse}_z'(v_r) = \frac{1}{v_r^2} \cdot \mathbb{E}\left( \mathsf{var}^2[Z|\sqrt{\mathsf{snr}}Z + N, Y] \right), \quad \forall v_r \in (0,1]. \tag{104}$$

Further,

$$\mathsf{mmse}_z(v_r) = \mathsf{mmse}(Z, \mathsf{snr}|Y) = \mathbb{E}\left( \mathsf{var}[Z|\sqrt{\mathsf{snr}}Z + N, Y] \right) \tag{105}$$

Combining (104) and (105), and applying Jensen's inequality proves $\mathsf{mmse}_z'(v_r) \geq \frac{1}{v_r^2} \cdot \mathsf{mmse}_z^2(v_r)$, and equality holds only when $\mathsf{var}[Z|\sqrt{\mathsf{snr}}Z + N, Y]$ is constant with respect to realizations of $\sqrt{\mathsf{snr}}Z + N$ and $Y$. This is only possible when $Z_y \sim P_{Z|Y=y}$ is Gaussian with $\mathsf{var}[Z_y]$ invariant to $y$ (including the degenerate case where $\mathsf{var}[Z_y] = 0$). Again, this is only possible when $f(z)$ is an

invertible function for which $Z_y$ is a constant and $\mathrm{var}[Z_y] = 0$). To summarize, when $f(z)$ is not an invertible function, (103a) holds and so $\phi$ is a strictly increasing function.

Finally, we verify $\phi(0)$ and $\phi(1)$. First, for any $v_r \in (0, 1)$, we have

$$
\begin{aligned}
\mathsf{mmse}_z(v_r) &= \mathsf{mmse}(Z, \mathsf{snr}|Y) \quad (\mathsf{snr} = v_r^{-1} - 1) \\
&\overset{(a)}{\leq} \mathsf{mmse}(Z, \mathsf{snr}) \\
&= \frac{1}{1 + \mathsf{snr}} \\
&= v_r
\end{aligned}
\tag{106}
$$

where step (a) is from the fact that conditioning reduces MMSE [29, Proposition 11]. Further, the inequality is strict for $v_r \neq 1$ ($\mathsf{snr} > 0$) whenever $f(Z)$ is not independent of $Z$. It follows that

$$
\phi(v_r) = \left( \frac{1}{\mathsf{mmse}_z(v_r)} - \frac{1}{v_r} \right)^{-1} \in [0, \infty), \quad \forall v_r \in (0, 1).
$$

Further, $\phi(v_r)$ is continuously increasing in $(0, 1)$ and so the limit $\lim_{v_r \to 0_+} \phi(v_r)$ exists (which is defined to be $\phi(0)$). Hence, $\phi(0) \geq 0$.

Lemma 16 shows $d(Y) = 1 - \mathscr{D}(Z|Y)$. Hence, if $d(Y) \neq 0$, we would have

$$
\mathscr{D}(Z|Y) \triangleq \lim_{\mathsf{snr} \to \infty} \mathsf{snr} \cdot \mathsf{mmse}(Z, \mathsf{snr}|Y) < 1.
$$

Then,

$$
\begin{aligned}
\phi(0) &\triangleq \lim_{v_r \to 0} \phi(v_r) \\
&= \lim_{v_r \to 0} \frac{\mathsf{mmse}_z(v_r)}{1 - \frac{\mathsf{mmse}_z(v_r)}{v_r}} \\
&\overset{(a)}{=} \lim_{\mathsf{snr} \to \infty} \frac{\mathsf{mmse}(Z, \mathsf{snr}|Y)}{1 - (\mathsf{snr} + 1)\mathsf{mmse}(Z, \mathsf{snr}|Y)} \quad (\mathsf{snr} = v_r^{-1} - 1) \\
&= 0
\end{aligned}
$$

where step (a) follows from the definition of $\mathsf{mmse}_z$ below (7), and the fact that $\lim_{\mathsf{snr} \to \infty} \mathsf{mmse}(Z, \mathsf{snr}|Y) = 0$ and $\lim_{\mathsf{snr} \to \infty} \mathsf{snr} \cdot \mathsf{mmse}(Z, \mathsf{snr}|Y) = \mathscr{D}(Z|Y) < 1$.

Finally,

$$
\begin{aligned}
\phi(1) &= \left( \frac{1}{\mathsf{mmse}_z(1)} - 1 \right)^{-1} \\
&= \left( \frac{1}{\mathsf{mmse}(Z, \mathsf{snr} = 0|Y)} - 1 \right)^{-1} \\
&= \left( \frac{1}{\mathbb{E}\left(\mathrm{var}[Z|Y]\right)} - 1 \right)^{-1} \\
&= \left( \frac{1}{\mathbb{E}\left(\mathbb{E}[Z^2|Y] - \mathbb{E}^2[Z|Y]\right)} - 1 \right)^{-1} \\
&= \left( \frac{1}{1 - \mathbb{E}\left(\mathbb{E}^2[Z|Y]\right)} - 1 \right)^{-1} \quad (\mathbb{E}[Z^2] = 1),
\end{aligned}
\tag{107}
$$

where $\mathbb{E}[Z^2] = 1$ since $Z \sim \mathcal{N}(0, 1)$. Hence, $\phi(1) \geq 0$ and $\phi(1) < \infty$ if $\mathbb{E}[Z|Y] \neq 0$.

*Proof of (ii):* Similar to the proof of part (i), to prove $\Phi(v_l)$ is increasing, we only need to verify

$$
\frac{1}{\delta} \mathbb{E}\left[ \left( \frac{v_l \lambda}{v_l + \lambda} \right)^2 \right] > \left( \frac{1}{\delta} \mathbb{E}\left[ \frac{v_l \lambda}{v_l + \lambda} \right] \right)^2, \quad \forall v_r \in (0, 1].
\tag{108}
$$

When $\delta > 1$, Jensen's inequality yields the result:

$$\frac{1}{\delta}\mathbb{E}\left[\left(\frac{v_l\lambda}{v_l+\lambda}\right)^2\right] > \frac{1}{\delta}\left(\mathbb{E}\left[\frac{v_l\lambda}{v_l+\lambda}\right]\right)^2 > \left(\frac{1}{\delta}\mathbb{E}\left[\frac{v_l\lambda}{v_l+\lambda}\right]\right)^2, \quad \forall v_r \in (0,1].$$

For $\delta \le 1$, note that $P_\lambda = (1-\delta)P_0 + \delta P_{\tilde{\lambda}}$ where $P_{\tilde{\lambda}}$ denotes the asymptotic eigenvalue distribution of $\boldsymbol{A}\boldsymbol{A}^{\mathsf{T}}$ (we have $\mathbb{E}[\tilde{\lambda}^2] = 1$). Hence, (108) can be reformulated as

$$\mathbb{E}\left[\left(\frac{v_l\tilde{\lambda}}{v_l+\tilde{\lambda}}\right)^2\right] > \left(\mathbb{E}\left[\frac{v_l\tilde{\lambda}}{v_l+\tilde{\lambda}}\right]\right)^2, \quad \forall v_r \in (0,1],$$

and holds due to Jensen's inequality. $\qquad\square$

**Lemma 20.** *If $f(z) = f(-z), \forall z$, then $\mathsf{mmse}_z(1) = 1$. Further, $(V_r, V_l) = (1, \infty)$ is a fixed point of the state evolution equations in (7).*

*Proof.* Recall that $\mathsf{mmse}_z(v_r) = \mathsf{mmse}(Z, v_r^{-1} - 1|Y)$. Hence, $\mathsf{mmse}_z(1) = \mathsf{mmse}(Z, \mathsf{snr} = 0|Y)$ and

$$\mathsf{mmse}(Z, \mathsf{snr} = 0|Y) = \mathbb{E}\left(\mathbb{E}[|Z|^2|Y] - \mathbb{E}^2[Z|Y]\right)$$
$$= \mathbb{E}\left(\mathbb{E}[|Z|^2|Y]\right)$$
$$= \mathbb{E}[|Z|^2] = 1.$$

A simple calculation shows that $(V_r, V_l) = (1, 0)$ is a fixed point of (7). $\qquad\square$

The following lemma summarizes a few useful properties of $\phi(v_r, \sigma_w^2)$ (which is the noisy counterpart of $\phi(v_r)$).

**Lemma 21.** *Define*

$$\mathsf{mmse}_z(v_r, \sigma_w^2) \triangleq \mathbb{E}\left[\left(\mathbb{E}[Z|Y_\sigma, Z_r] - Z\right)^2\right], \tag{109}$$

*where $Z_r = (1 - v_r)Z + \sqrt{v_r(1 - v_r)}N$, $Y_\sigma = f(Z + \sigma_w W)$, $Z, W, N$ are mutually independent standar Gaussian RVs. Define*

$$\phi(v_r, \sigma_w^2) = \left(\frac{1}{\mathsf{mmse}_z(v_r, \sigma_w^2)} - \frac{1}{v_r}\right)^{-1}. \tag{110}$$

*For any $\sigma_w > 0$ and $v_r \in (0, 1)$, $\phi(v_r, \sigma_w^2)$ satisfies the following:*

*(i) $\phi(v_r, \sigma_w^2)$ is continuous and increasing in $v_r \in [0, 1)$. Further, $\phi(v_r, \sigma_w^2) \ge 0$;*

*(ii) $\sigma_w^2 \le \phi(v_r, \sigma_w^2) < \infty$, $\forall v_r \in [0, 1)$.*

*Proof. Part (i):* Same as Lemma 19-(i).

*Part (ii):* We will show that $\mathsf{mmse}(v_r, \sigma_w^2)$ can be rewritten as

$$\mathsf{mmse}_z(v_r, \sigma_w^2) = \left(\frac{v_r}{v_r + \sigma_w^2}\right)^2 \cdot \mathbb{E}\left(U - \mathbb{E}[U|Z_r, Y_\sigma]\right)^2 + \frac{v_r\sigma_w^2}{v_r + \sigma_w^2}, \tag{111}$$

where $U = Z + \sigma_w W$, $Y_\sigma = f(U)$, and $(U, Z_r) \sim \mathcal{N}(\boldsymbol{0}, \boldsymbol{\Sigma})$ where

$$\boldsymbol{\Sigma} = \begin{bmatrix} 1 + \sigma_w^2 & 1 - v_r \\ 1 - v_r & 1 - v_r \end{bmatrix}.$$

From (111) we have

$$\mathsf{mmse}_z(v_r, \sigma_w^2) \ge \frac{v_r\sigma_w^2}{v_r + \sigma_w^2},$$

which together with (110) yields $\phi(v_r, \sigma_w^2) \geq \sigma_w^2$. We next prove the boundedness of $\phi(v_r, \sigma_w^2)$. Substituting (111) into (110) and after straightforward calculations, we have

$$\phi(v_r, \sigma_w^2) = \frac{v_r \cdot \mathbb{E}\Big(U - \mathbb{E}[U|Z_r, Y_\sigma]\Big)^2}{v_r + \sigma_w^2 - \mathbb{E}\Big(U - \mathbb{E}[U|Z_r, Y_\sigma]\Big)^2}.$$

Since conditioning reduces MMSE [30, Proposition 11], we have

$$\mathbb{E}\Big(U - \mathbb{E}[U|Z_r, Y_\sigma]\Big)^2 \leq \mathbb{E}\Big(U - \mathbb{E}[U|Z_r]\Big)^2 = v_r + \sigma_w^2,$$

where the inequality is strict whenever $Y_\sigma$ is not independent of $U$. All together, we have $\phi(v_r, \sigma_w^2) < \infty$.

It only remains to prove (111). Let us write $Z = Z_r + \tilde{Z}$, where $\tilde{Z} \sim \mathcal{N}(0, v_r)$ is independent of $Z_r$. We have

$$\tilde{U} \triangleq U - Z_r = \tilde{Z} + \sigma_w W.$$

Define

$$\tilde{Z}_{\tilde{u}}^{\perp} \triangleq \tilde{Z} - \frac{v_r}{v_r + \sigma_w^2}\tilde{U}.$$

By construction, $\tilde{Z}_{\tilde{u}}^{\perp} \perp\!\!\!\perp \tilde{U}$. We also have $\tilde{Z}_{\tilde{u}}^{\perp} \perp\!\!\!\perp Z_r$[6], since $\tilde{Z}_{\tilde{u}}^{\perp}$ a linear combination of $\tilde{Z}$ and $W$ and the latter two RVs are independent of $Z_r$. Also, $\tilde{Z}_{\tilde{u}}^{\perp} \perp\!\!\!\perp Y_\sigma$ according to Lemma 22. Hence,

$$\mathbb{E}[\tilde{Z}|Z_r, Y_\sigma] = \frac{v_r}{v_r + \sigma_w^2} \cdot \mathbb{E}[\tilde{U}|Z_r, Y_\sigma] + \mathbb{E}[\tilde{Z}_{\tilde{u}}^{\perp}|Z_r, Y_\sigma]$$

$$= \frac{v_r}{v_r + \sigma_w^2} \cdot \mathbb{E}[\tilde{U}|Z_r, Y_\sigma],$$

where the last step is due to the independence of $\tilde{Z}_{\tilde{u}}^{\perp}$ and $(Z_r, Y_\sigma)$ and the fact that $\tilde{Z}_{\tilde{u}}^{\perp}$ is zero-mean Gaussian. Hence, we have

$$\mathbb{E}\Big(Z - \mathbb{E}[Z|Z_r, Y_\sigma]\Big)^2 = \mathbb{E}\Big(\tilde{Z} - \mathbb{E}[\tilde{Z}|Z_r, Y_\sigma]\Big)^2$$

$$= \mathbb{E}\Big(\frac{v_r}{v_r + \sigma_w^2}\tilde{U} + \tilde{Z}_{\tilde{u}}^{\perp} - \mathbb{E}[\tilde{Z}|Z_r, Y_\sigma]\Big)^2$$

$$= \mathbb{E}\Big(\frac{v_r}{v_r + \sigma_w^2}\tilde{U} + \tilde{Z}_{\tilde{u}}^{\perp} - \frac{v_r}{v_r + \sigma_w^2} \cdot \mathbb{E}[\tilde{U}|Z_r, Y_\sigma]\Big)^2$$

$$\overset{(a)}{=} \Big(\frac{v_r}{v_r + \sigma_w^2}\Big)^2 \cdot \mathbb{E}\Big(\tilde{U} - \mathbb{E}[\tilde{U}|Z_r, Y_\sigma]\Big)^2 + \frac{v_r \sigma_w^2}{v_r + \sigma_w^2}$$

$$= \Big(\frac{v_r}{v_r + \sigma_w^2}\Big)^2 \cdot \mathbb{E}\Big(U - \mathbb{E}[U|Z_r, Y_\sigma]\Big)^2 + \frac{v_r \sigma_w^2}{v_r + \sigma_w^2}$$

where step (a) is due to the fact that $\tilde{Z}_{\tilde{u}}^{\perp} \perp\!\!\!\perp (\tilde{U}, Z_r, Y_\sigma)$ and $\mathbb{E}[(\tilde{Z}_{\tilde{u}}^{\perp})^2] = \frac{v_r \sigma_w^2}{v_r + \sigma_w^2}$. $\qquad \square$

**Lemma 22.** *Consider two independent Gaussian RVs: $Z \sim \mathcal{N}(0, \tau)$ and $W \sim \mathcal{N}(0, 1)$. Suppose $U = Z + \sigma_w W$ and $Y_\sigma \sim P_{Y_\sigma}$, where $P_{Y_\sigma} \propto P_U \cdot P_{Y_\sigma|U}$ and $P_{Y_\sigma|U}$ is an arbitrary distribution. Define $Z_u^{\perp} \triangleq Z - \frac{\tau}{\tau + \sigma_w^2}U$. Then, we have $Z_u^{\perp} \perp\!\!\!\perp (U, Y_\sigma)$.*

*Proof.* It is straightforward to show $Z_u^{\perp} \perp\!\!\!\perp U$. Since $Y_\sigma$ is generated from $U$, we also have $Z_u^{\perp} \perp\!\!\!\perp Y_\sigma$. $\qquad \square$

---

[6]Throughout this paper, $A \perp\!\!\!\perp B$ denotes the random variables $A$ and $B$ are independent