# OpenReview forum: "Analysis of Sensing Spectral for Signal Recovery under a Generalized Linear Model"
_NeurIPS.cc/2021/Conference — NeurIPS 2021 Poster_

### Official Review · Reviewer_fMoa · 2021-06-30

**Rating:** 7
**Confidence:** 3

**Summary:**

This paper studies the influence of the sensing matrix on the recovery of signals in noiseless generalized linear models The authors define a mathematical property of a given spectrum, that they call “spikiness”. Given a specific activation function, the authors characterize the behaviour of the MSE and of perfect recovery threshold of the signal (as a function of the sampling ratio) depending on the spikiness of the singular value distribution of the sensing matrix. In particular, they show that for all non-linearities, more “spiky” spectrums always possess smaller (i.e. better) perfect recovery thresholds. Some generalizations are also given for simple instances of noise, namely when it is added before the non-linearity as $\mathbf{y} = f(\mathbf{A}\mathbf{x} + \sigma_w \mathbf{w})$ with $\mathbf{w} \sim \mathcal{N}(0,1)$, and infinitely small $\sigma_w \downarrow 0$.

Note that I could not check the calculations and proofs given in the supplementary material.

**I edited this review to modify my score, according to the comments below**

**Limitations And Societal Impact:**

Yes, the authors addressed correctly the societal impact of their work.

**Main Review:**

I found the paper overall well-written (with very few typos, see below) and pleasant to read. However in my point of view, some points lack discussion, and it is hard to see the scope of many results, as I detail below.

Overall, I believe this paper is above the acceptance threshold of NeurIPS 2021, and could have interest for the community of information theory and statistical-physics approaches to learning. However the authors should dismiss some of the comments and criticisms expressed hereafter. I believe the majority of these issues are more related to the discussion of the results than to the scientific content itself, which I found quite interesting.

**Strengths of the paper**

- As I mentioned, the writing of the paper is good. Previous literature is cited quite thoroughly, with some minor misses, see below.

- Concerning the comparison of the MSE achieved by different spectrums (Section 4.2), for some very common choices of activation (sign, absolute value) the results are strong, as the function $G(v_r)$ is monotonous, so the results allow to precisely compare the performance of two spectrums (if one is more spiky than the other).

- On the other hand, for any non-linearity the authors present interesting results on the comparison of the perfect recovery thresholds for the algorithm.  In particular, they show that they can build sensing matrices that reach a spectrum-independent lower bound: e.g. in real noiseless phase retrieval, this amounts to build spectra that have perfect algorithmic recovery already at $\delta = 1$ (as in [1]).

[1] AUBIN, Benjamin, LOUREIRO, Bruno, BAKER, Antoine, et al. Exact asymptotics for phase retrieval and compressed sensing with random generative priors. In : Mathematical and Scientific Machine Learning. PMLR, 2020. p. 55-73.

- Previous results are shown to be stable against some types of noise (in Sections 2.3 and 4.4), which adds relevance to the analysis, as in many actual applications (e.g. compressive sensing) the observations are indeed affected by noise.

**Main concerns and remarks**

- Being quite used with the AMP literature, I am a bit puzzled by the use of the “GLM-EP” algorithm. While the authors mention briefly its links with Vector Approximate Message Passing in the introduction, I would have liked to understand better if there is an actual reason to not choose this algorithm which is described for the GLM in [2] (and also possesses a well-studied State Evolution, and is in my point of view more widely used), or if it would be an equivalent choice which would yield the same results.

[2] SCHNITER, Philip, RANGAN, Sundeep, and FLETCHER, Alyson K. Vector approximate message passing for the generalized linear model. In : 2016 50th Asilomar Conference on Signals, Systems and Computers. IEEE, 2016. p. 1525-1529.

- I am bit puzzled about the impact of the results for discrete outputs: the lower bound of Theorem 1 is $\delta_\mathrm{opt} = + \infty$ since $d(f(Z)) = 0$.  Are Theorems 1, 2 and 3 informative for such problems (e.g. 1-bit compressed sensing with $f(z) = \mathrm{sign}(z)$, which is taken as example in the paper) ?

- I believe the paper lacks discussion of the notion of spikiness of the spectrum, defined by the authors in Definition 3. As I was not familiar with this definition, I was a bit surprised that orthogonal matrices are less “spiky” than Gaussian matrices, which seems strange: if I am not mistaken, the Marchenko-Pastur density is more spiky than a delta distribution according to this definition. This is counter-intuitive when considering spiky as the inverse of flatness (however when looking at it as a quantification of “wealth inequality” as briefly mentioned in the paper it makes a bit more sense), and I believe that the paper would benefit from adding a clearer discussion of this point, with examples.
On a related note, here spikiness is only related to the asymptotic eigenvalue distribution of the sensing matrix. I find the word choice confusing, as in this context I expected rather spikiness to describe the presence of isolated eigenvalues in the spectrum of the sensing matrix (commonly referred to as “spikes” in the literature), that are not present in the asymptotic eigenvalue distribution but that might be very important in the dynamics of the algorithms, a topic not discussed in the paper.

- Importantly, spikiness in the sense of Definition 3 is not a complete order between spectra: two arbitrary spectra are in general not comparable, and perhaps the authors could discuss further how this point can impact the scope of their results.

**Minor points and questions**

- Could some results be generalized to stronger versions of the noise? By “stronger” I mean either not infinitely small, or more generic type of noises added after the non-linearity, which are standard to take into account in message-passing approaches, and (I believe) more interesting for real applications, e.g. Poisson-noise phase retrieval, compressive sensing, etc... I am thinking in particular to the results of Section 4.2 on the comparisons of the MSE (as perfect recovery might be impossible with noise).

- In hypothesis (A.2) the authors assume that the support of the limiting distribution is an interval $[a,b]$. Is this necessary, or would any compact support work? I don’t see where the fact that it is an interval is needed (but it might be a technicality in the proofs).

- Similarly, in (A.2) both right and left rotation invariance of the sensing matrix are assumed. Is it necessary? For instance, the VAMP approach requires only right-rotation invariance, see [2].

- In their discussion of the literature on the performance of message-passing approaches, I believe the authors should mention [3], which precisely analyses the state evolution of the Vector Approximate Message Passing approach for rotationally-invariant generalized linear models, from the point of view of statistical physics.

[3] TAKAHASHI, Takashi et KABASHIMA, Yoshiyuki. Macroscopic analysis of vector approximate message passing in a model mismatch setting. In : 2020 IEEE International Symposium on Information Theory (ISIT). IEEE, 2020. p. 1403-1408.

**Typos**

- Line 31: “has certain level” → “has a certain level”
- Line 101: “we define, enable” → “we define enable”

**Time Spent Reviewing:**

around 5 hours

---

> ### Author Response · Authors · 2021-08-09
> **Response to Reviewer fMoa**
>
> We thank the reviewer for the detailed comments.
>
> - **The work of Aubin et al.:** We thank the reviewer for pointing out this interesting paper, which considers a multi-layer network and uses AMP for reconstruction. While the setup of this paper is different from ours, it is very relevant and we will include it in our related work section.
>
> - **Connection of GLM-EP with G-VAMP of Schniter et al.:** We choose [21] as our main reference for state evolution of GLM-EP because it has a Theorem (with verifiable assumptions) and provides a proof. Algorithmically, the GLM-EP algorithm is equivalent to G-VAMP of Schniter et al. To our knowledge, the work of Fletcher et al. [20] appeared earlier on arXiv than Schniter et al. We therefore used the name "GLM-EP'', similar to the name "expectation consistent inference'' as adopted in [20]. We will cite the paper by Schniter et al. and state more explicitly that GLM-EP (which follows from [20]) is equivalent to G-VAMP algorithmically.
>
> - **Discrete outputs:** Our results imply that that perfect recovery is not possible in the asymptotic setting where $m/n\to\delta\in(1,\infty)$ in this case. We will clarify this point in a future revision.
>
> - **Discussion of Spikeness:** We agree with the reviewer that the word "spiky'' might cause some confusion. We will add more discussions and clarify the definition of spikeness. We currently do not have a better word for this notion. However, we are open to the suggestions we receive from the reviewers. Spikness might be interpreted intuitively in the following way. Consider the non-zero singular values of a $m\times n$ partial orthogonal matrix and an i.i.d. Gaussian matrix (with proper normalization). If we sort the $n$ singular values and plot the resulting vector (not a histogram), then the plot for the partial orthogonal matrix is flat because the non-zero singular values are identical. In comparison, the plot for the i.i.d. Gaussian matrix is less uniform. This name "spiky" might make sense if we consider this interpretation.
>
> - **Lorenz order is not a complete order:** Yes, you are correct. We will add some comments about this point.
>
> - **Other types of noisy models:** The same approach can be used for studying more generic noise models. However, the impact of such models on functions $g$ and $G$ should be studied more carefully. Hence, we have decided to explore this direction in a future paper.
>
>
> - **Support of the eigenvalue distribution:** This is mainly for the simplicity of analysis. For instance, it guarantees existence of various expectations involving the limiting eigenvalue distribution. Compact support also works (with the smallest eigenvalue strictly positive).
>
> - **Left and right rotational invariance:** We believe both left and right rotational invariance are required. Notice that only right invariance is required for the linear system $\mathbf{y}=\mathbf{Ax}+\mathbf{n}$ if $\mathbf{n}$ is zero-mean iid Gaussian, since the left orthogonal matrix of $\mathbf{A}$ can be pre-multiplied without changing the distribution of the noise. However, left invariance is also required for the GLM problem. To see this, consider the following simple argument. Let $\mathbf{z}_\star:=\mathbf{UDV}^T\mathbf{x}_\star$. When $\mathbf{V}$ is a Haar matrix, $\mathbf{V}^T\mathbf{x}_\star$ is approximately Gaussian. When the limiting distribution of $\mathbf{D}$ of non-Gaussian, the empirical distribution of $\mathbf{DV}^T\mathbf{x}_\star$ would also be non-Gaussian. When $\mathbf{U}$ is not Haar (e.g., $\mathbf{U}=\mathbf{I}$), then $\mathbf{z}_\star$ would generally be non-Gaussian, and the state evolution in this paper would not hold anymore. We checked the referred paper by Schniter et al. and found that it explicitly stated in the simulation section (Section IV) that $\mathbf{A}$ is both left and right rotationally invariant.
>
> - **The work of Takashi and Kabashima:** We thank the reviewer for pointing out this paper. We will cite it in our related work section.

---

> > ### Comment · Reviewer_fMoa · 2021-08-25
> > **Response to the authors**
> >
> > I thank the authors for their detailed response, in which my comments were well taken into account and my questions thoroughly answered. I look forward to the revised version of this work.
> > However, after discussions and readings, while I believe that this paper is slightly above the acceptance threshold, due to the acknowledged lack of numerical simulations to support the claims of the paper as pointed out by other reviewers, I will decrease my score to 6.

---

> > > ### Author Response · Authors · 2021-08-27
> > > **Response to Reviewer fMoa**
> > >
> > > As we mention in our response we are planning to include the following simulations in the paper.
> > >
> > > Denote $f_1(z)=|z|$; (2): $f_2(z) = |z| * I_{|z|<1} + (|z|-1) * I_{|z|>=1}$, where $I$ is an indicator function; (3) $f_3(z)=sign(z)$; , We tested various settings of $m$ and $n$. Below, we report our numerical results for two settings of $n$ (with $\delta:=m/n$ fixed): $n = 10^5$ and $n=2000$. The spectrum of the sensing matrix is drawn from a "geometric distribution”: $P(\lambda):=1/(\beta*\lambda)$ for $\lambda\in(\delta*A(\beta)*e^{-\beta},\delta*A(\beta))$, where $\beta>0$ is a parameter controlling the spikiness and $A(\beta):=\beta/(1-e^{-\beta})$.
> > >
> > > **Experiment 1:** $\beta=20$. $n=10^5$. $m=\lceil(n*1.01)\rceil $. The signal is iid standard Gaussian. The results are obtained via 1000 independent runs. Note that $\delta\approx1.01$ is very close to the information theoretic threshold for $f_1$ and $f_2$. This experiment demonstrates that with spiky enough spectrum the GLM-EP algorithm can indeed approach (at large system size) the information theoretic lower bound.
> > >
> > > | $f_1$ | $t=1$ | $t=2$| $t=3$ | $t=4$| $t=5$ | $t=6$| $t=7$ | $t=8$|
> > > | ----| ----| ----| ----| ----| ----| ----| ----| ----|
> > > | MSE (SE theory)  | 0.9491  |  0.8396  |  0.7050  |  0.5472  |  0.3575  |  0.1332  |  0.0044  |  0.0001 |
> > > | MSE (mean)  | 0.9479 |  0.8384 |  0.7032  |  0.5445  |  0.3542  |  0.1285  |  0.0040  |  0.0001 |
> > > | MSE (std)  | 0.0020  |  0.0042  |  0.0044  |  0.0057  |  0.0069  |  0.0089  |  0.0012  |  0.0000 |
> > >
> > > | $f_2$ | $t=1$ | $t=2$| $t=3$ | $t=4$| $t=5$ | $t=6$| $t=7$ | $t=8$| $t=9$| $t=10$| $t=11$| $t=12$|
> > > | ----| ----| ----| ----| ----| ----| ----| ----| ----| ----| ----| ----| ----|
> > > | MSE (SE theory)  | 0.9632  |  0.9033  |  0.8660 |   0.8433   | 0.8204  |  0.7879 |   0.7295 |  0.6059  |  0.4280  |  0.2134  |  0.0193  |  0.0004 |
> > > | MSE (mean)  | 0.9627  |  0.9028  |  0.8659  |  0.8431 |   0.8200  |  0.7868  |  0.7270 |   0.5999  |  0.4207  |  0.2057  |  0.0222  |  0.0004 |
> > > | MSE (std)  | 0.0022 |  0.0032  |  0.0031  |  0.0034 |   0.0046  | 0.0076  |  0.0151  |  0.0311  |  0.0378  |  0.0446  |  0.0170  |  0.0004 |
> > >
> > > **Experiment 2:** $f=f_1$ (i.e., phase retrieval). $n=2000$. Other settings are the same as in experiment 1. (We only provide the simulation results for $f_1$ for this experiment.)
> > >
> > > | $f_1$| $t=1$ | $t=2$| $t=3$ | $t=4$| $t=5$ | $t=6$| $t=7$ | $t=8$|
> > > | ----| ----| ----| ----| ----| ----| ----| ----| ----|
> > > | MSE (SE theory)  | 0.9491  |  0.8396  |  0.7050  |  0.5472  |  0.3575  |  0.1332  |  0.0044  |  0.0001 |
> > > | MSE (mean)  | 0.9584  |  1.2574  |  0.7225 |   0.5709  |  0.3816  |  0.1602  |  0.0266  |  0.0020 |
> > > | MSE (std)  | 0.0370  |  1.4092 |   0.0531   | 0.0818  |  0.0844  |  0.1048   | 0.0852   | 0.0163 |
> > >
> > > **Experiment 3:** $f=f_3$ (1-bit CS). $n=10^5$. $\beta = 0$ (i.e., column-orthogonal matrix). The MSE is averaged over 100 independent runs. Number of iterations: 20.
> > >
> > > | $\delta=1.5$ | $\delta=2$ | $\delta=2.5$ | $\delta=3$ |
> > > | ----| ----| ----| ----|
> > > | 0.2771 |   0.2091  |  0.1622 |   0.1286 |
> > >
> > > **Experiment 4:** Noisy setting. $f = f_1$ (phase retrieval). $\delta=1.1$. $n=10^5$. $\beta=10$. The MSE is averaged over 100 independent runs. Number of iterations: 10.
> > >
> > > | SNR: 30dB | SNR: 35dB  | SNR: 40dB | SNR: 45dB | SNR: 50dB |
> > > | ----| ----| ----| ----| ----|
> > > | 1.2889e-01 | 5.9298e-02 | 2.1847e-02 | 6.9412e-03 | 2.1481e-03 |
> > >
> > > **Experiment 5:** A phase diagram plot for $f_1$ and $f_2$. We will not list the detailed results here due to space limitation.

---

> > > > ### Comment · Reviewer_fMoa · 2021-08-30
> > > > **Response to the authors**
> > > >
> > > > I thank the authors for sharing the results of their numerical experiments, which should definitely be a part of the paper. In accordance with another reviewer, I believe that these experiments and their discussion can really improve a lot the paper, but would surely require another rewriting. Adding these in a supplementary material to the present paper would increase its scientific value (and I have put back my grade to its original value accordingly, as the authors responded to an importance criticism), but might not do these results justice. On the other hand, adding them in the main part would perhaps require another round of reviews. In accordance with another reviewer, I will let the AC decide on this point.

---

### Official Review · Reviewer_h9Kx · 2021-06-30

**Rating:** 6
**Confidence:** 3

**Summary:**

This work investigate the problem of recovering a signal vector x from measurements y=f(Ax) where A is a known matrix and f is a scalar function that is applied coordinate-wise.

In many applications, the practitioner has the freedom of the choice of the matrix A. The present paper seek to understand the influence of the spectrum of A on the estimation performance.

The authors study this problem in the asymptotic regime where the number of observations as well as the number of entries to estimate (the dimension of x) go to infinity, while their ratio converges to some positive number $\delta$. They also assume the entries of x to be iid from some prior distribution with unit second moment, and A to be random with SVD $A = U \Sigma V^T$ where U and V independent and Haar-distributed and that the empirical distribution of the square of the diagonal values of Sigma converges to some distribution $P$.

Most of the paper is devoted to the noiseless case (f is a deterministic function). In this setting, the authors show:
1) A lower bound delta_p on delta (ratio #observations / # unknown variables) that is needed in order to reconstruct x perfectly whp.
2) Introduce an Expectation-Propagation algorithm, whose asymptotic behavior can be precisely tracked by a simple scalar recursion. The authors therefore obtain a precise formula for MSE(P,f), the limiting mean square error of the EP-algorithm.
3) Derive a function G (that only depends on f - and delta I presume, otherwise I guess it would be incompatible with (4) below) such that:
                - If G is increasing, then spikier distributions P lead to smaller MSE(P,f)
                - If G is decreasing, then spikier distributions P lead to larger MSE(P,f)

4) Show that spikier distributions P always need a smaller number of measurement in order to achieve perfect reconstruction with the EP algorithm.

A noisy case is also investigated but with less results. Assume now that y=f(Ax + w) where w ~ N(0, sigma Id) is some  Gaussian noise. The authors show in that case that
1) One need to have delta > delta_p in order to have a bounded noise-sensitivity.
2) That the MSE of the EP-algorithm verify when sigma -> 0:
   MSE(sigma) ~ Constant(delta,f) E_P[ 1 / lambda ] sigma^2


**Ethical Concerns:**

No issue here.

**Limitations And Societal Impact:**

Yes, the author brilliantly address these limitations. No issue here.

**Main Review:**

Strength:
This paper is relatively original and novel: there is not that many result about the impact of the spectrum of the sensing matrix for such nonlinear inverse problem. The insights from this paper could potentially be very useful in practice.

The results obtained are also surprising: spikier eigenvalues distribution need less samples in order to achieve perfect recovery, but may lead to larger MSE (for some delta). I may miss something here, because this seems a bit contradictory to me:
if $MSE_{P_1}(delta, f) \geq MSE_{P_2}(delta,f)$ for all $\delta$, then one has $\delta^p_{P_2} \leq \delta^p_{P_1}$. I guess that my issue comes from the fact that the function $G$ in Lemma 6 depends on $\delta$ and that its monotonicity may change with $\delta$. If this is correct, I guess it would be important to mention it and specify the value of $\delta$ chosen in Figure 2. Again, I might be missing something important here.


Weaknesses:
In my view, the main weakness of this paper is that most of the results only apply to the EP-algorithm. Does the phenomena observed here also apply to other algorithms ? For example, does a spikier spectrum leads to a smoother optimisation landscape and help gradient-based method to converge ?
Also, the information - theoretic results of the paper are relatively weak: only a lower bound is obtain and no claim is made about the dependency of the information-theoretic threshold on the eigenvalue distribution.

**Time Spent Reviewing:**

3 hours

---

> ### Author Response · Authors · 2021-08-09
> **Response to Reviewer h9Kx**
>
> We thank the reviewer for a nice summary of our contributions and the detailed comments.
>
> - **Possible contradiction on the impact of MSE and threshold:** We thank the reviewer for pointing this out. Indeed, since $G(v_r):=\max(g(v_r),0)$ depends on $\delta$, the monotonicity of $G$ can change with $\delta$. Note that $G$ for $f=|z|$ and $f=\text{sign}(z)$ is monotonic for all $\delta$ as the function $g$ is monotonic. We will clarify this point in the revised paper. Notice that even when the monotonicity of $G$ does not change with $\delta$, our conclusions about MSE and threshold do not contradict. To see this, consider the "saturation function'' $f(z)=\min(\max(z,-1),1)$. For this $f$, the function $G$ is decreasing (strictly speaking, non-increasing) for all $\delta$. For this function, we have $MSE_{P_1}(\delta,f) \ge MSE_{P_2}(\delta,f)$ for all $\delta$, and at the same time it can be shown that $\delta^p_{P_1}=\delta_{P_2}^p$. Therefore, both Lemma 6 and Theorem 3 hold.
>
> - **Implication to other algorithms:** The impact of the spectrum on the optimization landscape and the performance of other algorithms is an exciting research direction, that we would like to pursue in the future. Our current results do not apply to other algorithms.
>
> - **Dependency of IT threshold on spectrum:** The reviewer is correct that we only provide a lower bound and do not claim the dependency of the information theoretic (IT) threshold on the eigenvalue distribution. The actual IT threshold may be calculated using the replica method. Our guess is that our lower bound is the IT threshold (as calculated by replica method), regardless of the eigenvalue distribution. However, our guess has yet to be confirmed, which we wish to finish before our next revision. Finally, whether replica method is rigorous or not for the general rotationally-invariant model is still not known, and it remains an important open problem in the field.

---

> > ### Comment · Reviewer_h9Kx · 2021-08-21
> > **Response to the authors**
> >
> > - I would like to warmly thank the authors for their answer and in particular for the clarification of the first point. It makes sense for me now.
> >
> > - Your remark about the IT threshold is very interesting, I look forward for the next revision.
> >
> >
> > All in all I think that this paper - even if its result are mostly limited to the EP algorithm - contains many interesting contributions and should be accepted in my view.

---

### Official Review · Reviewer_Mw3P · 2021-07-14

**Rating:** 5
**Confidence:** 3

**Summary:**

The authors consider the recovery of unstructured random signals x from measurements of the form y=f(Ax), where f(.) is a component-wise nonlinear function, A is a large and rotationally invariant random matrix, and the dimension of y is strictly larger than the dimension of x.  They assume the use of the GLM-EP recovery algorithm, for which a state-evolution has been previously derived in [21].  (Actually, they use a simplification GLM-EP since they assume an unstructured signal.)  Their primary focus is understanding how the spectrum of A affects reconstruction performance under the stated assumptions.  For this, they define a measure of "spikiness" based on the Lorenz partial order.  First, they show that, for some nonlinearities (e.g., the magnitude nonlinearity used in phase retrieval), more spikiness reduces MSE; for other nonlinearities (e.g., 1-bit sensing), less spikiness reduces MSE, and for yet other nonlinearities, there is no monotonic relationship between spikiness and MSE.  Second, they show that, in the noiseless setting, for any nonlinearity, more spikiness reduces the number of measurements needed for perfect recovery.

**Ethical Concerns:**

There are no ethical issues in this paper.

**Limitations And Societal Impact:**

There is no potential negative societal impact of this work.

**Main Review:**

The results are interesting, and the paper is clearly written (for the most part).  Although the state-evolution analysis appears to be borrowed entirely from [21], the use of the Lorenz partial order to characterize the recovery performance is novel.

That said, I think the title, abstract, and introduction significantly over-sell the scope of the results, as described below.
1. The paper is not about "designing optimal sensing matrices" for signal recovery, but only about analyzing the effect of the spectrum of a random matrix for signal recovery.  There is a large literature on the design of optimal sensing matrices for signal recovery (e.g., for MRI), but that problem is entirely different than the one considered by the authors.  First, no spectrum "design" procedure is given in the paper, only an analysis of a given spectrum.  Second, the authors never give an expression for the "optimal" spectrum.  Third, the design quantity under consideration (the spectrum) is not a matrix but a vector.  Fourth, the authors only consider a very particular form of "nonlinear inverse problem," which is the GLM.  For all these reasons, the title should be changed to "Analysis of Sensing Spectra for Signal Recovery under a Generalized Linear Model."
2. The considered recovery problem is non-compressive, in that the number of measurements must be strictly larger than the signal dimension.  Thus, it does not apply to compressed sensing as claimed in the abstract.  The restriction to non-compressive sensing should be clearly stated up front.
3. Unlike nearly every existing paper on AMP/EP/GLM/CS, the method under review has no ability to leverage signal structure. Yet, the signal is still not "generic" as claimed on line 129, because it must be drawn i.i.d. from an absolutely continuous distribution.  So, for example, finite-alphabet signals are not allowed, and even sparse signals are not allowed.  The i.i.d. structural assumption is also restrictive and not "standard" as claimed on line 130, because there are now numerous papers on AMP/EP methods with non-separable denoisers designed to recover non-iid signals (e.g., natural images).

The paper also falls significantly short on practical demonstration.  For example, although the authors prove that MSE can be decreased with an appropriate spectrum, but how much the MSE decreases in practice (e.g., large but finite dimension, large but finite SNR) in the (real) phase-retrieval and 1-bit sensing problems is not clear.

Minor comments:
1. It would be helpful to label the axes in Figure 1.
2. It would be helpful to label the subplots in Figure 2 with the corresponding f(.) functions.



**Time Spent Reviewing:**

3 hours

---

> ### Author Response · Authors · 2021-08-09
> **Response to Reviewer Mw3P**
>
> We thank the reviewer for the detailed comments.
>
> - **Paper title:** We agree that this work is mainly about analyzing the impact of the spectrum. We will revise the title of this paper accordingly.
>
> - **Non-compressive:** Yes, this is indeed an important point that we should emphasize. We will clearly state this in the abstract and introduction in the revised version.
>
> - **Signal not generic:** The assumption that the signal distribution is absolutely-continuous is mainly for establishing the information theoretic limit. When the signal distribution is "structured" (for instance, if it is a mixed discrete-continuous distribution), it is possible to design an EP algorithm to incorporate such information to achieve improved performance. While this is certainly an important research direction, for clarity and conciseness we have left the pursuit of this direction for future research.
>
> - **Practical demonstrations:** We plan to include more simulations in a future version of this paper. Our simulation result can demonstrate the claimed benefit of designing spectral for practical (finite size, noisy) phase retrieval and 1-bit CS problems.

---

> > ### Comment · Reviewer_Mw3P · 2021-08-23
> > **Reply to author's response**
> >
> > I thank the authors for their response.  Due to the acknowledge limitations on the need for signal structure and the acknowledged lack of convincing simulations in the current version of the paper, I will decrease my score to 5.

---

> > > ### Author Response · Authors · 2021-08-27
> > > **Response to Reviewer Mw3p**
> > >
> > > Regarding the structure, please note that given the generality level of this problem (general nonlinearity, generic spectrum, generic structure), it is impractical (and most probably impossible) to address all the challenges in one paper. Even in simpler problems, such as the analysis of phase retrieval problem (without considering the impact of spectrum), there have been more than 50 papers published in top journal and conference papers. We mention a few below:
> > >
> > > 1. Ma, Junjie, Ji Xu, and Arian Maleki. "Approximate message passing for amplitude based optimization." International Conference on Machine Learning. PMLR, 2018.
> > > 2. Chen, Yuxin, and Emmanuel J. Candès. "Solving random quadratic systems of equations is nearly as easy as solving linear systems." Advances in Neural Information Processing Systems 2015 (2015): 739-747.
> > > 3. Candes, Emmanuel J., Xiaodong Li, and Mahdi Soltanolkotabi. "Phase retrieval via Wirtinger flow: Theory and algorithms." IEEE Transactions on Information Theory 61.4 (2015): 1985-2007.
> > > 4. Goldstein, Tom, and Christoph Studer. "Phasemax: Convex phase retrieval via basis pursuit." IEEE Transactions on Information Theory 64.4 (2018): 2675-2689.
> > >
> > > If the main concern of the reviewer is the lack of simulations, as we discussed before we aim to include more simulations in a revised version. We list a few experimental results below (we will include other simulation results, e.g., phase diagram plot in the paper):
> > >
> > > Denote $f_1(z)=|z|$; (2): $f_2(z) = |z| * I_{|z|<1} + (|z|-1) * I_{|z|>=1}$, where $I$ is an indicator function; (3) $f_3(z)=sign(z)$; , We tested various settings of $m$ and $n$. Below, we report our numerical results for two settings of $n$ (with $\delta:=m/n$ fixed): $n = 10^5$ and $n=2000$. The spectrum of the sensing matrix is drawn from a "geometric distribution”: $P(\lambda):=1/(\beta*\lambda)$ for $\lambda\in(\delta*A(\beta)*e^{-\beta},\delta*A(\beta))$, where $\beta>0$ is a parameter controlling the spikiness and $A(\beta):=\beta/(1-e^{-\beta})$.
> > >
> > > **Experiment 1:** $\beta=20$. $n=10^5$. $m=\lceil(n*1.01)\rceil $. The signal is iid standard Gaussian. The results are obtained via 1000 independent runs. Note that $\delta\approx1.01$ is very close to the information theoretic threshold for $f_1$ and $f_2$. This experiment demonstrates that with spiky enough spectrum the GLM-EP algorithm can indeed approach (at large system size) the information theoretic lower bound.
> > >
> > > | $f_1$ | $t=1$ | $t=2$| $t=3$ | $t=4$| $t=5$ | $t=6$| $t=7$ | $t=8$|
> > > | ----| ----| ----| ----| ----| ----| ----| ----| ----|
> > > | MSE (SE theory)  | 0.9491  |  0.8396  |  0.7050  |  0.5472  |  0.3575  |  0.1332  |  0.0044  |  0.0001 |
> > > | MSE (mean)  | 0.9479 |  0.8384 |  0.7032  |  0.5445  |  0.3542  |  0.1285  |  0.0040  |  0.0001 |
> > > | MSE (std)  | 0.0020  |  0.0042  |  0.0044  |  0.0057  |  0.0069  |  0.0089  |  0.0012  |  0.0000 |
> > >
> > > | $f_2$ | $t=1$ | $t=2$| $t=3$ | $t=4$| $t=5$ | $t=6$| $t=7$ | $t=8$| $t=9$| $t=10$| $t=11$| $t=12$|
> > > | ----| ----| ----| ----| ----| ----| ----| ----| ----| ----| ----| ----| ----|
> > > | MSE (SE theory)  | 0.9632  |  0.9033  |  0.8660 |   0.8433   | 0.8204  |  0.7879 |   0.7295 |  0.6059  |  0.4280  |  0.2134  |  0.0193  |  0.0004 |
> > > | MSE (mean)  | 0.9627  |  0.9028  |  0.8659  |  0.8431 |   0.8200  |  0.7868  |  0.7270 |   0.5999  |  0.4207  |  0.2057  |  0.0222  |  0.0004 |
> > > | MSE (std)  | 0.0022 |  0.0032  |  0.0031  |  0.0034 |   0.0046  | 0.0076  |  0.0151  |  0.0311  |  0.0378  |  0.0446  |  0.0170  |  0.0004 |
> > >
> > > **Experiment 2:** $f=f_1$ (i.e., phase retrieval). $n=2000$. Other settings are the same as in experiment 1. (We only provide the simulation results for $f_1$ for this experiment.)
> > >
> > > | $f_1$| $t=1$ | $t=2$| $t=3$ | $t=4$| $t=5$ | $t=6$| $t=7$ | $t=8$|
> > > | ----| ----| ----| ----| ----| ----| ----| ----| ----|
> > > | MSE (SE theory)  | 0.9491  |  0.8396  |  0.7050  |  0.5472  |  0.3575  |  0.1332  |  0.0044  |  0.0001 |
> > > | MSE (mean)  | 0.9584  |  1.2574  |  0.7225 |   0.5709  |  0.3816  |  0.1602  |  0.0266  |  0.0020 |
> > > | MSE (std)  | 0.0370  |  1.4092 |   0.0531   | 0.0818  |  0.0844  |  0.1048   | 0.0852   | 0.0163 |
> > >
> > > **Experiment 3:** $f=f_3$ (1-bit CS). $n=10^5$. $\beta = 0$ (i.e., column-orthogonal matrix). The MSE is averaged over 100 independent runs. Number of iterations: 20.
> > >
> > > | $\delta=1.5$ | $\delta=2$ | $\delta=2.5$ | $\delta=3$ |
> > > | ----| ----| ----| ----|
> > > | 0.2771 |   0.2091  |  0.1622 |   0.1286 |
> > >
> > > **Experiment 4:** Noisy setting. $f = f_1$ (phase retrieval). $\delta=1.1$. $n=10^5$. $\beta=10$. The MSE is averaged over 100 independent runs. Number of iterations: 10.
> > >
> > > | SNR: 30dB | SNR: 35dB  | SNR: 40dB | SNR: 45dB | SNR: 50dB |
> > > | ----| ----| ----| ----| ----|
> > > | 1.2889e-01 | 5.9298e-02 | 2.1847e-02 | 6.9412e-03 | 2.1481e-03 |

---

### Official Review · Reviewer_kRWV · 2021-07-16

**Rating:** 6
**Confidence:** 4

**Summary:**

This paper studies the impact of the sensing matrix spectrum in nonlinear inverse
problems. In the first three sections, the authors present the
information theoretical limits to perfect recovery, the expectation propagation
(EP) like algorithm to solve such nonlinear inverse problems, and its MSE performance
predicted by the state evolution formalism. However the main focus of this
research is the last section, where a measure of spikiness of the
sensing matrix spectrum is defined (based on Lorenz partial order), and its
impact on EP recovery is investigated, both in term of the MSE performance and
the measurement threshold for perfect recovery. This impact is shown to depend
on the type of nonlinearity used in the inverse problem, for instance for the absolute value
a spikier spectrum helps while for the sign function it hurts.

**Limitations And Societal Impact:**

The main limitation of this work is I think the assumption A1 on the prior,
which if I understand correctly excludes for instance a binary prior or a
spike-and-slab prior. Unfortunately this means that the perfect
recovery for a perceptron with binary weights, or the compressed sensing
application, are outside the scope of this study.
This limitation seems manifest in the GLM-EP algorithm which do not include a
prior denoising step. Perhaps this limitation could be better acknowledged by the
authors and highligthed in the introduction and discussion of the results.

Also the results of Lemma 6
(spikier spectrum leads to higher MSE for functions like sign) and
Theorem 3 (spikier spectrum always leads to lower perfect recovery threshold
whatever the nonlinearity) seem contradictory at first. How can the MSE
curve (vs $\delta$) be above for spikier spectrum
while at the same time have a lower perfect recovery threshold ?
If I understand correctly, the contradiction is resolved by Theorem 1,
as the optimal threshold $\delta^p_{opt}$ is
infinite for the sign activation. Perhaps this point could be clarified in
discussing Theorem 3.

Minor typos/details:
- lines 194 and 219 "where $V_r^{-1} = 0$"may be confusing. Perhaps "initialised at
$V_r^{t-1} \vert_{t=0} = 0$" would be clearer.
- line 273 "GMM-EP-app"
- line 290 "lemme"

**Main Review:**

Overall the manuscript is very clearly written, well referenced and the
main line of investigation is easy to follow. The results of the first three
sections, on the EP algorithm, its recovery performance and
information theoretical limits for the GLM are already known in the
literature. However these results are clearly meant to lay ground for the final
fourth section (impact of the sensing matrix spectrum), which is the
main goal of this research; they are nicely put together and provide all the
necessary information. The most interesting results are Lemma 6
(which shows how the spectrum spikiness of the spectrum helps or hurts the MSE performance
depending on the nonlinearity) and Theorem 3 (that the perfect recovery threshold is
always lower for spikier spectrum). As far as I know these results are
original and certainly help in designing better sensing matrices.

**Time Spent Reviewing:**

8

---

> ### Author Response · Authors · 2021-08-09
> **Response to Reviewer kRWV**
>
> We thank the reviewer for the encouraging comments.
>
> - **Prior assumption:** We agree with the reviewer that incorporating signal prior is an important topic. Following the reviewer's suggestion, we will acknowledge and highlight this point in the Introduction and discussions.
>
> - **Lemma 6 and Theorem 3:** The reviewer's interpretation about the sign function is correct. There are other functions (e.g., the saturation function $f(z)=\min(\max(z,-1),1)$) where spikier spectrum leads to higher MSE and at the same time all spectra have the same recovery threshold ($<\infty$). Hence, Lemma 6 and Theorem 3 also do not contradict for this function. This point is indeed a bit confusing, and we will clarify it in a revised version. We thank the reviewer for pointing this out.

---

> > ### Comment · Reviewer_kRWV · 2021-08-25
> > **Response to the authors**
> >
> > I thank the authors for their response. I look forward to the revised version where the limitation (on the signal prior /input structure) will be clearly acknowledged. This paper shoud still be accepted in my view, but I will decrease my score to 6 due to the lack of numerical illustrations as pointed out by the other reviewers.

---

> > > ### Author Response · Authors · 2021-08-27
> > > **Response to Reviewer kRWV**
> > >
> > > As we mention in our response we are planning to include the following simulations in the paper.
> > >
> > > Denote $f_1(z)=|z|$; (2): $f_2(z) = |z| * I_{|z|<1} + (|z|-1) * I_{|z|>=1}$, where $I$ is an indicator function; (3) $f_3(z)=sign(z)$; , We tested various settings of $m$ and $n$. Below, we report our numerical results for two settings of $n$ (with $\delta:=m/n$ fixed): $n = 10^5$ and $n=2000$. The spectrum of the sensing matrix is drawn from a "geometric distribution”: $P(\lambda):=1/(\beta*\lambda)$ for $\lambda\in(\delta*A(\beta)*e^{-\beta},\delta*A(\beta))$, where $\beta>0$ is a parameter controlling the spikiness and $A(\beta):=\beta/(1-e^{-\beta})$.
> > >
> > > **Experiment 1:** $\beta=20$. $n=10^5$. $m=\lceil(n*1.01)\rceil $. The signal is iid standard Gaussian. The results are obtained via 1000 independent runs. Note that $\delta\approx1.01$ is very close to the information theoretic threshold for $f_1$ and $f_2$. This experiment demonstrates that with spiky enough spectrum the GLM-EP algorithm can indeed approach (at large system size) the information theoretic lower bound.
> > >
> > > | $f_1$ | $t=1$ | $t=2$| $t=3$ | $t=4$| $t=5$ | $t=6$| $t=7$ | $t=8$|
> > > | ----| ----| ----| ----| ----| ----| ----| ----| ----|
> > > | MSE (SE theory)  | 0.9491  |  0.8396  |  0.7050  |  0.5472  |  0.3575  |  0.1332  |  0.0044  |  0.0001 |
> > > | MSE (mean)  | 0.9479 |  0.8384 |  0.7032  |  0.5445  |  0.3542  |  0.1285  |  0.0040  |  0.0001 |
> > > | MSE (std)  | 0.0020  |  0.0042  |  0.0044  |  0.0057  |  0.0069  |  0.0089  |  0.0012  |  0.0000 |
> > >
> > > | $f_2$ | $t=1$ | $t=2$| $t=3$ | $t=4$| $t=5$ | $t=6$| $t=7$ | $t=8$| $t=9$| $t=10$| $t=11$| $t=12$|
> > > | ----| ----| ----| ----| ----| ----| ----| ----| ----| ----| ----| ----| ----|
> > > | MSE (SE theory)  | 0.9632  |  0.9033  |  0.8660 |   0.8433   | 0.8204  |  0.7879 |   0.7295 |  0.6059  |  0.4280  |  0.2134  |  0.0193  |  0.0004 |
> > > | MSE (mean)  | 0.9627  |  0.9028  |  0.8659  |  0.8431 |   0.8200  |  0.7868  |  0.7270 |   0.5999  |  0.4207  |  0.2057  |  0.0222  |  0.0004 |
> > > | MSE (std)  | 0.0022 |  0.0032  |  0.0031  |  0.0034 |   0.0046  | 0.0076  |  0.0151  |  0.0311  |  0.0378  |  0.0446  |  0.0170  |  0.0004 |
> > >
> > > **Experiment 2:** $f=f_1$ (i.e., phase retrieval). $n=2000$. Other settings are the same as in experiment 1. (We only provide the simulation results for $f_1$ for this experiment.)
> > >
> > > | $f_1$| $t=1$ | $t=2$| $t=3$ | $t=4$| $t=5$ | $t=6$| $t=7$ | $t=8$|
> > > | ----| ----| ----| ----| ----| ----| ----| ----| ----|
> > > | MSE (SE theory)  | 0.9491  |  0.8396  |  0.7050  |  0.5472  |  0.3575  |  0.1332  |  0.0044  |  0.0001 |
> > > | MSE (mean)  | 0.9584  |  1.2574  |  0.7225 |   0.5709  |  0.3816  |  0.1602  |  0.0266  |  0.0020 |
> > > | MSE (std)  | 0.0370  |  1.4092 |   0.0531   | 0.0818  |  0.0844  |  0.1048   | 0.0852   | 0.0163 |
> > >
> > > **Experiment 3:** $f=f_3$ (1-bit CS). $n=10^5$. $\beta = 0$ (i.e., column-orthogonal matrix). The MSE is averaged over 100 independent runs. Number of iterations: 20.
> > >
> > > | $\delta=1.5$ | $\delta=2$ | $\delta=2.5$ | $\delta=3$ |
> > > | ----| ----| ----| ----|
> > > | 0.2771 |   0.2091  |  0.1622 |   0.1286 |
> > >
> > > **Experiment 4:** Noisy setting. $f = f_1$ (phase retrieval). $\delta=1.1$. $n=10^5$. $\beta=10$. The MSE is averaged over 100 independent runs. Number of iterations: 10.
> > >
> > > | SNR: 30dB | SNR: 35dB  | SNR: 40dB | SNR: 45dB | SNR: 50dB |
> > > | ----| ----| ----| ----| ----|
> > > | 1.2889e-01 | 5.9298e-02 | 2.1847e-02 | 6.9412e-03 | 2.1481e-03 |
> > >
> > > **Experiment 5:** A phase diagram plot for $f_1$ and $f_2$. We will not list the detailed results here due to space limitation.

---

### Official Review · Reviewer_d33P · 2021-07-16

**Rating:** 6
**Confidence:** 4

**Summary:**

The paper considers a non-linear inverse problem y = f(Ax). The goal is to recover the signal x from the observations y, where A is the sensing matrix and f is component-wise non-linear function. The paper studies the impact of the sensing matrix spectrum and f properties on the recovery performance. Their theorems show that depending on f, the spikiness of the sensing spectrum may improve or worsen the expectation propagation algorithm recovery.

**Ethical Concerns:**

No.

**Limitations And Societal Impact:**

Limitations are breifly discussed in the conclusion. However, no societal impact is provided.

**Main Review:**

Non-linear inverse problems is of interest to the society. The paper takes an interesting approach to study the recovery using the spectrum of the sensing matrix and the property of non-linear function. Proper citations and explanation is provided on the assumptions made. The paper proposes several theorems such as the impact of the sensing matrix spectrum on signal recovery. However, the main concern is that there is no numerical results to support and verify the theorems. In the absence of such numerical experiments showing the recovery performance as f, the size of A (i.e., m and n) and its spectrum changes, paper lacks the NeurIPS publication standards. Below are minor comments.

- The wording and the organization need improvement.

- Sufficient related work is provided. For better clarity and organization, I suggest that the content related to the author's work in Section 1.2 to be moved to Section 1.1, and leave Section 1.2 to focus on related works.

- Abbreviations (e.g., MMSE and AWGN) should be defined at the first time that they are used.

- The statement at line 109-110 is repeated multiple times within the paper. I recommend to emphasize only once or twice.

- For better clarity, a brief notation explanation is recommended (e.g., capital letters stands for ..., bold letters stands for ...)

- Recommend to show Section 3.1 as an "Algorithm".

- Line 262, There are typos (e.g., "is is ..").

**Time Spent Reviewing:**

8

---

> ### Author Response · Authors · 2021-08-09
> **Response to Reviewer d33P**
>
> We thank the reviewer for the detailed comments. We plan to include the following numerical results in a revised version of this paper:
>
> - Simulation results of the EP algorithm under various choices of $f$. In particular, we will demonstrate that, with proper design of $\mathbf{A}$, the EP algorithm can indeed perform very close to the information theoretic limit for phase retrieval (say, $\delta=1.01$).
>
> - Simulation results of the EP algorithm for the 1-bit CS problem.
>
> - Plots of the phase transition under various choices of $f$ and spectrum spikeness.
>
> - Simulation results of the EP algorithm under noisy settings.

---

> > ### Comment · Reviewer_d33P · 2021-08-23
> > **Request more Details on the Numerical Experiments from the Authors**
> >
> > I thank the authors for their response. The abovementioned plan for the numerical results is appealing. However, it does not explain the results. Could the authors provide details of the numerical results? For example,
> >
> > 1. What are the functions $f$ and the range size of $\bf A$ in the new experiment?
> > 2. How is the performance in all their listed experiments?
> > 3. What is the effect of noise? What is the SNR in the noisy setting?
> > 4. Any discussion on explaining the new results?
> >
> > Given the information provided in the authors' response, I do not find the author's response satisfactory. The experiments seem more like a plan rather than obtained results.

---

> > > ### Author Response · Authors · 2021-08-27
> > > **Response to Reviewer d33P**
> > >
> > > We thank the reviewer for the detailed comments.
> > >
> > > - **Response to Q1:** We tested three $f$: (1) $f_1(z)=|z|$; (2): $f_2(z) = |z| * I_{|z|<1} + (|z|-1) * I_{|z|>=1}$, where $I$ is an indicator function; (3) $f_3(z)=sign(z)$; , We tested various settings of $m$ and $n$. Below, we report our numerical results for two settings of $n$ (with $\delta:=m/n$ fixed): $n = 10^5$ and $n=2*10^3$.
> > >
> > > - **Response to Q2:** We list some experimental results (relative MSE performances of the GLM-EP algorithm) below. In these experiments, the spectrum of the sensing matrix is drawn from a "geometric distribution”: $P(\lambda):=1/(\beta*\lambda)$ for $\lambda\in(\delta*A(\beta)*e^{-\beta},\delta*A(\beta))$, where $\beta>0$ is a parameter controlling the spikiness and $A(\beta):=\beta/(1-e^{-\beta})$.
> > >
> > > **Experiment 1:** $\beta=20$. $n=10^5$. $m=\lceil(n*1.01)\rceil $. The signal is iid standard Gaussian. The results are obtained via 1000 independent runs. Note that $\delta\approx1.01$ is very close to the information theoretic threshold for $f_1$ and $f_2$. This experiment demonstrates that with spiky enough spectrum the GLM-EP algorithm can indeed approach (at large system size) the information theoretic lower bound.
> > >
> > > | $f_1$ | $t=1$ | $t=2$| $t=3$ | $t=4$| $t=5$ | $t=6$| $t=7$ | $t=8$|
> > > | ----| ----| ----| ----| ----| ----| ----| ----| ----|
> > > | MSE (SE theory)  | 0.9491  |  0.8396  |  0.7050  |  0.5472  |  0.3575  |  0.1332  |  0.0044  |  0.0001 |
> > > | MSE (mean)  | 0.9479 |  0.8384 |  0.7032  |  0.5445  |  0.3542  |  0.1285  |  0.0040  |  0.0001 |
> > > | MSE (std)  | 0.0020  |  0.0042  |  0.0044  |  0.0057  |  0.0069  |  0.0089  |  0.0012  |  0.0000 |
> > >
> > > | $f_2$ | $t=1$ | $t=2$| $t=3$ | $t=4$| $t=5$ | $t=6$| $t=7$ | $t=8$| $t=9$| $t=10$| $t=11$| $t=12$|
> > > | ----| ----| ----| ----| ----| ----| ----| ----| ----| ----| ----| ----| ----|
> > > | MSE (SE theory)  | 0.9632  |  0.9033  |  0.8660 |   0.8433   | 0.8204  |  0.7879 |   0.7295 |  0.6059  |  0.4280  |  0.2134  |  0.0193  |  0.0004 |
> > > | MSE (mean)  | 0.9627  |  0.9028  |  0.8659  |  0.8431 |   0.8200  |  0.7868  |  0.7270 |   0.5999  |  0.4207  |  0.2057  |  0.0222  |  0.0004 |
> > > | MSE (std)  | 0.0022 |  0.0032  |  0.0031  |  0.0034 |   0.0046  | 0.0076  |  0.0151  |  0.0311  |  0.0378  |  0.0446  |  0.0170  |  0.0004 |
> > >
> > > **Experiment 2:** $f=f_1$ (i.e., phase retrieval). $n=2000$. Other settings are the same as in experiment 1. (We only provide the simulation results for $f_1$ for this experiment.)
> > >
> > > | $f_1$| $t=1$ | $t=2$| $t=3$ | $t=4$| $t=5$ | $t=6$| $t=7$ | $t=8$|
> > > | ----| ----| ----| ----| ----| ----| ----| ----| ----|
> > > | MSE (SE theory)  | 0.9491  |  0.8396  |  0.7050  |  0.5472  |  0.3575  |  0.1332  |  0.0044  |  0.0001 |
> > > | MSE (mean)  | 0.9584  |  1.2574  |  0.7225 |   0.5709  |  0.3816  |  0.1602  |  0.0266  |  0.0020 |
> > > | MSE (std)  | 0.0370  |  1.4092 |   0.0531   | 0.0818  |  0.0844  |  0.1048   | 0.0852   | 0.0163 |
> > >
> > > **Experiment 3:** $f=f_3$ (1-bit CS). $n=10^5$. $\beta = 0$ (i.e., column-orthogonal matrix). The MSE is averaged over 100 independent runs. Number of iterations: 20.
> > >
> > > | $\delta=1.5$ | $\delta=2$ | $\delta=2.5$ | $\delta=3$ |
> > > | ----| ----| ----| ----|
> > > | 0.2771 |   0.2091  |  0.1622 |   0.1286 |
> > >
> > > **Experiment 4:** Noisy setting. $f = f_1$ (phase retrieval). $\delta=1.1$. $n=10^5$. $\beta=10$. The MSE is averaged over 100 independent runs. Number of iterations: 10.
> > >
> > > | SNR: 30dB | SNR: 35dB  | SNR: 40dB | SNR: 45dB | SNR: 50dB |
> > > | ----| ----| ----| ----| ----|
> > > | 1.2889e-01 | 5.9298e-02 | 2.1847e-02 | 6.9412e-03 | 2.1481e-03 |
> > >
> > > - **Response to Q3:** Experiment 4 above shows the performance of GLM-EP in the noisy setting. We see that noise degrades the performance gracefully.
> > >
> > > - **Response to Q4:** Here are some of the conclusions of our experiments: Our theoretical results (which are based on asymptotic analysis) are quite accurate for even medium problem sizes (and close to phase transition). For instance, in Experiment 2, $m=2020$ and $n =2000$. Yet, our theoretical results are quite accurate. Also, our simulation results confirm that as predicted by Lemma 7, as the variance of the noise increases the performance degrades slowly.

---

> > > > ### Comment · Reviewer_d33P · 2021-08-29
> > > > **Positive Numerical Results**
> > > >
> > > > I thank the authors for sharing the details of their numerical experiment. The experiments are positive, and I strongly suggest including them in the paper. In addition, I suggest the authors have a more extensive discussion on their results.
> > > >
> > > > I have increased my score accordingly. It would be nice to know if the authors plan to include the above experiments in the main paper or supplementary. With this much modification to the main paper, a suitable option for the paper might be to go over another round of reviews before getting published. However, I will let the AC decide on this last point.

---

> > > > > ### Author Response · Authors · 2021-08-30
> > > > > **numerical results**
> > > > >
> > > > > We thank the reviewer for the suggestion. By making some minor changes to our text we have been able to fit Experiment 1 in the paper after Section 4. We plan to include the rest of the experiments in the supplementary file. Some of the changes we have made to create space for Experiment 1 are: (a) Making Figure 2 slightly smaller. (b) Combining Examples 1,2, and 3 into one Example. (c) Shortening Section 3.1. (d) Inlining some of the equations such as 6(f), (16) and the Equations of Lemma 1. (e) Shortening the Conclusion.

---

### Official Review · Reviewer_7HR7 · 2021-07-30

**Rating:** 7
**Confidence:** 5

**Summary:**

The paper considers a general class of non-linear inverse problems. It first provides a necessary condition for Lipschitz guarantee in terms of information dimension based on nonlinear operations. The authors also consider expectation propagation algorithms and their analysis using state evolution. They introduce the notion of spikiness based on the impact of sensing matrix spectrum on MSE analyzed using SE. An interesting result of the paper is that if the spectrum is spiky enough, the information theoretic bound can be achieved (Theorem 3).


**Main Review:**

Overall, the paper is very well written and I enjoyed reading it. The proofs seem sound, although I glanced over them and their structure. Using Lorenz order for performance analysis of EP seems novel and interesting.

I would like to raise two issues of the paper and am curious about the answers:
- Input structure is overlooked. This is mentioned in the conclusion part too, but I think it deserves more detailed discussions in the main part of the paper. As I argue below, some of the conditions are overly pessimistic.
- The results are purely qualitative. The authors could use the algorithm and plot the phase transitoin for different choice of non-linearity and spikiness. The experiments could help to compare the current result along side similar result on the generalized linear models (see comments below).


More comments:

- How does the result of Theorem 1 compare with Wu-Verdu bounds? As I see it now, the information dimension in Theorem 1 depends merely on the structure of the function f and not the input. Therefore, the theorem cannot give Wu-Verdu’s result as a special case, which is stated in terms of input information dimension. For example, for identity function, delta should be bigger than 1, which is very pessimistic.

- The result of Theorem 1 is also pessimistic for 1-bit compressed sensing. There are already recovery guarantees for 1-bit CS assuming input sparsity for Gaussian and sub-Gaussian matrices (see the survey paper from S. Dirksen). But the current theorem gives delta>infty, which is a sort of impossibility result.

- Following up on 1-bit CS, if d(f(Z))=0 for this case, how can one interpret theorem 3 for EP algorithm? The authors should clarify if we expect to get better non-asymptotic results for EP and therefore using it in this context is still sensible. Having numerical results could help better interpret the results of the paper.

-On the other hand, the result for phase retrieval is optimistic with d(f(Z))=1, which is the same as not having any non-linearity. Having numerical results could significantly help the message of the paper.

-The notion of spikiness is also very interesting but lacks experimental backup. The  authors could design experiments verifying each one the claims for different setups for example 1-bit CS and phase retrieval.

- How does the information dimension bound compares with the recovery guarantee for Generalized lasso in Plan-Vershynin paper?
https://arxiv.org/abs/1502.04071
In that paper, the authors consider similar GLM setup and provide bounds that depend, in a similar fashion, on the non-linearity modulated by a Gaussian random variable (parameters mu and sigma in that paper). The comparison can be very interesting and relevant.


**Time Spent Reviewing:**

2 hours

---

> ### Author Response · Authors · 2021-08-09
> **Response to reviewer 7HR7**
>
> We thank the reviewer for the detailed comment.
>
> - **Input structure is overlooked:** We agree with the reviewer that structured signal recovery (e.g., sparsity) is an important topic. However, we think that the current setup (which is an extension of the phase retrieval problem) is already important for applications and yields some interesting findings.
>
> - **Results are purely qualitative:** We thank the reviewer for this suggestion. We will definitely include more numerical results (possibly in the supplementary material) in a future version. For instance, for the phase retrieval problem, the perfect recovery threshold with a column orthogonal matrix is about $1.6$, while with a spiky enough spectrum this threshold can be as low as $1.01$ (which will be shown in our simulation results). We will also provide an extensive experimental study on the phase transition under other choices of $f$ and spectra.
>
> - **Comparison of Theorem 1 to Wu-Verdu:** Theorem 1 does not cover Wu-Verdu's result as a special case. The reason is that we did not impose any structural assumption on the signal. (More specifically, we assume the signal distribution to be absolutely continuous w.r.t. Lebesgue measure.) It is expected that by imposing a structural assumption (e.g., sparsity) on the signal, both the fundamental limit and the performance of the EP algorithm would improve. A detailed analysis of this setting is not pursued here though.
>
> - **Concerns about our results on 1-bit CS:** We thank the reviewer for this comment and pointing Dirksen's paper to us. We agree that this issue needs further clarification and we plan to do in a revised version. We first emphasize that our results holds in the asymptotic setting $m,n\to\infty$ with $m/n\to\delta\in(1,\infty)$. Our results imply that zero recovery distortion (as measured by MSE) for 1-bit CS cannot happen at finite $\delta$ (it could decrease to zero as $\delta\to\infty$ though). This is in contrast to other "easier" nonlinear problems (e.g., the phase retrieval problem) where there exist finite thresholds of $\delta$ above which zero recovery MSE can be achieved. Notice that our result does not contradict with existing 1-bit CS results. For instance, Theorem 2 in Diksen's paper analyzes the number of measurements needed to achieve a non-zero target recovery distortion $\rho$, and the bound would blow up to infinity as $\rho\to0$. Further, many order-wise sample complexity results in the literature hide the dependency on the target distortion, which might cause confusion when comparing with our result. We emphasize that sparsity does not make the 1-bit CS problem fundamentally easier if we only consider the linear sparsity regime, i.e., the number of nonzero elements of the signal is of the same order as its ambient dimension. To see this, suppose we have oracle information about the signal support. This would effectively increase our number of measurements by a constant factor (as we assume linear sparsity), and by the result of this paper we cannot achieve zero MSE at finite $\delta$.
>
> - **Numerical results for 1-bit CS and phase retrieval:** We agree with your point and will include numerical results to support our claims. For phase retrieval, our simulation results will show that with properly designed $\mathbf{A}$, the EP algorithm could recover the signal when $\delta\approx1.01$ which is very close to the lower bound.
>
> - **Comparison with Plan and Vershynin:** We think that our result is not directly comparable to that of Plan and Vershynin. Proposition 1 of Plan and Vershynin's paper provides an upper bound for the reconstruction error when $m\gtrsim d$ (corresponding to our $n$), which hides a constant factor. In comparison, our work can provide a precise characterization of this constant, which depends on the information dimension.

---

### Decision · Program_Chairs · 2021-09-27

**Decision:**

Accept (Poster)

**Comment:**

In this paper the author studies the impact of the choice of the sensing matrix spectrum in nonlinear inverse problems. In the first three sections, the authors present the information theoretical limits to perfect recovery, the expectation propagation (EP) message-passing type algorithm to solve such nonlinear inverse problems, and its MSE performance predicted by the state evolution formalism. The main result is presented in the last section, the impact of the spectrum is investigated, both in term of the MSE performance and the measurement threshold for perfect recovery. This impact is shown to depend on the type of nonlinearity used in the inverse problem.

The consensus among the reviewers was  that the paper was overall well-written, and above the acceptance threshold of NeurIPS 2021. It was judge to  have interest for the community of information theory and statistical-physics approaches to learning. A number of comments and criticisms expressed were expressed during the rebuttal, especially concerning the discussion of the results. For instance, the title was judge misleading, as the emphasis on the spectrum should be clearer.

During the rebuttal the authors seem to have answer most of the critics in a satisfactory way to a majority of reviewers. They also shared additional result from their numerical experiments. We believe these new results should definitely be a part of the paper. The reviewers believe that these experiments and their discussion can really improve a lot the paper, and should not only be include din the supplement, but should also be fairly discussed in their new version of the main paper, as they answer a important criticism.

Overall, there is an agreement on the reviewers side (that is, to all but one) to accept the paper for publication at Neurips. Given the good grading, and the fact that authors successfully answered most criticisms from the referee, I therefore recommend acceptance to Neurips, and that the author proceed to the promised change to the paper.